# Privacy Amplification by Structured Subsampling for Deep Differentially Private Time Series Forecasting

Jan Schuchardt [1][†]  Mina Dalirrooyfard [2]  Jed Guzelkabaagac [1]
Anderson Schneider [2]  Yuriy Nevmyvaka [2]  Stephan Günnemann [1]

## Abstract

Many forms of sensitive data, such as web traffic, mobility data, or hospital occupancy, are inherently sequential. The standard method for training machine learning models while ensuring privacy for units of sensitive information, such as individual hospital visits, is differentially private stochastic gradient descent (DP-SGD). However, we observe in this work that the formal guarantees of DP-SGD are incompatible with time series specific tasks like forecasting, since they rely on the *privacy amplification* attained by training on small, unstructured batches sampled from an unstructured dataset. In contrast, batches for forecasting are generated by (1) sampling sequentially structured time series from a dataset, (2) sampling contiguous subsequences from these series, and (3) partitioning them into context and ground-truth forecast windows. We theoretically analyze the privacy amplification attained by this *structured subsampling* to enable the training of forecasting models with sound and tight event- and user-level privacy guarantees. Towards more private models, we additionally prove how data augmentation amplifies privacy in self-supervised training of sequence models. Our empirical evaluation demonstrates that amplification by structured subsampling enables the training of forecasting models with strong formal privacy guarantees.

## 1. Introduction

The need for privacy in Machine Learning (ML) tasks is becoming more apparent every day with an ongoing stream of studies on the privacy risks of ML ((Rigaki & Garcia, 2023)) and new methods to tackle these challenges (Liu et al., 2021; Pan et al., 2024). Among these works, Differential Privacy (DP) (Dwork, 2006) plays a particularly prominent role as a formal privacy model and paradigm for privacy protection.

Historically, works on DP primarily focused on privately querying unstructured databases (Dwork et al., 2010), and later works on differentially private ML continued to focus on learning from unstructured datasets (e.g. (Abadi et al., 2016)). Recently, there has been more attention on structured data, such as graphs (Nissim et al., 2007; Kasiviswanathan et al., 2013; Mueller et al., 2022), text (Yue et al., 2023; Charles et al., 2024; Chua et al., 2024), and time series (Mao et al., 2024). Time series in particular are of interest from both a ML and DP perspective. For example, data from traffic sensors (Chen et al., 2001) can be used to train forecasting models for use in tasks like transport planning and logistics (Lana et al., 2018). Simultaneously, traffic data and its downstream use may expose sensitive information such as individual movement profiles (Giannotti & Pedreschi, 2008). However, most studies only focus on releasing time series or statistics thereof, rather than training ML models (e.g. (Shi et al., 2011; Fan & Xiong, 2014)).

A popular algorithm for private learning from unstructured datasets is DP-SGD (Song et al., 2013; Abadi et al., 2016), which is a simple modification of SGD. Given an unstructured set of input–target pairs $x = \{(x_1, o_1), \ldots, (x_N, o_N)\}$, DP-SGD samples a batch $y \subseteq x$, computes clipped per-sample gradients, and adds Gaussian noise. This yields privacy guarantees for insertion/removal or substitution of a single element $(x_n, o_n)$. DP-SGD has also been used to train models for time series (Mercier et al., 2021; Imtiaz et al., 2020; Arcolezi et al., 2022). However, these works directly apply known privacy guarantees for DP-SGD in a black-box manner. This neglects the structured nature of the data and the structured way in which batches are sampled in tasks like time series forecasting. It may thus lead to an under- or over-estimation of privacy.

This work answers the following research question: *How private is DP-SGD when adapted to sequentially structured data, and specifically time series forecasting?*

[1]Technical University of Munich & Munich Data Science Institute [2]Machine Learning Research, Morgan Stanley †Work done during an internship at Morgan Stanley. Correspondence to: Jan Schuchardt <j.schuchardt@tum.de>.

Figure 1: High-level view on batching in global forecasting, which (1) selects one or multiple sequences ("top-level sample"), (2) selects one or multiple contiguous subsequences per sequence ("bottom-level sample"), and (3) partitions these subsequences for self-supervised training ("context–forecast split"). Elements of sensitive information from a short subsequence (red) may appear in the batch multiple times at different positions.

## 1.1. Our contribution

Our main goal is to provide sound and tight bounds on how much private information is leaked when introducing gradient clipping and noise into the training of forecasting models that generalize across multiple time series ("global forecasting" (Januschowski et al., 2020)).

Fig. 1 provides a high-level view of how a single batch is sampled in commonly used forecasting libraries like GluonTS (Alexandrov et al., 2019), and what level of privacy leakage this can cause. Assume we have a dataset $x = \{x_1, x_2, x_3\}$ of three time series and want to protect any length-2 subsequence ("2-event-level privacy" (Kellaris et al., 2014)). First, a subset of time series is selected ("top-level sample"). Then, one or multiple contiguous subsequences are sampled per sequence ("bottom-level sample"). Finally, each subsequence is split into a context window and a ground-truth forecast. As shown in Fig. 1, our length-2 subsequence may appear in multiple subsequences, with each of its element contributing either to the context or the forecast window. Thus, these elements may leak their information through multiple clipped and noised per-subsequence gradients — either as context via the model's computation graph or as ground-truth via the loss function.

This risk of multiple leakage is underestimated if we apply privacy guarantees for standard DP-SGD in a black-box manner (see also Appendix C.3), and overestimated if we assume that every subsequence always contains every piece of sensitive information (e.g. (Arcolezi et al., 2022)).

Our main contributions are that we, for the first time,

- derive event- and user-level privacy guarantees for bottom-level sampling of contiguous subsequences,
- analyze how the strength of these guarantees can be amplified by top-level sampling,
- and prove how data augmentation can exploit the context–forecast split to further amplify privacy.

Beyond these main contributions, our work demonstrates the usefulness of coupling-based subsampling analysis (Balle et al., 2018; Schuchardt et al., 2024), which has thus far only been applied to unstructured subsampling, in analyzing non-standard, structured subsampling schemes.

## 2. Related Work

In this section, we discuss some of the most directly related work, and refer the reader to Appendix A for further details.

**DP Time Series Release.** Koga et al. (2022) and Li et al. (2023) use subsampling to amplify the privacy of differentially private time series release. In particular, Koga et al. (2022) consider a *single time series* in which any individual contributes to a bounded number of steps. They use subsampling in the time domain to reduce the probability of accessing these steps. Note that their sampling distribution ignores temporal structure and yields irregularly sampled time series. Li et al. (2023) combine amplification by subsampling and shuffling on the *dataset level*, i.e., they only randomize which time series is accessed and not which part of the time series. In general, our goal is training private models rather than publishing sanitized data.

**Application of DP-SGD to Time Series.** Various works have applied DP-SGD (Mercier et al., 2021; Imtiaz et al., 2020; Arcolezi et al., 2022) or random input perturbations (Li et al., 2019) in specific domains like healthcare data and human mobility. However, they do not adapt their analysis or algorithms to time series data, and instead use DP-SGD or other mechanisms in a black-box manner. Similarly, some works have applied DP-SGD to generative models for time series (Frigerio et al., 2019; Wang et al., 2020a; Torfi et al., 2022) or applied PATE (Papernot et al., 2017) in conjunction with DP-SGD (Lamp et al., 2024). This paper differs from prior work in that we specifically tailor our analysis to the structured nature of time series and the structured sampling of batches in forecasting.

**Bi-Level Subsampling for LLMs.** Charles et al. (2024) and Chua et al. (2024) use bi-level subsampling schemes for centralized finetuning of language models on the data of multiple users with multiple sensitive records. However, their privacy analysis only leverages the randomness induced by one of the sampling levels. The other level could equivalently be replaced by a deterministic procedure (see Section A for more details). In comparison, we analyze the interplay of the randomness inherent to both levels. Further note that their analysis considers arbitrary records, and is not tailored to the sequential structure of natural language.

# 3. Background and Preliminaries

## 3.1. Differential Privacy

The goal of differential privacy (Dwork, 2006) is to map from a dataset space $\mathbb{X}$ to an output space $\mathbb{O}$ while ensuring indistinguishability of any neighboring pair of datasets $x \simeq x'$ that differ in one unit of sensitive information (e.g., two sets that differ in one element). In the following, we assume $\mathbb{O} = \mathbb{R}^D$. Differential privacy achieves this goal of indistinguishability via randomization, i.e., mapping to outputs via a random mechanism $M : \mathbb{X} \to \mathbb{R}^D$. The random outputs $M(x), M(x')$ are considered indistinguishable if the probability of any event $O \subseteq \mathbb{R}^D$ only differs by a small factor and constant, i.e., $\Pr[M(x) \in O] \leq e^\varepsilon \cdot \Pr[M(x') \in O] + \delta$. This is equivalent to bounding the hockey stick divergence of output distributions $M_x, M_{x'}$ (Barthe & Olmedo, 2013):

**Definition 3.1.** Mechanism $M : \mathbb{X} \to \mathbb{R}^D$ is $(\varepsilon, \delta)$-DP if and only if $\forall x \simeq x' : H_{e^\varepsilon}(M_x || M_{x'}) \leq \delta$ with $H_\alpha(M_x || M_{x'}) = \int_{\mathbb{R}^D} \max\{\frac{\mathrm{d}M_x}{\mathrm{d}M_{x'}}(\boldsymbol{o}) - \alpha, 0\} \cdot \mathrm{d}M_x(\boldsymbol{o})$.

## 3.2. Private Training and Dominating Pairs

In the case of DP-SGD (Song et al., 2013), the mechanism $M : \mathbb{X} \to \mathbb{R}^D$ is a single training step or epoch that maps training samples to $D$ updated model weights (for details, see Section 4). A training run is the repeated application of this mechanism. A central notion for determining privacy parameters $(\varepsilon', \delta')$ of such a *composed* mechanism is that of dominating pairs (Zhu et al., 2022), which fully characterize the tradeoff between $\varepsilon$ and $\delta$ of its *component* mechanisms.

**Definition 3.2.** Distributions $(P, Q)$ are a *dominating pair* for component mechanism $M$ if $\sup_{x \simeq x'} H_\alpha(M_x || M_{x'}) \leq H_\alpha(P || Q)$ for all $\alpha \geq 0$. If the bound holds with equality for all $\alpha \geq 0$, then $(P, Q)$ are a *tight dominating pair*.

Tight dominating pairs optimally characterize the trade-off between DP parameters $(\varepsilon, \delta)$. We will repeatedly show $P$ and $Q$ to be univariate Gaussian mixtures, for which we use the following short-hand (Choquette-Choo et al., 2024).

**Definition 3.3.** The mixture-of-Gaussians distribution with means $\boldsymbol{\mu} \in \mathbb{R}^K$, standard deviation $\sigma \in \mathbb{R}_+$, and weights $\boldsymbol{p} \in [0, 1]^K$ is $MoG(\boldsymbol{\mu}, \boldsymbol{p}, \sigma) = \sum_{k=1}^K \mathcal{N}(\mu_k, \sigma) \cdot p_k$.

Given dominating pairs for each component mechanism, one can determine $\varepsilon'$ and $\delta'$ of the composed mechanism (training run) via *privacy accounting* methods, such as moments accounting (Abadi et al., 2016) or privacy loss distribution accounting (Meiser & Mohammadi, 2018; Sommer et al., 2019), which we explain in more detail in Appendix D.3.

## 3.3. Amplification by Subsampling and Couplings

A key property that enables private training for many iterations with strong privacy guarantees is *amplification by subsampling* (Kasiviswanathan et al., 2011): Computing gradients for randomly sampled batches strengthens differential privacy (Abadi et al., 2016). More generally, one can use a *subsampling scheme* $S : \mathbb{X} \to \mathbb{Y}$ that maps from dataset space $\mathbb{X}$ to a space of batches $\mathbb{Y}$ and an $(\varepsilon', \delta')$-DP *base mechanism* $B : \mathbb{Y} \to \mathbb{R}^D$ that maps these batches to outputs to construct a more private *subsampled mechanism* $M = B \circ S$. Balle et al. (2018) propose the use of *couplings* as a tool for analyzing subsampled mechanisms.

**Definition 3.4.** A coupling $\Gamma$ between distributions $S_x, S_{x'}$ of randomly sampled batches $S(x), S(x') \in \mathbb{Y}$ is a joint distribution on $\mathbb{Y}^2$ whose marginals are $S_x$ and $S_{x'}$.

Intuitively, $\Gamma$ indicates which batches from the support of $S_x$ correspond to which batches from the support of $S_{x'}$ (for a more thorough introduction, see (Villani, 2009)). Balle et al. (2018) prove that any such coupling yields a bound on the divergence of the subsampled output distributions $M_x$ and $M_{x'}$. More recently, Schuchardt et al. (2024) have generalized this tool to enable the derivation of dominating pairs for subsampled mechanisms, which we utilize in our proofs and explain in more detail in Appendix D.1.

## 3.4. Differential Privacy for Time Series

In the following, we consider the domain $\mathbb{A} = \mathbb{R}^L$ of univariate time series of length $L$. We discuss the straightforward generalization of our results to multivariate time series in Appendix I. The goal of DP time series analysis is to compute statistics for a single series $a \in \mathbb{A}$ while protecting short contiguous subsequences ("$w$-event-level privacy" (Kellaris et al., 2014)) or all steps to which an individual contributed ("user-level privacy" (Dwork et al., 2010)). For our deep learning context, we define the dataset space to be the powerset $\mathbb{X} = \mathcal{P}(\mathbb{A})$ and generalize these notions of indistinguishability to datasets as follows:

**Definition 3.5.** Datasets $x = \{x_1, \ldots, x_N\}$ and $x' = \{x'_1, \ldots, x'_N\}$ are $w$-event-level neighboring ($x \overset{w}{\simeq}_E x'$) if they only differ in a single pair of sequences $x_n, x'_n$ that only differ in a range of indices of length $w$, i.e., $x_n[t : t + w - 1] \neq x'_n[t : t + w - 1]$ for some $1 \leq t \leq L$.

**Definition 3.6.** Datasets $x = \{x_1, \ldots, x_N\}$ and $x' = \{x'_1, \ldots, x'_N\}$ are $w$-user-level neighboring ($x \overset{w}{\simeq}_U x'$) if they only differ in a single pair of sequences $x_n, x'_n$ that only differ in $w$ indices, i.e., $||x_n - x'_n||_0 \leq w$.

For example, if our data were the number of patients in $N$ hospitals over $L = 365$ days, then 14-event level privacy would protect a patient's visit to a hospital for up to 14 days while 14-user-level privacy would also protect multiple shorter visits.[1] Depending on the domain, these relations

---

[1]The number of elements $w$ in $w$-user-level privacy is often omitted for historical reasons, but still used in deriving privacy guarantees, see, e.g., $\kappa$ in Table 1 and Fig. 2 of (Mao et al., 2024).

can be made more precise by constraining the magnitude of change (e.g., (Koga et al., 2022)). For instance, an individual can only change the number of patients on a day by $\pm 1$. We refer to this as $(w, v)$-event and $(w, v)$-user-level privacy.

**Definition 3.7.** Consider datasets $x \overset{w}{\simeq}_E x'$ or $x \overset{w}{\simeq}_U x'$ that differ in sequences $x_n, x'_n$. If $\forall n, t : |x_n[t] - x'_n[t]| \leq v$, then we refer to them as $(w, v)$-event and $(w, v)$-user-level neighboring ($x \overset{w,v}{\simeq}_E x'$ and $x \overset{w,v}{\simeq}_U x'$) , respectively.

# 4. Deep Differentially Private Forecasting

Now that we have the language to formally reason about privacy, let us turn to our original goal of training forecasting models. Algorithm 1 describes the use of top- and bottom-level sampling (recall Fig. 1), which we instantiate shortly. Given a dataset of sequences $x = \{x_1, \ldots, x_N\}$, we sample a subset of sequences, and then independently sample $\lambda \in \mathbb{N}$ subsequences from each of them. All subsequences are then aggregated into a single batch $y$. The size of the top-level sample is chosen such that we (up to modulo division) attain a batch size of $\Lambda \in \mathbb{N}$. Algorithm 2 formalizes the splitting of each subsequence $y_i \in y$ into a context and ground-truth forecast window. Unlike in standard training, we clip the gradient of the corresponding loss and add calibrated Gaussian noise with covariance matrix $\sigma^2 C^2 \boldsymbol{I}$. This makes the training step differentially private under insertion/removal or substitution of a batch element (Abadi et al., 2016).

**Contribution.** Importantly, we neither claim the batching procedure nor the noisy training step to be novel in isolation. Our novel contribution lies in analyzing the interesting and non-trivial way in which the components of the batching procedure interact to amplify the privacy of training steps.

---

**Algorithm 1** DP-SGD Epoch for Global Forecasting

---

**Input:** Data $x = \{x_1, \ldots, x_N\}$, context length $L_C$, forecast length $L_F$, expected number of subsequences $\lambda$, expected batch size $\Lambda$, model $f_\theta$, learning rate $\eta$, noise scale $\sigma$, clipping norm $C$
**for** $b \leftarrow 1$ **to** $\lfloor N \cdot \lambda / \Lambda \rfloor$ **do**
    $y \leftarrow \{\}$
    **for** $x_n$ **in** sample_top_level($N, \lambda, \Lambda, b$) **do**
        $y \leftarrow y \cup$ sample_bottom_level($x_n, L_C, L_F, \lambda$)
    $\theta \leftarrow$ noisy_training_step($y, L_C, L_F, \Lambda, f_\theta, \eta, \sigma, C$)
**return** $\theta$

---

**Simplifying assumptions.** For the sake of exposition and to simplify notation, we assume that $L - L_F + 1 \geq L_C + L_F$ and focus on 1-event-level privacy. In Appendix I, we discuss how to easily generalize our guarantees to $w$-event and $w$-user-level privacy with arbitrary $w \in \mathbb{N}$, as well as variable-length and multivariate time series.

---

**Algorithm 2** Noisy Training Step

---

**Input:** Batch of subsequences $y = \{y_1, \ldots, y_I\}$, context length $L_C$, forecast length $L_F$, expected batch size $\Lambda$, model $f_\theta$, learning rate $\eta$, noise scale $\sigma$, clipping norm $C$
$\hat{g} \leftarrow 0$
**for** $y_i \in y$ **do**
    $y_{C,i} \leftarrow y_i[1 : L_C]$                {context window}
    $y_{F,i} \leftarrow y_i[L_C + 1 : L_C + L_F]$     {forecast window}
    $g_i \leftarrow \nabla_\theta \mathcal{L}(f_\theta(y_{C,i}), y_{F,i})$
    $\hat{g} \leftarrow \hat{g} + g_i / \max\{1, \frac{||g_i||_2}{C}\}$    {gradient clipping}
$\tilde{g} \leftarrow \frac{1}{\Lambda}(\hat{g} + \mathcal{N}(\boldsymbol{0}, \sigma^2 C^2 \boldsymbol{I}))$
**return** $\theta - \eta \tilde{g}$

---

### 4.1. Bottom-Level Subsampling

Let us begin by focusing on the amplification attained via bottom-level sampling of temporally contiguous subsequences. To this end, we assume that the top-level sampling procedure simply iterates deterministically over our dataset and yields $\lfloor \Lambda / \lambda \rfloor$ sequences per batch (see Algorithm 3). As our bottom-level scheme, we use Algorithm 4, which samples $\lambda$ subsequences per sequence *with replacement* to achieve a fixed batch size of $\Lambda$. In Appendix E.2, we additionally consider *Poisson sampling*, which independently includes each element at a constant rate. In forecasting frameworks, these methods are also referred to as *number of instances sampling* and *uniform split sampling*, respectively (Alexandrov et al., 2019). In the following, mechanism $M$ refers to a *single epoch* with top-level iteration and bottom-level sampling with replacement.

---

**Algorithm 3** Top-Level Deterministic "Sampling"

---

**Input:** Data $x = \{x_1, \ldots, x_N\}$, expected subsequences $\lambda$, expected batch size $\Lambda$, batch number $b$
$N' \leftarrow \lfloor \Lambda / \lambda \rfloor$                  {"Sample" size}
**for** $n \leftarrow 1 + (b - 1) \cdot N'$ **to** $b \cdot N'$ **do**
    **yield** $x_n$

---

**Algorithm 4** Bottom-Level Sampling with Replacement

---

**Input:** Sequence $x_n$, context length $L_C$, forecast length $L_F$, expected subsequences $\lambda$
$x'_n \leftarrow$ prepend_zeros($x_n, L_C$)          {padding}
$T \leftarrow L - L_F + 1$         {maximum start index}
**for** $j \leftarrow 1$ **to** $\lambda$ **do**
    $t \leftarrow \text{Uniform}(\{1, \ldots, T\})$
    **yield** $x'_n[t : t + L_C + L_F - 1]$     {cropping}

---

**Effect of Number of Subsequences $\lambda$.** Before we proceed to deriving amplification guarantees, note that bi-level subsampling introduces an additional degree of freedom not present in DP-SGD for unstructured data: A batch of

size $\Lambda$ can be composed of many subsequences from few sequences ($\lambda$ large) or few subsequences from many sequences ($\lambda$ small). Intuitively, the latter should be more private, because there are fewer chances to access sensitive information from any specific sequence $x_n$. In fact, we can prove the correctness of this intuition via stochastic dominance of amplification bounds (see Appendix E.3):

**Theorem 4.1.** *Let $P^*(\lambda)$, $Q^*(\lambda)$ be a tight dominating pair of epoch $M$ for bottom-level Poisson sampling or sampling with replacement and $\lambda \in \mathbb{N}$ (expected) subsequences. Then $H_\alpha(P^*(\lambda), Q^*(\lambda))$ is minimized by $\lambda = 1$ for all $\alpha \geq 0$.*

**Guarantees for Optimal $\lambda$.** Based on this result, let us first focus on the case $\lambda = 1$ that minimizes per-epoch privacy leakage. Because we consider subsequences of length $L_C + L_F$, even a single sensitive element of a sequence $x_n$ can contribute to $L_C + L_F$ different subsequences at different positions. Since we deterministically iterate over our dataset, these subsequences can contribute to exactly one training step per epoch. The resultant privacy is tightly bounded by the following result (proof in Appendix E).

**Theorem 4.2.** *Consider the number of sampled subsequences $\lambda = 1$, and let $r = \frac{L_C + L_F}{L - L_F + 1}$ be the probability of sampling a subsequence containing any specific element. Define $P(1) = MoG(\boldsymbol{\mu}, \boldsymbol{p}, \sigma)$ with means $\boldsymbol{\mu} = \begin{bmatrix} 0 & 2 \end{bmatrix}^T$ and weights $\boldsymbol{p} = \begin{bmatrix} 1 - r & r \end{bmatrix}^T$. Further, define per-epoch privacy profile $H(\alpha) = \sup_{x \simeq_E x'} H_\alpha(M_x || M_{x'})$. Then,*

$$H(\alpha) = \begin{cases} H_\alpha(P(1) || \mathcal{N}(0, \sigma)) & \text{if } \alpha \geq 1, \\ H_\alpha(\mathcal{N}(0, \sigma) || P(1)) & \text{if } 0 \leq \alpha < 1. \end{cases}$$

In Appendix E.1.5, we discuss the tight dominating pairs corresponding to this bound. Intuitively, as $r$ decreases, $P(1)$ converges to $\mathcal{N}(0, 1)$ and the hockey stick divergence decreases. This means that the mechanism becomes more private with increasing sequence length $L$ or decreasing context length $L_C$ and forecast length $L_F$.

**Other Guarantees.** In Appendix E, we derive tight dominating pairs for Poisson sampling and $\lambda \geq 1$, as well as dominating pairs for sampling with replacement and $\lambda \geq 1$. The special case of $N = 1$, $\Lambda = \lambda$ is equivalent to sampling from a set in which $L_C + L_F$ elements are substituted, i.e., subsampled group privacy (Ganesh, 2024; Schuchardt et al., 2024; Jiang et al., 2025). Thus far, dominating pairs for sampling with replacement have only been known for individual substitutions, i.e., our group privacy guarantees for sampling with replacement are of interest beyond forecasting.

**Epoch Privacy vs Length.** Despite the optimality guarantee from Theorem 4.1, we need to consider that sampling few subsequences ($\lambda$ small) means that more sequences contribute to a batch ($\lfloor \Lambda / \lambda \rfloor$ large). We thus need more epochs for the same number of training steps, and each epoch has

the potential of leaking private information. In Section 5, we will demonstrate numerically that composing many short, more private epochs ($\lambda = 1$) nevertheless offers stronger privacy for the same number of training steps. As baselines for this experiment, we will use the following *optimistic lower bounds* (proof in Appendix E.1.4):

**Theorem 4.3.** *Consider the number of subsequences $\lambda \geq 1$, and let $r = \frac{L_C + L_F}{L - L_F + 1}$. Define $\underline{P}(\lambda) = MoG(\boldsymbol{\mu}, \boldsymbol{p}, \sigma)$ with means $\boldsymbol{\mu} \in \mathbb{N}_0^{\lambda+1}$ and weights $\boldsymbol{p} \in [0, 1]^{\lambda+1}$ with $\mu_i = 2(i - 1)$ and $p_i = \text{Binomial}(i \mid \lambda, r)$. Further, define per-epoch privacy profile $H(\alpha) = \sup_{x \simeq_E x'} H_\alpha(M_x || M_{x'})$. Then,*

$$H(\alpha) \geq \begin{cases} H_\alpha(\underline{P}(\lambda) || \mathcal{N}(0, \sigma)) & \text{if } \alpha \geq 1, \\ H_\alpha(\mathcal{N}(0, \sigma) || \underline{P}(\lambda)) & \text{if } 0 \leq \alpha < 1. \end{cases}$$

Intuitively, each mixture mean $\mu_i$ corresponds to the event that $i - 1$ subsequences with information of a specific individual are sampled, i.e., more information is leaked.

### 4.2. Top-Level Subsampling

Thus far, we only focused on how randomly selecting subsequences from longer sequences amplifies privacy. However, standard batching procedures for time series forecasting admit two levels of randomness (recall Fig. 1). In the following, let us explore how randomizing which sequences $x_n$ contribute to a batch can further amplify privacy. For this, we use Algorithm 5, which samples without replacement. This will eliminate the chance that any particular sequence $x_n$ can have its information leaked through more than $\lambda$ subsequences per batch. From here on, mechanism $\tilde{M}$ refers to a *single training step* using top-level sampling without replacement and bottom-level sampling with replacement.

---

**Algorithm 5** Top-Level Sampling Without Replacement

**Input:** Data $x = \{x_1, \dots, x_N\}$, expected subsequences $\lambda$, expected batch size $\Lambda$, batch number $b$
$N' \leftarrow \lfloor \Lambda / \lambda \rfloor$            {Sample size}
$\pi \leftarrow \text{random\_permutation}(N)$
**for** $n \leftarrow 1$ **to** $N'$ **do**
    **yield** $x_{\pi(n)}$

---

**Theorem 4.4.** *Consider the number of subsequences $\lambda = 1$ and batch size $\Lambda$. Let $r = \frac{L_C + L_F}{L - L_F + 1}$ and let $\rho = \lfloor \Lambda / \lambda \rfloor / N$ be the probability of sampling any specific sequence. Define $\tilde{P}(1) = MoG(\boldsymbol{\mu}, \tilde{\boldsymbol{p}}, \sigma)$ with $\tilde{\boldsymbol{\mu}} = \begin{bmatrix} 0 & 2 \end{bmatrix}^T$ and $\tilde{\boldsymbol{p}} = \begin{bmatrix} (1 - \rho) + \rho \cdot (1 - r) & \rho \cdot r \end{bmatrix}^T$. Then, per-step privacy profile $\tilde{H}(\alpha) = \sup_{x \simeq_E x'} H_\alpha(\tilde{M}_x || \tilde{M}_{x'})$ fulfills*

$$\tilde{H}(\alpha) = \begin{cases} H_\alpha(\tilde{P}(1) || \mathcal{N}(0, \sigma)) & \text{if } \alpha \geq 1, \\ H_\alpha(\mathcal{N}(0, \sigma) || \tilde{P}(1)) & \text{if } 0 \leq \alpha < 1. \end{cases}$$

Put simply, the probability $\rho$ of sampling a sequence with sensitive information multiplies with the probability $r$ of including this sensitive information in the sampled subsequence. This determines the weight $\rho \cdot r$ of the mixture component with mean 2 that indicates privacy leakage.

**Other guarantees.** In Appendix F, we additionally derive dominating pairs for $\lambda \geq 1$ when using bottom-level sampling with replacement or Poisson sampling. There, $\rho$ still has a similar effect of attenuating privacy leakage.

**Step- vs Epoch-Level Accounting.** While Theorem 4.4 shows that top-level sampling amplifies privacy, it yields bounds for each training step $\tilde{M}$ instead of each epoch $M$ (cf. Theorem 4.2). We need to self-compose these bounds $\lfloor N \cdot \lambda / \Lambda \rfloor$ times to obtain epoch-level guarantees (see Algorithm 1). In Section 5 we confirm that the resulting privacy guarantees can nevertheless be stronger than our original epoch-level guarantee. This observation is consistent with works on DP-SGD for unstructured data that self-compose subsampled mechanisms instead of deterministically iterating over datasets (e.g. (Abadi et al., 2016)).

**Choice of $\lambda$.** As before, we can ask ourselves which number of subsequences per sequence $y \in \mathbb{N}$ we should choose. In bi-level subsampling, there is a more intricate trade-off, because increasing $\lambda$ decreases $\rho$, i.e., strengthens top-level amplification, but weakens bottom-level amplification (recall Theorem 4.1). In Section 5, we demonstrate numerically that $\lambda = 1$ is still preferable under composition. As fair baselines for this experiment, we use *optimistic lower bounds* for $\lambda > 1$ that we derive in Appendix F.4.

### 4.3. Context–Forecast Split

We have already successfully analyzed how top- and bottom-level subsampling interact to amplify the privacy of clipped and noised gradients $g_i = \nabla_\theta \mathcal{L}(f_\theta(y_{C,i}), y_{F,i})$. However, we can use yet another level of forecasting-specific randomness — if we assume that an individual can change each value of a series by at most $v \in \mathbb{R}_+$, i.e., we assume $(w, v)$-event or $(w, v)$-user-level privacy (Definition 3.7). We propose to augment the context and forecast window with Gaussian noise $Z_C \sim \mathcal{N}(\mathbf{0}, \sigma_C^2 \cdot v^2 \cdot \mathbf{I})$, $Z_F \sim \mathcal{N}(\mathbf{0}, \sigma_F^2 \cdot v^2 \cdot \mathbf{I})$:

$$g_i = \nabla_\theta \mathcal{L}(f_\theta(y_{C,i} + Z_C), y_{F,i} + Z_F). \quad (1)$$

Unlike the input perturbations from (Arcolezi et al., 2022) which are an offline pre-processing that privatizes the dataset, Eq. (1) is an online data augmentation that serves as an integral part of the (now continuous) subsampling procedure. In the following, let $\hat{M}$ refer to a *single training step* when combining top-level sampling without replacement, bottom-level sampling with replacement, and Eq. (1).

**Amplification by Data Augmentation.** Intuitively, any element can only contribute to gradient $g_i$ either via context

$y_{C,i}$ or via ground-truth forecast $y_{F,i}$. Even if this element changes by $\pm v$, there is a chance that we sample the same value after adding Gaussian noise, i.e., have zero leakage. In Appendix G, we use conditional couplings in conjunction with the maximal couplings originally used for subsampling analysis by Balle et al. (2018) to formalize "sampling the same value". The following result shows the special case $\sigma_C = \sigma_F$ where the context and forecast noise scale are identical (for the general case, see Theorem G.1).

**Theorem 4.5.** *Consider $\lambda = 1$, batch size $\Lambda$, as well as context and forecast standard deviations $\sigma_C, \sigma_F \in \mathbb{R}_+$ with $\sigma_C = \sigma_F$. Let $r = \frac{L_C + L_F}{L - L_F + 1}$ and $\rho = \lfloor \Lambda / \lambda \rfloor / N$. Define $\hat{P}(1) = MoG(\boldsymbol{\mu}, \tilde{\boldsymbol{p}}, \sigma)$ with means $\hat{\boldsymbol{\mu}} = \begin{bmatrix} 0 & 2 \end{bmatrix}^T$ and weights $\tilde{\boldsymbol{p}} \in [0, 1]^2$, with $p_1 = 1 - p_2$ and*

$$p_2 = \rho \cdot r \cdot \mathrm{TVD}\left(\mathcal{N}(0, \sigma_F), \mathcal{N}(1, \sigma_F)\right),$$

*with total variation distance $\mathrm{TVD}(P, Q) = H_1(P \| Q)$. Then, $\hat{H}(\alpha) = \sup_{x \simeq_E x'}^{1, v} H_\alpha(\hat{M}_x \| \hat{M}_{x'})$ fulfills*

$$\hat{H}(\alpha) \leq \begin{cases} H_\alpha(\hat{P}(1) \| \mathcal{N}(0, \sigma)) & \text{if } \alpha \geq 1, \\ H_\alpha(\mathcal{N}(0, \sigma) \| \hat{P}(1)) & \text{if } 0 \leq \alpha < 1. \end{cases}$$

Intuitively, this shows that Gaussian data augmentation has a similar effect to subsampling in that it shifts probability mass to the mixture component that indicates zero leakage. Unlike top- and bottom-level subsampling, which are interdependent through number of subsequences $\lambda$, this additional layer of amplification can be independently controlled through $\sigma_C$ and $\sigma_F$. Amplification by data augmentation thus expands the space of possible privacy–utility trade-offs.

**Amplification by Label Perturbation.** An interesting special case is $0 = \sigma_C < \sigma_F$, where privacy is only further amplified when sensitive information appears as a ground-truth forecast, i.e., we are in the "label privacy" (Chaudhuri & Hsu, 2011) setting. A standard technique for deep learning with label privacy is using random label perturbations as an offline pre-processing step (Ghazi et al., 2021). Our results in Appendix G show for the first time how online label perturbations can amplify privacy in settings where we randomly switch between feature- and label-privacy, such as self-supervised (pre-)training of sequence models.

### 4.4. Additional Inference-Time Privacy

Like other works on DP-SGD, we focus on ensuring privacy of parameters $\theta$ to guarantee that information from any training sequence $x_n$ does not leak when releasing the model $f_\theta$ or forecasts $f_\theta(x_m)$ for other sequences $x_m$. In Appendix H, we additionally explore the use of input perturbations in combination with subsampling and imputation to ensure privacy for elements of $x_n$ when releasing forecast $f_\theta(x_n)$.

### 4.5. Limitations and Future Work

Since we are first to analyze forecasting-specific subsampling, there are still opportunities for improvement, namely by tightening our guarantees for (1) bottom-level sampling with replacement and $\lambda > 1$, (2) top-level sampling without replacement and $\lambda > 1$, and potentially (3) amplification by augmentation. A somewhat subtle challenge when considering top-level sampling without replacement, bottom-level sampling with replacement, and number of subsequences $\lambda > 1$ relates to the fact that subsampling analysis (see Appendix D.1) revolves around defining correspondences between batches from the support of different distributions (via couplings) and determining their similarity (via induced distances): Assume two datasets $x = \overset{1}{\simeq}_E x'$ differ by one sensitive element $x_n[t] \neq x'_n[t]$ in a pair of sequences $x_n \neq x'_n$. Then, any batch that does not contain subsequences of $x_n$ or $x'_n$ will differ in *at least* $\lambda$ subsequences from any batch that contains subsequences of $x_n$ or $x'_n$ – no matter if the sensitive element is contained once, twice, $\lambda$ times or zero times. For this reason, we posit that the lower bound from Theorem F.7 (even though it appears like a natural generalization of Theorem 4.4) is *not* a tight upper bound, and advise future work to not overlook this issue.

While we numerically investigate the discussed trade-offs in parameterizing our subsampling scheme, future work may also want to investigate them analytically. In particular, central limit theorems of composition (Sommer et al., 2019; Dong et al., 2022) could potentially be used to understand why $\lambda = 1$ offers the best privacy for an increasingly wide range of $\varepsilon$ as the number of composition steps grows.

Finally, the connection between event and $w$-event-level privacy and token- or sentence-level private language modeling (Hu et al., 2024) (i.e., autoregressive probabilistic forecasting for discrete-valued time series) is immediate and should be explored in future work. Specifically, time series in a global forecasting dataset correspond to documents in a corpus. Sensitive events correspond to sensitive sentences. The notion of context windows is shared between both domains. Ground-truth forecasts correspond to ground-truth sequences that can, for example, be learned via teacher-forcing. Beyond event-level privacy, the derived bounds are also applicable to user-level private learning with bi-level subsampling under a fixed number of substitutions per user (such as user-level private LLM fine-tuning, see discussion of (Chua et al., 2024; Charles et al., 2024) in Appendix A).

## 5. Experimental Evaluation

We already achieved our primary objective of deriving time series specific subsampling guarantees for DP-SGD adapted to forecasting. Our goal for this section is to investigate the trade-offs we discovered in discussing these guarantees.

In addition, we train common probabilistic forecasting architectures on standard datasets to verify the feasibility of training deep differentially private forecasting models while retaining meaningful utility. The full experimental setup is described in Appendix B. An implementation will be made available at cs.cit.tum.de/daml/dp-forecasting.

### 5.1. Trade-Offs in Structured Subsampling

For the following experiments, we assume that we have $N = 320$ sequences, batch size $\Lambda = 32$, and noise scale $\sigma = 1$. We further assume $L = 10(L_F + L_C) + L_F - 1$, so that the chance of bottom-level sampling a subsequence containing any specific element is $r = 0.1$ when choosing $\lambda = 1$ as the number of subsequences. In Appendix C.1, we repeat all experiments with a wider range of parameters. All results are consistent with the ones shown here.

**Number of Subsequences $\lambda$.** The first trade-off is inherent to bi-level subsampling: One can achieve the same batch size $\Lambda$ with different $\lambda$, each leading to different top- and bottom-level amplification. We claim that $\lambda = 1$ (i.e., maximum bottom-level amplification) is preferable. For a fair comparison, we compare our provably tight guarantee for $\lambda = 1$ (Theorem 4.4) with optimistic lower bounds for $\lambda > 1$ (Theorem F.7) instead of our sound upper bounds (Theorem F.1), i.e., we make the competitors stronger. As shown in Fig. 2a, $\lambda = 1$ only has smaller $\delta(\varepsilon)$ for $\varepsilon \geq 10^{-1}$ when considering a single training step. However, after 100-fold composition, $\lambda = 1$ achieves smaller $\delta(\varepsilon)$ even in $[10^{-3}, 10^{-1}]$ (see Fig. 2b). Our explanation is that $\lambda > 1$ results in larger $\delta(\varepsilon)$ for large $\varepsilon$, i.e., is more likely to have a large privacy loss. Because the privacy loss of a composed mechanism is the sum of component privacy losses (Sommer et al., 2019), this is problematic when performing multiple training steps. We will thus later use $\lambda = 1$ for training.

**Step- vs Epoch-Level Accounting.** Next, we show the benefit of top-level sampling sequences (Theorem 4.4) instead of deterministically iterating over them (Theorem 4.2), even though we risk privacy leakage at every training step. For our parameterization and $\lambda = 1$, top-level sampling with replacement requires 10 compositions per epoch. As shown in Fig. 3, the resultant epoch-level profile is nevertheless smaller, and remains so after 10 epochs. This is consistent with any work on DP-SGD (e.g., (Abadi et al., 2016)) that uses subsampling instead of deterministic iteration.

**Epoch Privacy vs Length.** In Appendix C.1.4 we additionally explore the fact that, if we wanted to use deterministic top-level iteration, the number of subsequences $\lambda$ would affect epoch length. As expected, we observe that composing many private mechanisms ($\lambda = 1$) is preferable to composing few much less private mechanisms ($\lambda > 1$) when considering a fixed number of training steps.

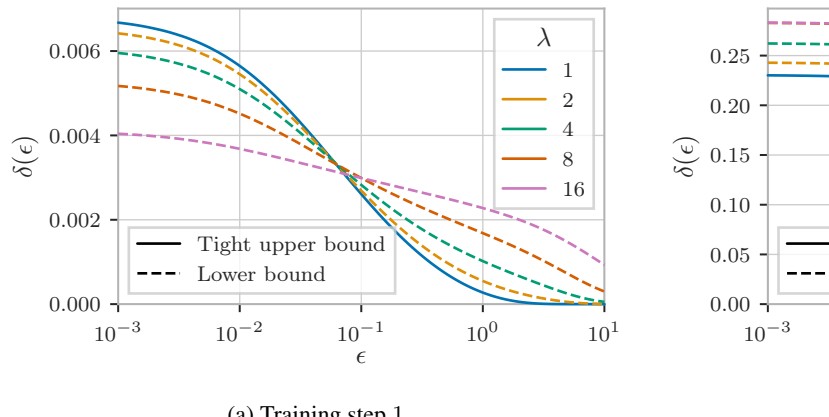 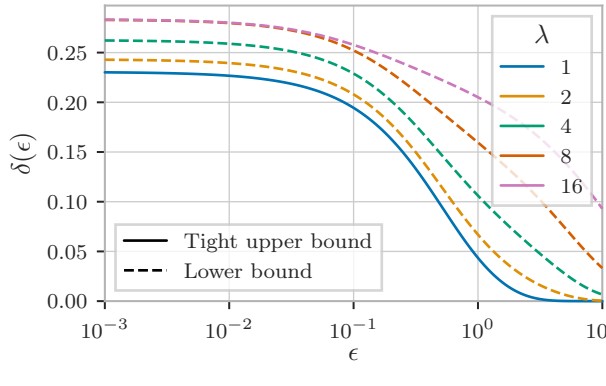

(a) Training step 1                         (b) Training step 100

Figure 2: Top-level WOR and bottom-level WR sampling under varying number of subsequences. Under composition, even optimistic lower bounds (Theorem F.7) indicate worse privacy for $\lambda > 1$ than our tight upper bound for $\lambda = 1$ (Theorem 4.4).

**Amplification by Label Perturbation.** Finally, because the way in which adding Gaussian noise to the context and/or forecast window amplifies privacy (Theorem G.1) may be somewhat opaque, let us consider top-level sampling without replacement, bottom-level sampling with replacement, $\lambda = 1$, $\sigma_C = 0$, and varying label noise standard deviations $\sigma_F$. As shown in Fig. 4, increasing $\sigma_F$ has the same effect as letting the forecast length $L_C$ go to zero, i.e., eliminates the risk of leaking private information if it appears in the forecast window. Of course, this data augmentation will have an effect on model utility, which we investigate shortly.

### 5.2. Application to Probabilistic Forecasting

Our previous experiments show how different parameterizations of the subsampling scheme affect the privacy of DP-SGD applied to time series forecasting. However, altering how batches are sampled will affect the training dynamics of forecasting models. Parameterizations that offer strong privacy (small $\lambda$ and $\Lambda$) could potentially result in low model utility. The following experiments serve to show that we can in fact train neural forecasting models with strong privacy guarantees while retaining better utility than non-neural methods. In short: DP-SGD for time series forecasting offers a good privacy–utility trade-off.

Table 1: Average CRPS on `traffic` for $\delta = 10^{-7}$. Seasonal, AutoETS, and models with $\varepsilon = \infty$ are without noise.

| MODEL | $\varepsilon = 0.5$ | $\varepsilon = 1$ | $\varepsilon = 2$ | $\varepsilon = \infty$ |
|---|---|---|---|---|
| SIMPLEFF | 0.207 | 0.195 | 0.193 | 0.136 |
| DEEPAR | **0.157** | **0.145** | **0.142** | **0.124** |
| ITRANSF. | 0.211 | 0.193 | 0.188 | 0.135 |
| DLINEAR | 0.204 | 0.192 | 0.188 | 0.140 |
| SEASONAL | 0.251 | 0.251 | 0.251 | 0.251 |
| AUTOETS | 0.407 | 0.407 | 0.407 | 0.407 |

**Datasets, Models, and Metrics.** We consider three standard benchmarks: `traffic`, `electricity`, and `solar_10_minutes` as used in (Lai et al., 2018). We further consider four common architectures: A two-layer feed-forward neural network ("SimpleFeedForward"), a recurrent neural network ("DeepAR" (Salinas et al., 2020)), an encoder-only transformer ("iTransformer" (Liu et al., 2024)), and a refined feed-forward network proposed to compete with attention-based models ("DLinear" (Zeng et al., 2023)). We let these architectures parameterize elementwise $t$-distributions to obtain probabilistic forecasts. We measure the quality of these probabilistic forecasts using continuous ranked probability scores (CRPS), which we approximate via mean weighted quantile losses (details in Appendix B.3). As a reference for what constitutes "meaningful utility", we compare against seasonal naïve forecasting and exponential smoothing ("AutoETS") without introducing any noise. All hyperparameter values are specified in Appendix B. All experiments are repeated with 5 random seeds.

**Event-Level Privacy.** Table 1 shows CRPS of all models on the `traffic` test set when setting $\delta = 10^{-7}$, and training on the training set until reaching a pre-specified $\varepsilon$ with 1-event-level privacy. For the other datasets and standard deviations, see Appendix C.2.1. The column $\varepsilon = \infty$ indicates non-DP training. As can be seen, models can retain much of their utility and outperform the baselines, even for $\varepsilon \leq 1$ which is generally considered a small privacy budget (Ponomareva et al., 2023). For instance, the average CRPS of DeepAR on the traffic dataset is 0.124 with non-DP training and 0.157 for $\varepsilon = 0.5$. Note that, since all models are trained using our tight privacy analysis, which specific model performs best on which specific dataset is orthogonal to our contribution.

**Other results.** In Appendix C.2.2 we additionally train probabilistic forecasting models with $w$-event and $w$-user

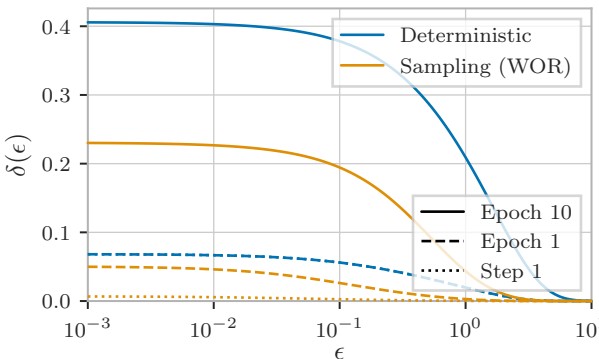

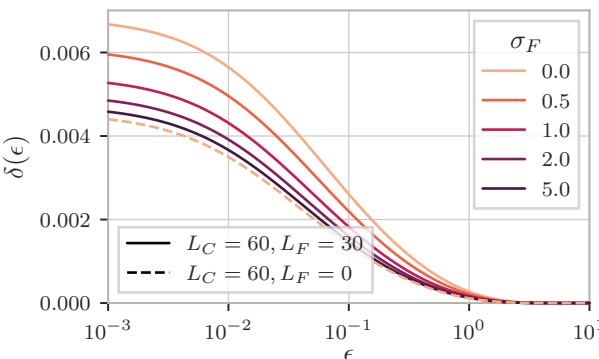

Figure 3: Top-level deterministic iteration (Theorem 4.2) vs top-level WOR sampling (Theorem 4.4) for $\lambda = 1$. Sampling is more private despite requiring more compositions.

Figure 4: Varying label noise $\sigma_F$ for top-level WOR and bottom-level WR (Theorem G.1) with $\sigma_C = 0, \lambda = 1$. Increasing $\sigma_F$ is equivalent to decreasing forecast length.

privacy. In Appendix C.2.3, we demonstrate that label perturbations can further improve the privacy–utility trade-off. In Appendix C.2.4, we perform non-private training with different subsampling schemes to verify that the good privacy–utility trade-off achieved via bi-level subsampling is not simply due to improvements in raw predictive performance. All results confirm that our guarantees for DP-SGD adapted to forecasting are strong enough to enable private training while retaining meaningful levels of utility.

# 6. Conclusion

In this work, we answer the question how DP-SGD can be adapted to time series forecasting while accounting for domain- and task-specific aspects. We derive privacy amplification guarantees for sampling contiguous subsequences and for combining this bottom-level sampling with top-level sampling of sequences, and additionally prove that partitioning subsequences into context and ground-truth forecasts enables privacy amplification by data augmentation. We further identify multiple trade-offs inherent to bi-level subsampling which we investigate theoretically and/or numerically. Finally, we confirm empirically that it is feasible to train differentially private forecasting models while retaining meaningful utility. Adapting our results to natural language represents a promising direction for future work towards trustworthy machine learning on structured data.

# Acknowledgements

We would like to thank Dominik Fuchsgruber, Jonas Dornbusch, Marcel Kollovieh, and Leo Schwinn for proofreading the manuscript. We are also grateful to Marcel Kollovieh, Kashif Rasul, and Marin Biloš for providing valuable insights into practical aspects of training forecasting models. This research was partially funded by the German Research Foundation (grant GU 1409/4-1) and by the DAAD program Konrad Zuse Schools of Excellence in Artificial Intelligence, sponsored by the German Federal Ministry of Education and Research.

# Impact Statement

This work is specifically aimed at mitigating negative societal impact of machine learning by provably ensuring that forecasts made by a model trained on sensitive data does not allow for any form of membership inference or reconstruction attack. As such, even though there are many potential societal consequences of our work, we do not feel that any of them must be specifically highlighted here.

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

# A. Additional Related Work

Below we discuss additional related work in differential privacy and sequential data, and how our work differentiates from them. Note that there are works on privacy for time series outside differential privacy such as (Falcetta & Roveri, 2022; Yue et al., 2021; Shi et al., 2011) that use homomorphic encryption, but that is outside the scope of our paper.

## A.1. Differentially Private Time Series Release

Publishing sanitized time series data has been the most studied application of differential privacy to temporally structured data. Most often, the goal is to release event-level differentially private time series, which are often an aggregate statistic of multiple private time series (Shi et al., 2011; Fan & Xiong, 2014; Wang et al., 2016; 2020b; Zhang et al., 2017; Fioretto & Van Hentenryck, 2019; Kellaris et al., 2014; Mao et al., 2024; Katsomallos et al., 2019). The high level approach in most of the mentioned works is to sample a subset of time steps, add noise, and use these samples to impute missing time steps. Here, sampling helps in improving utility by using the fact that they do not add noise to *all* data points. However, the sampling is not used for reducing the sensitivity, i.e., privacy amplification. Variants of this method include adaptive sampling based on an error estimate (Fan & Xiong, 2014), releasing DP time series over infinite time horizon (Kellaris et al., 2014), and releasing histograms only when there has been a significant change (Li et al., 2015). Zhang et al. (2022) uses learned autocorrelation in the data instead of subsampling to publish sanitized time series under continual observation.

The fundamental difference between this class of work and our paper is that we are interested in training models in a DP manner rather than publishing sanitized data, and the fact that we subsample for the sake of privacy amplification.

## A.2. DP-SGD for NLP/LLMs

Time series data and text data are similar in their temporal structure, and for that we give an overview of most relevant works on differential privacy in natural language models and LLMs. Perhaps the best starting point is Table 1 in (Hu et al., 2024), which lists over two dozen works that applied gradient perturbation (DP-SGD) for differentially private training in NLP. All of these works are categorized as providing *sample-level* or *user-level* privacy (not to be confused with user-level privacy in time series). That is, they consider natural language datasets as an unstructured set of atomic objects. Works that explicitly consider the sequential structure within these objects (categorized as token-, word-, or sentence-private) exclusively use random input perturbations or ensemble-based methods. Of course, this does not in any way mean that the resultant privacy guarantees are invalid or too pessimistic (under their considered neighboring relation).

These works include, for example, DP-SGD fine-tuning of LLMs (Yue et al., 2023; Carranza et al., 2024; Lee & Søgaard, 2023; Wunderlich et al., 2022). There are also various works on DP federated learning for natural language learning models (McMahan et al., 2018; Ramaswamy et al., 2020). See (Hu et al., 2024) for a broader overview.

**Bi-Level Subsampling for LLMs** Charles et al. (2024) and (Chua et al., 2024) both consider two specific algorithms for differentially private training in a setting where $N$ data holders each have an arbitrary number of records and one wants to ensure privacy for insertion or removal of a data holder. These two algorithms are referred to as DP-SGD-ELS and DP-SGD-ULS by (Charles et al., 2024) and "Group Privacy" and "User-wise DP-SGD" by (Chua et al., 2024). In DP-SGD-ELS, one randomly samples a fixed number $G_{\text{ELS}}$ of samples to construct a new composite dataset of $N \cdot K$ records. This reduces the problem of fine-tuning with user-level privacy to that of DP-SGD training with group privacy (Ganesh, 2024). In DP-SGD-ULS, one randomly samples a variable-sized set of users $U$ via Poisson sampling. For each user in $U$, one then randomly samples $G_{\text{ULS}}$ records, computes an average per-user gradient, clips the per-user gradient, accumulates them, and adds noise. This reduces the problem of fine-tuning with user-level privacy to that of standard DP-SGD training, where one user behaves like one record in standard DP-SGD. Importantly, the sampling of records from users only serves to bound their number to $G_{\text{ELS}}$ or $G_{\text{ULS}}$. Equivalently, one could use a deterministic procedure that returns the first $G_{\text{ELS}}$ or $G_{\text{ULS}}$ records of each user. Using our terminology, these works do not analyze any form of amplification attained via the randomness in their bottom-level sampling procedure.

Note that *this is not a limitation in the considered setting of Charles et al. (2024) and (Chua et al., 2024)*, as one has to make the worst case assumption that each inserted user has arbitrary worst-case records. Our results on top- and bottom-level subsampling instead correspond to what is essentially user-level privacy where we have a fixed number of users $N$ users, and $L_C + L_F$ records of a single user are substituted, meaning there is some chance of accessing non-substituted records.

# B. Experimental Setup

## B.1. Datasets

We use the `traffic`, `electricity`, and `solar_10_minutes` dataset as originally preprocessed by (Lai et al., 2018). We use the standard train–test splits as per GluonTS version $0.15.1$ and additionally remove the last $5 \cdot L_F$ forecast windows of each train set sequence for validation.

**Traffic.** The `traffic` dataset was originally sourced from the following domain: http://pems.dot.ca.gov/. It consists of hourly measurements from 862 traffic sensors, with each time series covering 17544 hours. The forecast length $L_F$ is 24. Although our experiments are mostly focused on verifying that our differentially private models can fit some non-trivial time series, traffic data may allow inference about personal movement profiles (Giannotti & Pedreschi, 2008).

**Electricity.** The `electricity` dataset was originally sourced from the following domain: https://archive.ics.uci.edu/ml/datasets/ElectricityLoadDiagrams20112014. It consists of hourly measurements from 321 electricity consumers, with each time series covering 26304 hours. The forecast length $L_F$ is 24. As before, the contribution of this work is mainly theoretical and the specific application domain is mostly irrelevant. The `electricity` dataset just happens to be commonly used for testing whether models can fit non-trivial time series.

**Solar.** The `solar_10_minutes` dataset was originally sourced from the following domain: http://www.nrel.gov/grid/solar-power-data.html. It consists of 6 measurements per hour from 137 photovoltaic power plants, with each time series covering 52560 10-minute intervals. The forecast length $L_F$ is 60.

## B.2. Models

### B.2.1. DEEP LEARNING MODELS

For all models, we use the standard hyper-parameters as per GluonTS version $0.15.1$. Per forecast step, we let the model parameterize a $t$-distribution for probabilistic forecasting. The only parameter we vary is the context length or the range of lagged values, as these affect our privacy analysis (see Appendix B.4.2). In particular, we use the following parameters per model:

**Simple Feed Forward.** We use two hidden layers with hidden dimension 64 and no batch normalization.

**DeepAR.** We use two hidden layers with hidden size 40, no dropout, no categorical embeddings, and activated target scaling.

**iTransformer.** We use latent dimension 32, 4 attention heads, 128 feed-forward neurons, no droput, ReLU activations, 2 encoder layers, and activated mean scaling.

**DLinear.** We use hidden dimension 20 and kernel size 25.

### B.2.2. TRADITIONAL BASELINES

**Seasonal Naïve.** We use season a season length of 24 (1 day) for `traffic` and `electricty`. We use a season length of $144 = 24 \cdot 6$ (1 day) for `solar_10_minutes`.

**AutoETS.** We use the standard implementation and parameters from `statsforecast` (Garza et al., 2022) version 1.7.8. We additionally set the season lengths to the same values as with the seasonal naïve predictor.

**AutoARIMA.** We attempted to also to use AutoARIMA with the standard implementation and parameters from `statsforecast` version 1.7.8. and with the above season lengths. However, the computation did not complete after 7 days on an AMD EPYC 7542 processor with 256GB RAM on any of the datasets. Other works on time series forecasting also report "d.n.f." for AutoARIMA, e.g., (Alexandrov et al., 2019; Shchur et al., 2023). Without specifying season lengths, AutoARIMA had a CRPS of $0.472$, $0.313$, and $7.193$ on the three datasets, i.e., it performed significantly worse than seasonal naïve prediction or AutoETS with appropriate season lengths.

## B.3. Metrics.

All experiments involving model training are repeated for 5 random seeds. We report means and standard deviation. As our training and validation loss, we use negative log likelihood.

For evaluating predictive performance, we use the Continuous Ranked Probability Score (CRPS), which is a proper scoring function (Gneiting & Raftery, 2007). Given cumulative distribution function $F$ and ground-truth $y \in \mathbb{R}$, it is defined as

$$\text{CRPS}(F^{-1}, y) = -1 \cdot \int_{-\infty}^{\infty} (F(x) - \mathbb{1}[x \geq y]\, \mathrm{d}x.$$

Like prior work, e.g., (Rasul et al., 2021; Chen et al., 2024; Kollovieh et al., 2024), and as implemented by default in GluonTS, we approximate it using quantile levels $\{0.1, 0.2, \ldots, 0.9\}$ ("Mean weighted quantile loss"). Note that this loss is $0$ for a Dirac-$\delta$ coinciding with ground-truth $y$, i.e., non-probabilistic models can also achieve a loss of $0$.

### B.4. Training

On all datasets, we train until reaching the prescribed privacy budget or some very liberally set maximum number of epochs that allows all models to train to convergence (details below). After training, we load the checkpoint with the lowest validation log-likelihood.

### B.4.1. STANDARD TRAINING PARAMETERS

We use ADAM with learning rate $10^{-3}$ and weight decay $10^{-8}$ for all models and datasets. `traffic`, `electricty`, and `solar_10_minutes` we train for 4000, 8000, and 16000 epochs, respectively.

### B.4.2. DP TRAINING PARAMETERS

We wrap the optimizer and models using the privacy engine (minus the accountant) from `opacus` (Yousefpour et al., 2021) (version 1.5.1), replacing all recurrent and self-attention layers with their DP-compatible PyTorch-only implementation. Following (Ponomareva et al., 2023), we iteratively decreased the gradient clipping norm $C$ until the validation loss without gradient noise increased, which led us to using $C = 10^{-4}$ on all datasets.

**Traffic.** We use batch size $\Lambda = 256$, noise multiplier $\sigma = 4.0$, and $L_C = 4 \cdot L_F$. For DeepAR, we use the following lag indices, which contribute to the context length $L_C$: $[1, 2, 3, 4, 5, 6, 7, 23, 24, 25, 47, 48, 49, 71, 72, 73, 95, 96, 97, 119, 120, 121, 143, 144, 145, 167, 168, 169]$.

**Electricity.** We use batch size $\Lambda = 128$, noise multiplier $\sigma = 2.0$, and $L_C = 1 \cdot L_F$. For DeepAR, we use the following lag indices, which contribute to the context length $L_C$: $[1, 2, 3, 4, 5, 6, 7, 23, 24, 25]$.

**Solar.** We use batch size $\Lambda = 128$, noise multiplier $\sigma = 4.0$, and $L_C = 4 \cdot L_F$. For DeepAR, we use the following lag indices, which contribute to the context length $L_C$: $[1, 2, 3, 4, 5, 6, 7, 23, 24, 25, 47, 48, 49, 71, 72, 73, 95, 96, 97, 119, 120, 121, 143, 144, 145, 167, 168, 169]$.

### B.4.3. PRIVACY ACCOUNTING PARAMETERS

We use privacy loss distribution accounting as implemented in the the Google `dp_accounting` library (Google Differential Privacy Team, 2024) (version 0.4.4) with a tail mass truncation constant of $10^{-15}$. We quantize privacy loss distributions using "connect-the-dots" (Doroshenko et al., 2022) with a value discretization interval of $10^{-3}$.

# C. Additional Experiments

## C.1. Trade-Offs in Structured Subsampling

### C.1.1. NUMBER OF SUBSEQUENCES IN BI-LEVEL SAMPLING

In the following, we repeat our experiment from Fig. 2, where we wanted to determine whether we should use small $\lambda$ (many top-level sequences, few bottom-level subsequences) or large $\lambda$ (many top-level sequences, few bottom-level subsequences) for some given batch size $\Lambda$.

From Theorem 4.4 and Theorem F.7 we know that our tight upper bound and our optimistic lower bounds only depend on $r = \frac{L_C + L_F}{L - L_F + 1}$ and $\rho = \lfloor \Lambda / \lambda \rfloor / N$. Thus (up to modulo division), the parameter space is fully characterized by $r$ and batch-to-dataset size ratio $\frac{\Lambda}{N}$.

In Figs. 5 to 7, we thus keep $N = 320$ and vary these two ratios between $0.5$ (little amplification) and $0.1$ (more amplification). In Fig. 5 we observe that $\lambda = 1$ offers better privacy than $\lambda > 1$ for all $\varepsilon \in [10^{-3}, 10^3]$ after 100 training steps. In Figs. 6 and 7, it already offers better privacy after 100 training steps. Again, this justifies our choice of $\lambda = 1$ in training models.

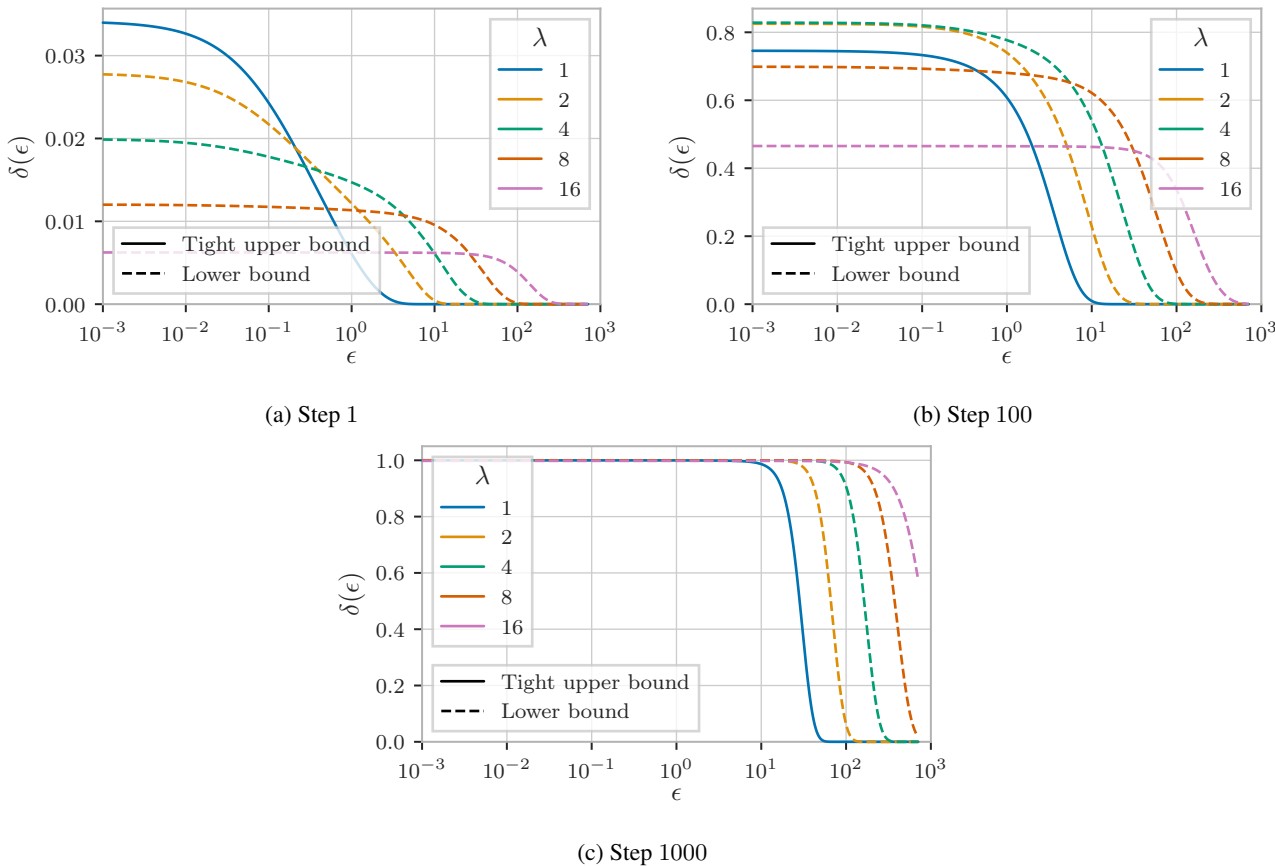

(a) Step 1

(b) Step 100

(c) Step 1000

Figure 5: Top-level WOR and bottom-level WR sampling under varying number of subsequences. Little bottom-level amplification ($r = 0.5$) and more top-level amplification ($\Lambda / N = 0.1$).

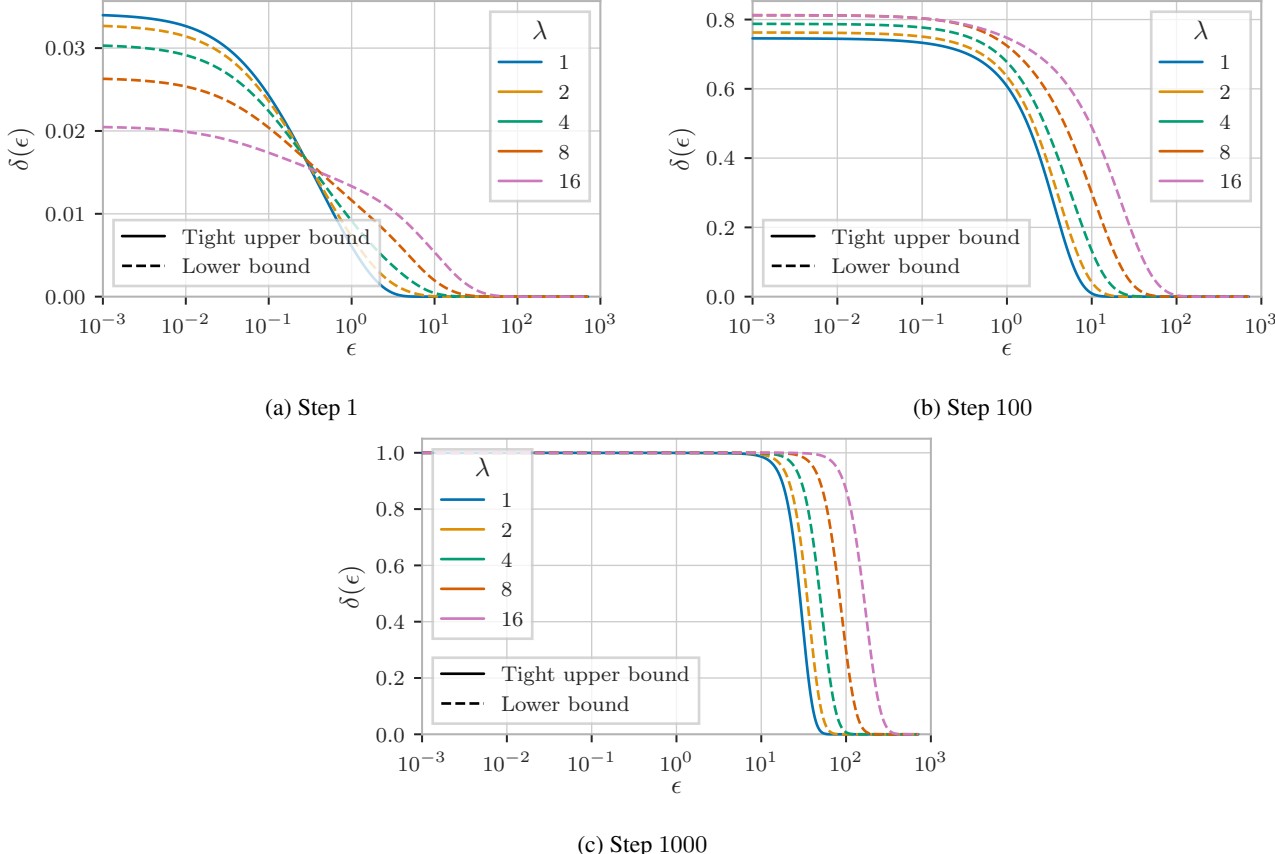

(a) Step 1

(b) Step 100

(c) Step 1000

Figure 6: Top-level WOR and bottom-level WR sampling under varying number of subsequences. More bottom-level amplification ($r = 0.1$) and less top-level amplification ($\Lambda / N = 0.5$).

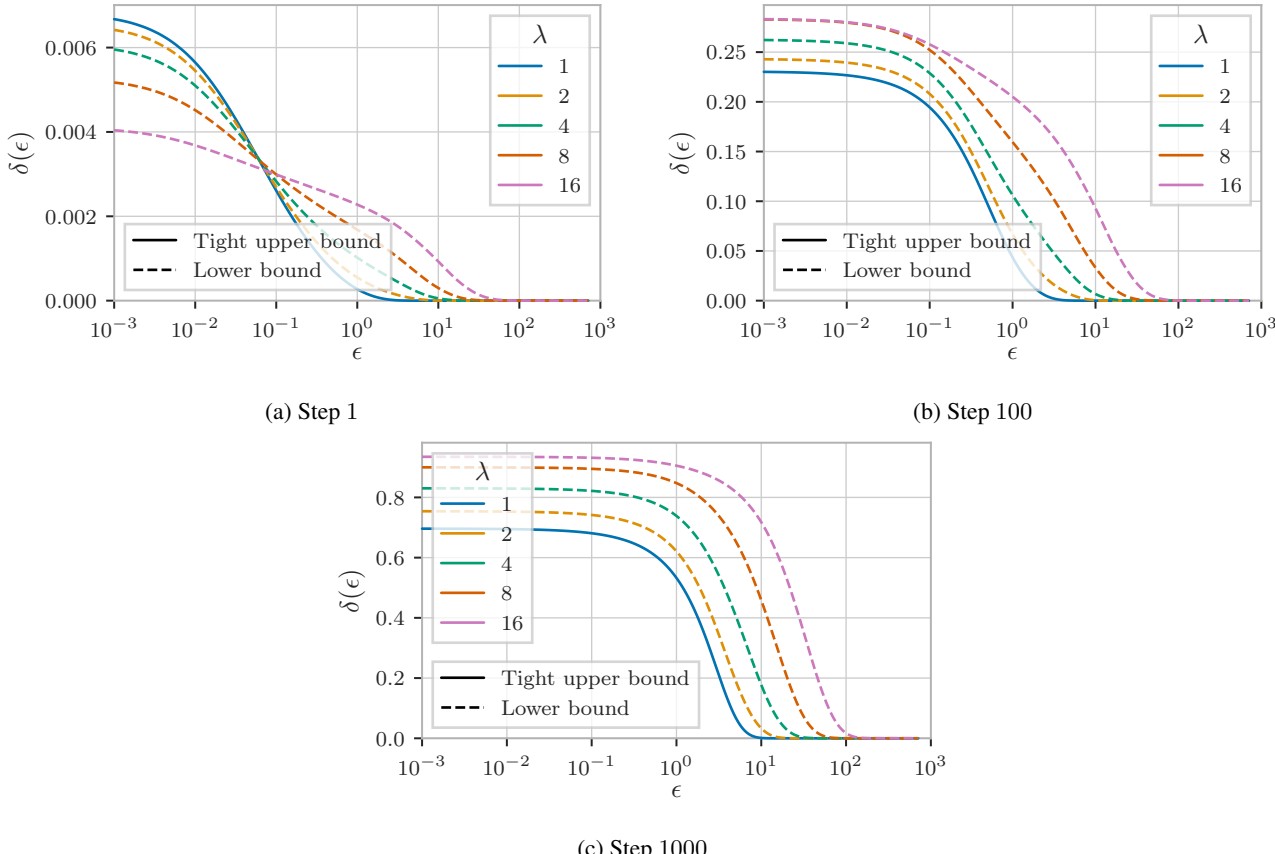

(a) Step 1

(b) Step 100

(c) Step 1000

Figure 7: Top-level WOR and bottom-level WR sampling under varying number of subsequences. Both significant bottom-level amplification ($r = 0.1$) and top-level amplification ($\Lambda / N = 0.1$).

### C.1.2. STEP- VS EPOCH-LEVEL ACCOUNTING

In this section, we repeat our experiment from Fig. 3 to demonstrate the benefit of top-level sampling sequences (Theorem 4.4) instead of deterministically iterating over them (Theorem 4.2), even though we risk privacy leakage at every training step.

Like in Appendix C.1.1, we observe that the deterministic-top-level guarantee is only dependent on $r = \frac{L_C + L_F}{L - L_F + 1}$, and that the WOR-top-level guarantee is only dependent on $r$ and $\rho = \lfloor \Lambda / \lambda \rfloor / N$.

As before, we thus keep dataset size $N = 320$ and vary $r$ and batch-to-dataset size ratio $\frac{\Lambda}{N}$ between 0.5 and 0.1 (see Fig. 8). In all cases, sampling without replacement offers stronger privacy after 1 step, 1 epoch, and 10 epochs.

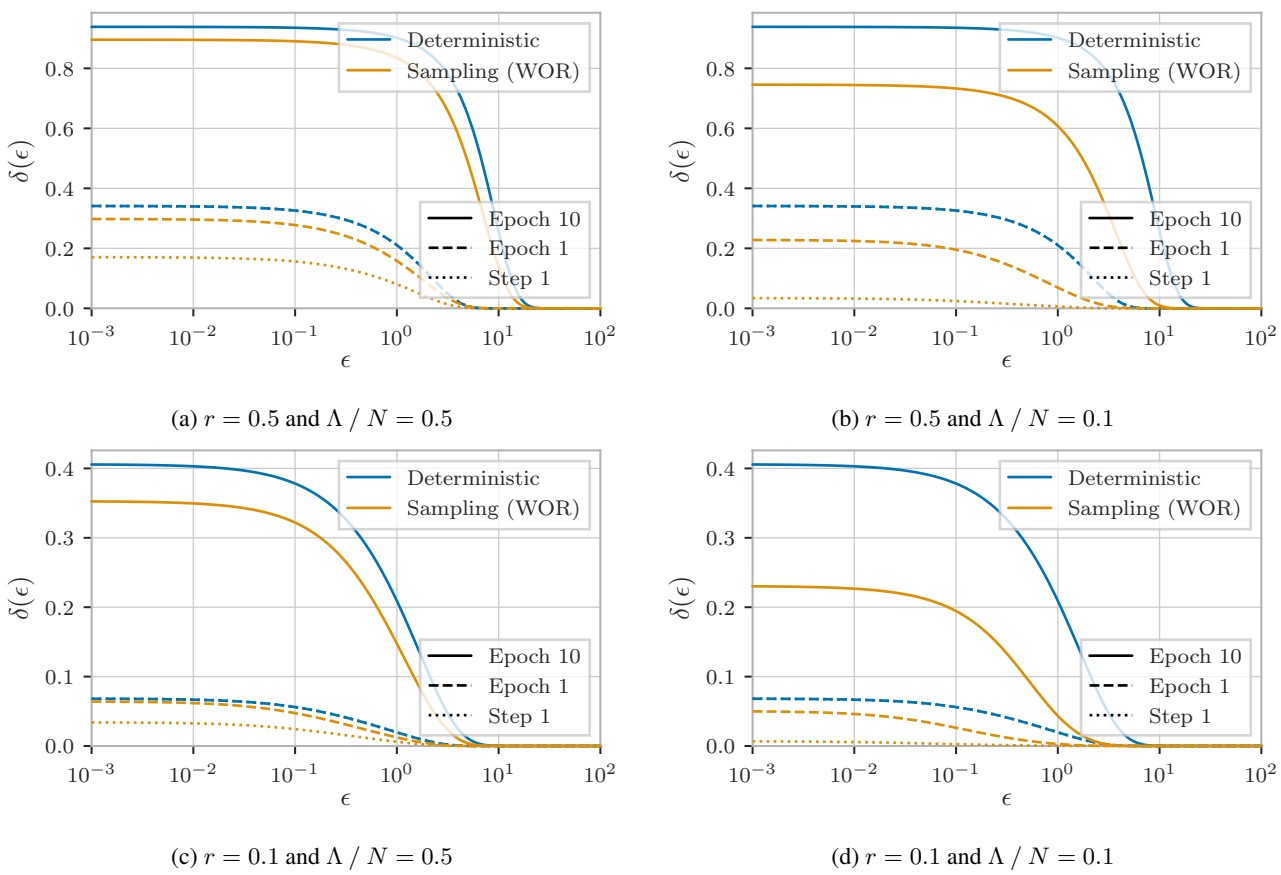

(a) $r = 0.5$ and $\Lambda / N = 0.5$

(b) $r = 0.5$ and $\Lambda / N = 0.1$

(c) $r = 0.1$ and $\Lambda / N = 0.5$

(d) $r = 0.1$ and $\Lambda / N = 0.1$

Figure 8: Top-level deterministic iteration (Theorem 4.2) vs top-level WOR sampling (Theorem 4.4) for $\lambda = 1$. We vary $r = (L_C + L_F) / (L - L_F + 1)$ and $\Lambda / N$, with smaller values corresponding to more bottom- and top-level amplification, respectively.

### C.1.3. AMPLIFICATION BY LABEL PERTURBATION

In this section, we repeat our experiment from Fig. 4, where we consider top-level WOR and bottom-level WOR sampling, and vary label noise standard deviation $\sigma_F \in \mathbb{R}_+$ to illustrate how test-time data augmentations (i.e. continuous-valued subsampling) can amplify privacy (Theorem G.1) in self-supervised training of sequence models.

Just like in Appendix C.1, we observe that the privacy guarantee, for fixed forecast-to-context ratio $\frac{L_F}{L_C+L_F}$, is only dependent on $r = \frac{L_C+L_F}{L-L_F+1}$, and $\rho = \lfloor \Lambda / \lambda \rfloor / N$.

As before, we thus keep dataset size $N = 320$ and vary $r$ and batch-to-dataset size ratio $\frac{\Lambda}{N}$ between 0.5 and 0.1 (see Fig. 9). In all cases, letting $\sigma_F \to \infty$ is equivalent to setting forecast length $L_F$ to 0.

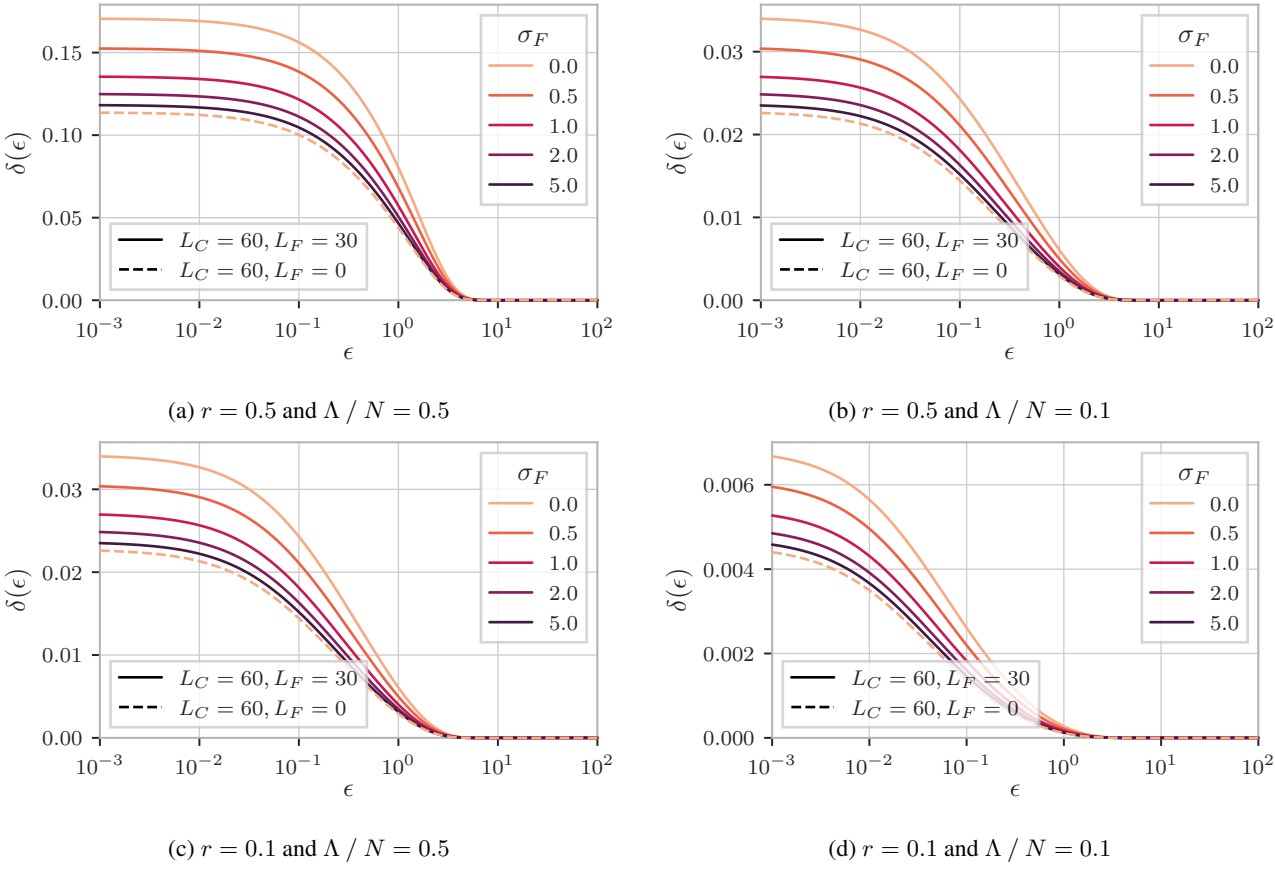

(a) $r = 0.5$ and $\Lambda / N = 0.5$

(b) $r = 0.5$ and $\Lambda / N = 0.1$

(c) $r = 0.1$ and $\Lambda / N = 0.5$

(d) $r = 0.1$ and $\Lambda / N = 0.1$

Figure 9: Varying label noise $\sigma_F$ for top-level WOR and bottom-level WR (Theorem G.1) with $\sigma_C = 0, \lambda = 1$. We additionally vary $r = (L_C + L_F) / (L - L_F + 1)$ and $\Lambda / N$, with smaller values corresponding to more bottom- and top-level amplification, respectively.

### C.1.4. EPOCH PRIVACY VS LENGTH

This experiment differs from the previous ones in that we exclusively focus on top-level deterministic iteration and bottom-level sampling with replacement (see Theorem 4.2). Recall from our discussion in Section 4.1 that $\lambda = 1$ minimizes the privacy of each epoch, but forces us to perform $k$ times as many epochs for the same number of training steps as $\lambda = k$. Thus, there are more chances for privacy leakage and we need to self-compose the privacy profile for $\lambda = 1$ exactly $k$ times more often.

In the following, we demonstrate that composing many epochs that are more private (i.e., $\lambda = 1$) can nevertheless be beneficial. To this end, we fix dataset size $N = 320$ and (to eliminate one redundant degree of freedom) batch size $\Lambda = 320$. With this parameterization, $\lambda = k$ means that we perform $k$ training steps in a single epoch. For our comparison, we thus self-compose our epoch-level mechanism 16 times for $\lambda = 1$, 8 times for $\lambda = 2$, 4 times for $\lambda = 4$ etc. to determine privacy for the same number of training steps. We additionally vary the amount of bottom-level amplification by varying $r = \frac{L_C + L_F}{L - L_F + 1}$ between 0.5 and 0.1.

As can be seen in Fig. 10, the number of sequences $\lambda = 1$ offers smaller $\delta(\varepsilon)$ at every training step that coincides with an epoch of $\lambda > 1$.

To conclude, choosing number of subsequences $\lambda = 1$ remains the preferred option (see also Theorem 4.1) even under composition.

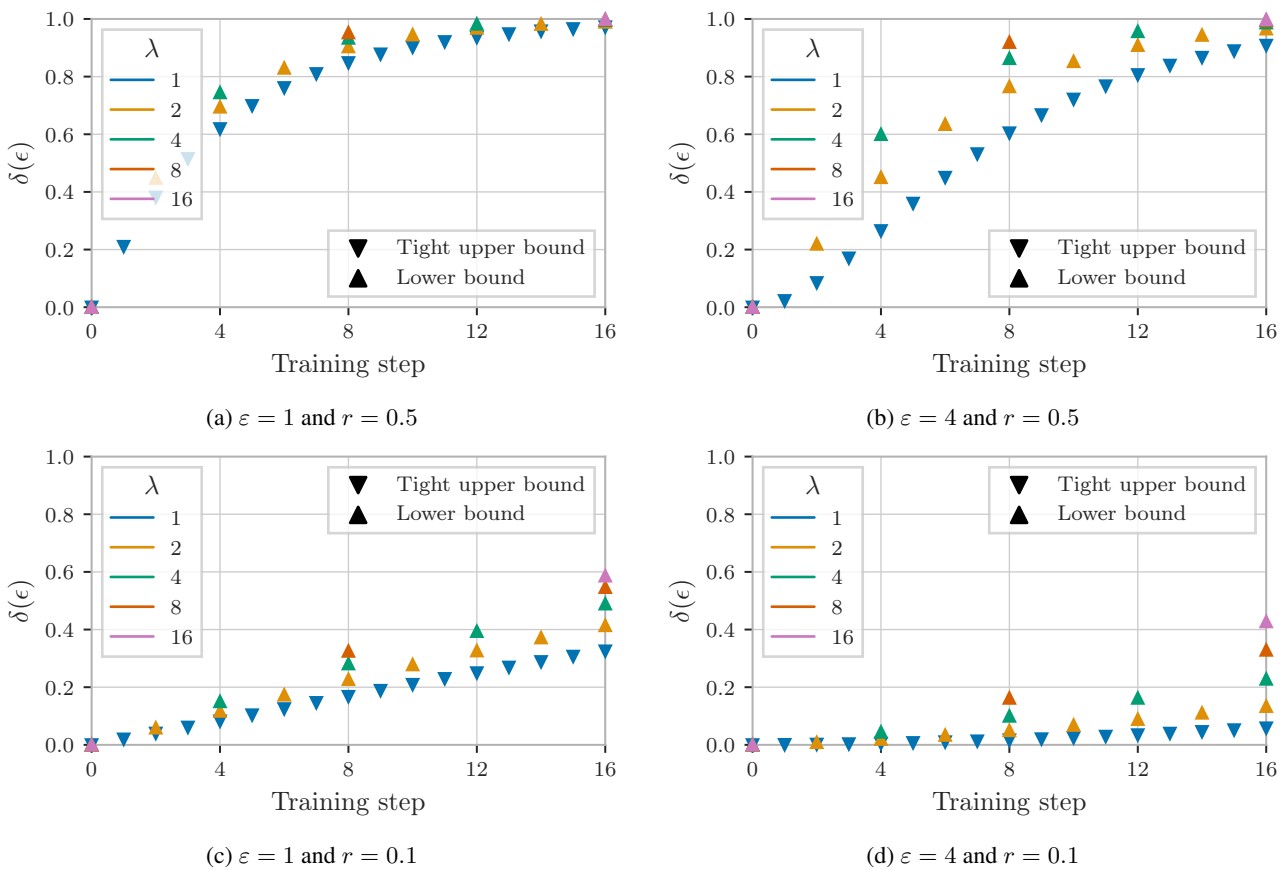

Figure 10: Privacy parameter $\delta(\varepsilon)$ for $\varepsilon \in \{1, 4\}$ over the course of 16 training steps when using top-level deterministic iteration and bottom-level sampling with replacement (Theorem 4.2). We additionally vary $r$ between 0.5 (less bottom-level amplification) and 0.1 (more bottom-level amplification).

## C.2. Application to Probabilistic Forecasting

### C.2.1. EVENT-LEVEL PRIVACY

Tables 2 to 4 show average CRPS after 1-event-level private training on our three standard benchmark datasets. Since $\delta^{-1}$ is approximately equal or greater than the dataset sizes, $\varepsilon \leq 1$ indicates strong privacy guarantees, whereas $2 \leq \varepsilon \leq 8$ would be more commonly expected values in private training of machine learning models (Ponomareva et al., 2023).

For a full description of all hyper parameters, see Appendix B.

In all cases, at least one model outperforms the traditional baselines without noise for all considered $\varepsilon$.

Table 2: Average CRPS on `traffic` for 1-event-level privacy and $\delta = 10^{-7}$. Seasonal, AutoETS, and models with $\varepsilon = \infty$ are without noise. Bold font indicates the best predictor per $\varepsilon$.

| MODEL | $\varepsilon = 0.25$ | $\varepsilon = 0.5$ | $\varepsilon = 1$ | $\varepsilon = 2$ | $\varepsilon = 4$ | $\varepsilon = 8$ | $\varepsilon = \infty$ |
|---|---|---|---|---|---|---|---|
| SIMPLEFF | $0.252$ ±0.007 | $0.207$ ±0.002 | $0.195$ ±0.003 | $0.193$ ±0.003 | $0.194$ ±0.002 | $0.193$ ±0.003 | $0.136$ ±0.001 |
| DEEPAR | $0.262$ ±0.015 | $\mathbf{0.157}$ ±0.002 | $\mathbf{0.145}$ ±0.001 | $\mathbf{0.142}$ ±0.001 | $\mathbf{0.141}$ ±0.002 | $\mathbf{0.141}$ ±0.002 | $\mathbf{0.124}$ ±0.0.001 |
| iTRANSF. | $0.260$ ±0.003 | $0.211$ ±0.004 | $0.193$ ±0.003 | $0.188$ ±0.004 | $0.188$ ±0.004 | $0.188$ ±0.004 | $0.135$ ±0.001 |
| DLINEAR | $\mathbf{0.236}$ ±0.006 | $0.204$ ±0.004 | $0.192$ ±0.001 | $0.188$ ±0.003 | $0.188$ ±0.003 | $0.188$ ±0.003 | $0.140$ ±0.000 |
| SEASONAL | $0.251$ | $0.251$ | $0.251$ | $0.251$ | $0.251$ | $0.251$ | $0.251$ |
| AUTOETS | $0.407$ | $0.407$ | $0.407$ | $0.407$ | $0.407$ | $0.407$ | $0.407$ |

Table 3: Average CRPS on `electricity` for 1-event-level privacy and $\delta = 10^{-7}$. Seasonal, AutoETS, and models with $\varepsilon = \infty$ are without noise. Bold font indicates the best predictor per $\varepsilon$.

| MODEL | $\varepsilon = 0.25$ | $\varepsilon = 0.5$ | $\varepsilon = 1$ | $\varepsilon = 2$ | $\varepsilon = 4$ | $\varepsilon = 8$ | $\varepsilon = \infty$ |
|---|---|---|---|---|---|---|---|
| SIMPLEFF | $0.087$ ±0.002 | $0.072$ ±0.001 | $0.065$ ±0.001 | $0.065$ ±0.001 | $0.065$ ±0.001 | $0.065$ ±0.002 | $\mathbf{0.058}$ ±0.001 |
| DEEPAR | $0.121$ ±0.014 | $0.071$ ±0.004 | $0.070$ ±0.004 | $0.068$ ±0.005 | $0.067$ ±0.004 | $0.068$ ±0.005 | $\mathbf{0.058}$ ±0.002 |
| iTRANSF. | $0.107$ ±0.004 | $0.081$ ±0.005 | $0.075$ ±0.002 | $0.074$ ±0.002 | $0.074$ ±0.002 | $0.074$ ±0.002 | $\mathbf{0.058}$ ±0.001 |
| DLINEAR | $\mathbf{0.076}$ ±0.003 | $\mathbf{0.064}$ ±0.000 | $\mathbf{0.061}$ ±0.001 | $\mathbf{0.061}$ ±0.001 | $\mathbf{0.061}$ ±0.001 | $\mathbf{0.061}$ ±0.001 | $0.059$ ±0.000 |
| SEASONAL | $0.070$ | $0.070$ | $0.070$ | $0.070$ | $0.070$ | $0.070$ | $0.070$ |
| AUTOETS | $0.064$ | $0.064$ | $0.064$ | $0.064$ | $0.064$ | $0.064$ | $0.064$ |

Table 4: Average CRPS on `solar_10_minutes` for 1-event-level privacy and $\delta = 10^{-7}$. Seasonal, AutoETS, and models with $\varepsilon = \infty$ are without noise. Bold font indicates the best predictor per $\varepsilon$.

| MODEL | $\varepsilon = 0.25$ | $\varepsilon = 0.5$ | $\varepsilon = 1$ | $\varepsilon = 2$ | $\varepsilon = 4$ | $\varepsilon = 8$ | $\varepsilon = \infty$ |
|---|---|---|---|---|---|---|---|
| SIMPLEFF | $1.108$ ±0.035 | $1.114$ ±0.043 | $1.114$ ±0.044 | $1.114$ ±0.040 | $1.118$ ±0.038 | $1.113$ ±0.028 | $0.766$ ±0.766 |
| DEEPAR | $\mathbf{0.910}$ ±0.017 | $\mathbf{0.820}$ ±0.030 | $\mathbf{0.803}$ ±0.023 | $\mathbf{0.792}$ ±0.016 | $\mathbf{0.787}$ ±0.023 | $\mathbf{0.787}$ ±0.023 | $\mathbf{0.654}$ ±0.654 |
| iTRANSF. | $1.134$ ±0.057 | $0.977$ ±0.065 | $0.956$ ±0.062 | $0.975$ ±0.066 | $0.974$ ±0.065 | $0.974$ ±0.065 | $0.804$ ±0.804 |
| DLINEAR | $1.434$ ±0.073 | $1.287$ ±0.222 | $1.152$ ±0.201 | $1.110$ ±0.145 | $1.048$ ±0.143 | $1.051$ ±0.140 | $0.860$ ±0.860 |
| SEASONAL | $1.120$ | $1.120$ | $1.120$ | $1.120$ | $1.120$ | $1.120$ | $1.120$ |
| AUTOETS | $6.494$ | $6.494$ | $6.494$ | $6.494$ | $6.494$ | $6.494$ | $6.494$ |

### C.2.2. W-EVENT AND W-USER-LEVEL PRIVACY

As discussed in Appendix I, we can (for sufficiently long sequences) generalize our bounds on the mechanism's privacy profile from 1-event- to $w$-event- or $w$-user-level privacy by replacing any occurrence of $L_C + L_F$ with $L_C + L_F + w - 1$ or $w \cdot (L_C + L_F)$. Since $w'$-event- and $w$-user-level privacy lead to identical results for some sufficiently large $w'$, it is sufficient to experiment with $w$-user-level privacy.

For a full description of all hyper parameters, see Appendix B.

Tables 5 to 7 show average CRPS after $w$-user-level private training on our three standard benchmark datasets.

Except for traffic and $w = 8$ (which, by the above argument and our choice of $L_C$ and $L_F$, is equivalent to requiring privacy for an event spanning multiple days in our hourly datasets), at least one model outperforms the traditional baselines without noise for all considered $w$.

Table 5: Average CRPS on traffic for $w$-user-level privacy and $\varepsilon = 4$, $\delta = 10^{-7}$. Seasonal and AutoETS are without noise. Bold font indicates the best predictor per $w$.

| MODEL | $w = 1$ | $w = 2$ | $w = 4$ | $w = 8$ |
|---|---|---|---|---|
| SIMPLEFF | $0.193 \pm 0.003$ | $0.194 \pm 0.003$ | $0.194 \pm 0.003$ | $0.211 \pm 0.001$ |
| DEEPAR | $\mathbf{0.142} \pm 0.003$ | $\mathbf{0.143} \pm 0.001$ | $\mathbf{0.145} \pm 0.001$ | $\mathbf{0.166} \pm 0.004$ |
| ITRANSF. | $0.188 \pm 0.004$ | $0.188 \pm 0.004$ | $0.193 \pm 0.002$ | $0.217 \pm 0.005$ |
| DLINEAR | $0.189 \pm 0.002$ | $0.189 \pm 0.003$ | $0.192 \pm 0.001$ | $0.208 \pm 0.004$ |
| SEASONAL | $0.251$ | $0.251$ | $0.251$ | $0.251$ |
| AUTOETS | $0.407$ | $0.407$ | $0.407$ | $0.407$ |

Table 6: Average CRPS on electricity for $w$-user-level privacy and $\varepsilon = 4$, $\delta = 10^{-7}$. Seasonal and AutoETS are without noise. Bold font indicates the best predictor per $w$.

| MODEL | $w = 1$ | $w = 2$ | $w = 4$ | $w = 8$ |
|---|---|---|---|---|
| SIMPLEFF | $0.064 \pm 0.001$ | $0.065 \pm 0.001$ | $0.064 \pm 0.001$ | $0.074 \pm 0.001$ |
| DEEPAR | $0.068 \pm 0.005$ | $0.069 \pm 0.005$ | $0.068 \pm 0.002$ | $0.073 \pm 0.003$ |
| ITRANSF. | $0.075 \pm 0.003$ | $0.074 \pm 0.002$ | $0.075 \pm 0.003$ | $0.083 \pm 0.004$ |
| DLINEAR | $\mathbf{0.061} \pm 0.001$ | $\mathbf{0.061} \pm 0.001$ | $\mathbf{0.061} \pm 0.001$ | $0.066 \pm 0.001$ |
| SEASONAL | $0.070$ | $0.070$ | $0.070$ | $0.070$ |
| AUTOETS | $0.064$ | $0.064$ | $0.064$ | $\mathbf{0.064}$ |

Table 7: Average CRPS on solar_10_minutes for $w$-user-level privacy and $\varepsilon = 4$, $\delta = 10^{-7}$. Seasonal and AutoETS are without noise. Bold font indicates the best predictor per $w$.

| MODEL | $w = 1$ | $w = 2$ | $w = 4$ | $w = 8$ |
|---|---|---|---|---|
| SIMPLEFF | $1.113 \pm 0.033$ | $1.116 \pm 0.038$ | $1.110 \pm 0.042$ | $1.107 \pm 0.034$ |
| DEEPAR | $\mathbf{0.783} \pm 0.019$ | $\mathbf{0.791} \pm 0.018$ | $\mathbf{0.807} \pm 0.024$ | $\mathbf{0.823} \pm 0.028$ |
| ITRANSF. | $0.951 \pm 0.061$ | $0.950 \pm 0.062$ | $0.956 \pm 0.062$ | $0.980 \pm 0.060$ |
| DLINEAR | $1.072 \pm 0.164$ | $1.122 \pm 0.158$ | $1.200 \pm 0.220$ | $1.298 \pm 0.205$ |
| SEASONAL | $1.120$ | $1.120$ | $1.120$ | $1.120$ |
| AUTOETS | $6.494$ | $6.494$ | $6.494$ | $6.494$ |

### C.2.3. AMPLIFICATION BY LABEL PERTURBATION

Finally, we can demonstrate the potential benefit of using online data augmentations to amplify privacy in training forecasting models (Theorem 4.5). Consider $(w, v)$-event-level or $(w, v)$-user-level privacy with sufficiently small $v$. If $v$ is sufficiently small compared to the scale of the dataset, i.e., each individual only makes a small contribution to the overall value of a time series at each time step, then we can introduce substantial context and label noise $\sigma_C$ and $\sigma_F$ to amplify privacy without significantly affecting utility. Simultaneously, we still benefit from amplification through top- and bottom-level subsampling, i.e., do not need to add as much noise as would be required for directly making the entire input dataset privacy.

For a full description of all hyper parameters, see Appendix B.

Tables 8 to 10 show average CRPS after $(1, v)$-event-level private training with $\varepsilon = 0.5$, $\delta = 10^{-7}$, i.e., strong privacy guarantees. Note that we use different $v$ per dataset, as they have different scale.

We observe that, for all models and all datasets, the best score is attained with label noise scale $\sigma_F = 2$ or $\sigma_F = 2$. This confirms that there a scenarios in which our novel amplification-by-augmentation guarantees can help improve the privacy–utility trade-off of forecasting models.

Table 8: Average CRPS on `traffic` for $(1, 0.001)$-user-level privacy and $\varepsilon = 0.5$, $\delta = 10^{-7}$. Seasonal and AutoETS are without noise. Bold font indicates the best label noise scale $\sigma_F$ per model, i.e., per row.

| MODEL | $\sigma_F = 0$ | $\sigma_F = 1$ | $\sigma_F = 2$ | $\sigma_F = 5$ |
|---|---|---|---|---|
| SIMPLEFF | $0.207 \pm 0.002$ | $0.205 \pm 0.002$ | $0.205 \pm 0.002$ | $\mathbf{0.205} \pm 0.001$ |
| DEEPAR | $0.156 \pm 0.003$ | $0.156 \pm 0.003$ | $0.156 \pm 0.003$ | $\mathbf{0.154} \pm 0.002$ |
| ITRANSF. | $0.211 \pm 0.004$ | $0.208 \pm 0.004$ | $0.205 \pm 0.003$ | $\mathbf{0.204} \pm 0.003$ |
| DLINEAR | $0.203 \pm 0.003$ | $0.202 \pm 0.003$ | $\mathbf{0.202} \pm 0.003$ | $0.203 \pm 0.003$ |
| SEASONAL | $0.251$ | $0.251$ | $0.251$ | $0.251$ |
| AUTOETS | $0.407$ | $0.407$ | $0.407$ | $0.407$ |

Table 9: Average CRPS on `electricity` for $(1, 0.1)$-user-level privacy and $\varepsilon = 0.5$, $\delta = 10^{-7}$. Seasonal and AutoETS are without noise. Bold font indicates the best label noise scale $\sigma_F$ per model, i.e., per row.

| MODEL | $\sigma_F = 0$ | $\sigma_F = 1$ | $\sigma_F = 2$ | $\sigma_F = 5$ |
|---|---|---|---|---|
| SIMPLEFF | $0.072 \pm 0.001$ | $0.069 \pm 0.001$ | $0.068 \pm 0.002$ | $\mathbf{0.067} \pm 0.002$ |
| DEEPAR | $0.071 \pm 0.006$ | $0.074 \pm 0.005$ | $0.069 \pm 0.005$ | $\mathbf{0.067} \pm 0.005$ |
| ITRANSF. | $0.081 \pm 0.005$ | $0.080 \pm 0.004$ | $0.080 \pm 0.005$ | $\mathbf{0.080} \pm 0.004$ |
| DLINEAR | $0.064 \pm 0.000$ | $0.062 \pm 0.000$ | $0.061 \pm 0.001$ | $\mathbf{0.061} \pm 0.001$ |
| SEASONAL | $0.070$ | $0.070$ | $0.070$ | $0.070$ |
| AUTOETS | $0.064$ | $0.064$ | $0.064$ | $0.064$ |

Table 10: Average CRPS on `solar_10_minutes` for $(1, 0.01)$-user-level privacy and $\varepsilon = 0.5$, $\delta = 10^{-7}$. Seasonal and AutoETS are without noise. Bold font indicates the best label noise scale $\sigma_F$ per model, i.e., per row.

| MODEL | $\sigma_F = 0$ | $\sigma_F = 1$ | $\sigma_F = 2$ | $\sigma_F = 5$ |
|---|---|---|---|---|
| SIMPLEFF | $1.117 \pm 0.034$ | $1.126 \pm 0.046$ | $\mathbf{1.117} \pm 0.042$ | $1.119 \pm 0.041$ |
| DEEPAR | $0.820 \pm 0.030$ | $0.818 \pm 0.028$ | $0.814 \pm 0.027$ | $\mathbf{0.813} \pm 0.026$ |
| ITRANSF. | $0.976 \pm 0.064$ | $0.970 \pm 0.062$ | $0.960 \pm 0.052$ | $\mathbf{0.959} \pm 0.056$ |
| DLINEAR | $1.282 \pm 0.211$ | $1.245 \pm 0.232$ | $\mathbf{1.238} \pm 0.240$ | $1.246 \pm 0.227$ |
| SEASONAL | $1.120$ | $1.120$ | $1.120$ | $1.120$ |
| AUTOETS | $6.494$ | $6.494$ | $6.494$ | $6.494$ |

C.2.4. EFFECT OF SUBSAMPLING PARAMETERS ON NON-PRIVATE UTILITY

In all of our experiments on probabilistic forecasting, we used top-level sampling without replacement, bottom-level sampling with replacement, and $\lambda = 1$ subsequence per sequence. This choice of subsampling scheme was based on our observation that (1) top-level sampling without replacement yields stronger privacy than deterministic iteration despite requiring multiple compositions per epoch (see Figs. 2 and 3), and that (2) $\lambda = 1$ yields stronger privacy under composition.

However, it could be that other parameterizations offer much better utility that could potentially outweigh their weaker privacy. Alternatively, it could be that the models only achieve a good privacy–utility trade-off because our specific choice of parametermization leads to a massive improvement in utility compared to other parameterizations.

To provide evidence against these two hypotheses, we evaluate the utility of the four different model architectures under different subsampling schemes when performing standard, non-DP training on the `traffic` dataset. Non-DP training is equivalent to using clipping constant $C \to \infty$, noise scale $\sigma \to 0$, and privacy budget $\varepsilon \to \infty$. All parameters that are not being varied are identical to our previous experiments (see also Appendix B).

In Fig. 11, we fix $\lambda = 1$ as in our other experiments, and vary both the batch size $\Lambda \in \{16, 32, 64, 128\}$ and the top-level sampling scheme. In addition to deterministic iteration and sampling without replacement, we also evaluate iteration over a random permutation of the sequences. As can be seen, neither the batch size nor the top-level scheme has any significant effect on utility. In Fig. 12, we fix batch size $\Lambda = 256$ as in our other experiments, and vary the number of subsequences $\lambda \in \{1, 2, 4, 8, 16\}$ and the top-level sampling scheme. Again, there is no significant effect on model utility.

It can be concluded that the good privacy–utility trade-off attained in private training is not simply due to a massive increase in utility caused by our chosen subsampling parameters.

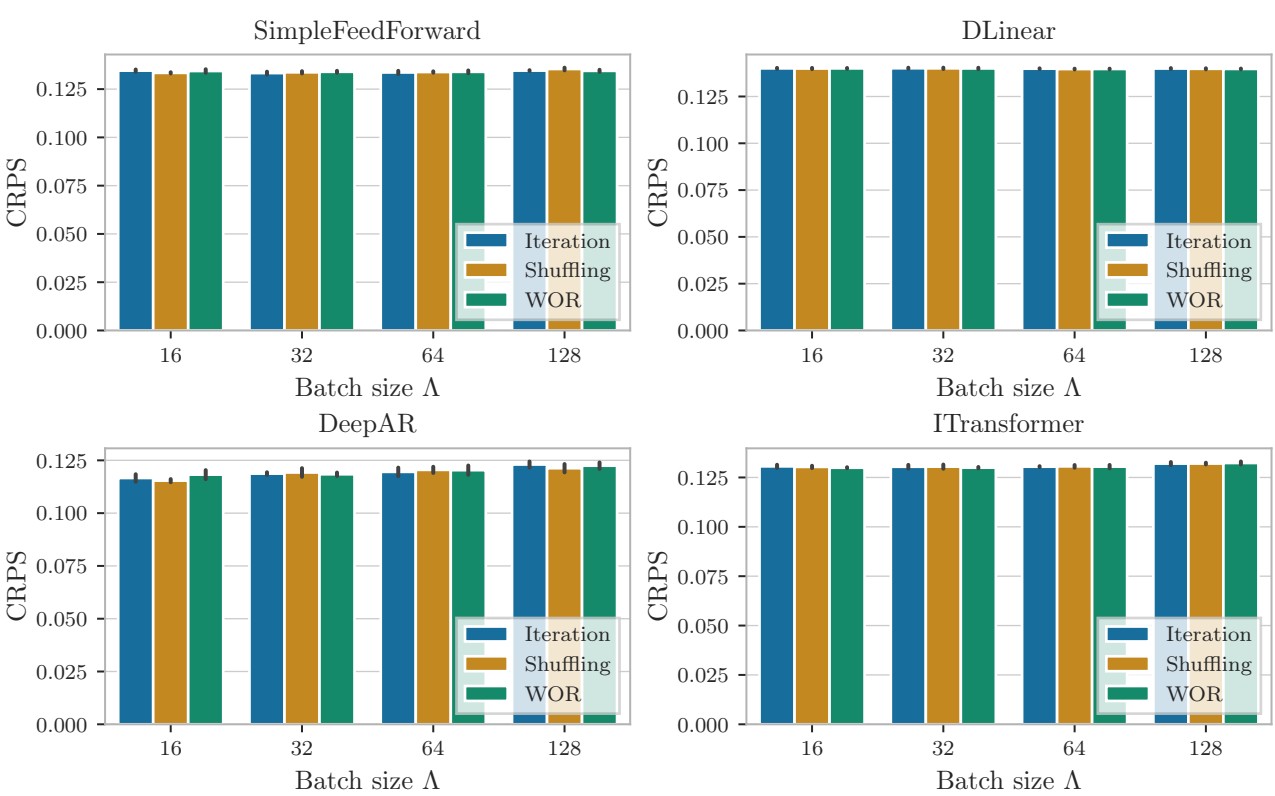

Figure 11: Average CRPS on `traffic` for non-DP training with $\lambda = 1$ under varying top-level scheme and batch size $\Lambda$.

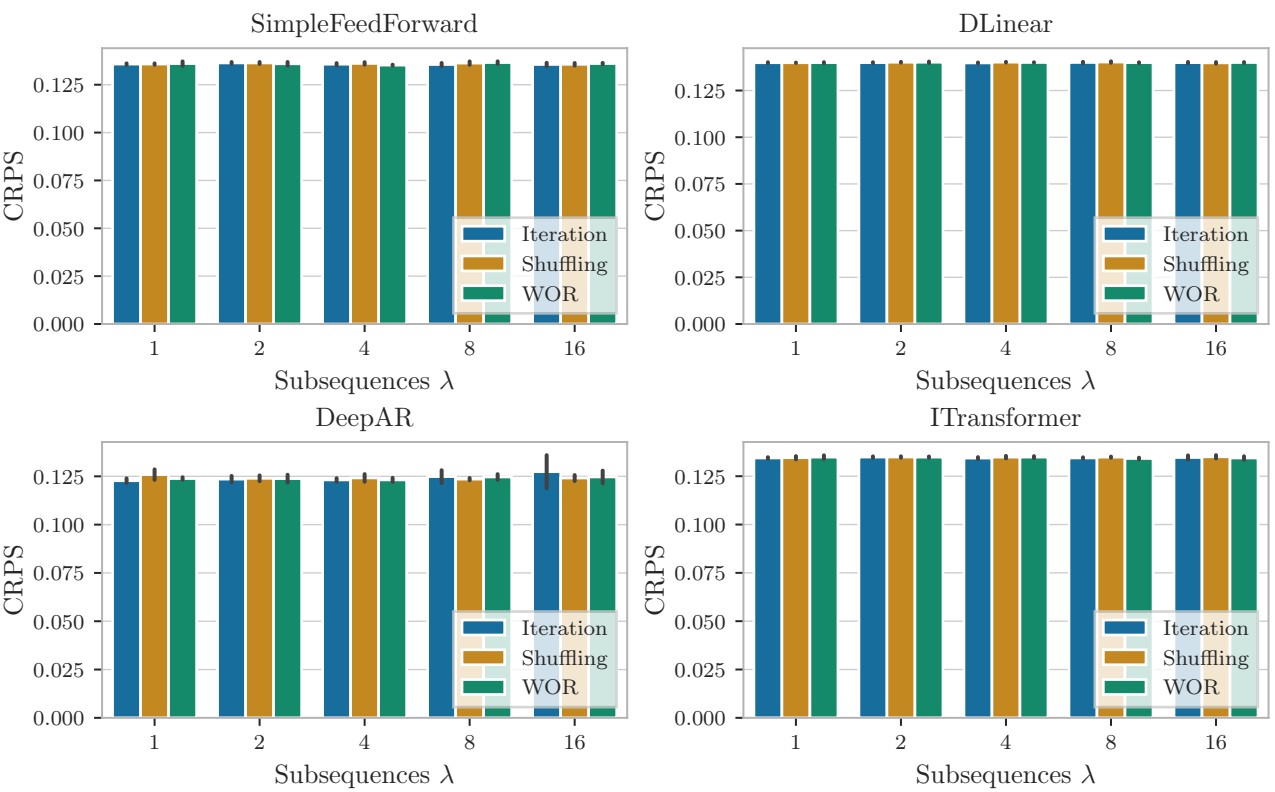

Figure 12: Average CRPS on `traffic` for non-DP training with batch size $\Lambda = 256$ under varying top-level scheme and number of subsequences $\lambda$.

## C.3. Comparison to Standard DP-SGD

Our analysis focuses on the batching procedure in Fig. 1, in which one first selects a set of sequences ("top-level sampling") and then selects subsequences from these specific sequences ("bottom-level sampling") to generate a batch of size $\Lambda$. However, one could in principle also accumulate all possible subsequences of all sequences from a timeseries dataset $x \subseteq \mathbb{R}^L$ with $|x| = N$ into one large dataset $\tilde{x} \subseteq \mathbb{R}^{L_C+L_F}$ of size $|x| = N_{\text{total}} = N \cdot (L - L_F + 1)$. Once this dataset is constructed, one could sample a batches of size $\Lambda$ using a standard subsampling distribution (e.g., sampling uniformly at random). Such a batching approach is, for example, implemented in the Darts forecasting library (Herzen et al., 2022). After this dataset is constructed, one could directly train models via existing implementations of DP-SGD.

However, one must not directly apply the privacy guarantees provided by existing implementations because they assume individual-level modifications of the dataset: For substitution relation $\tilde{x} \simeq_\Delta \tilde{x}'$, which indicates that there is a pair of records $a \in \tilde{x}, a' \notin \tilde{x}$ such that $\tilde{x}' = \tilde{x} \setminus \{a\} \cup \{a'\}$, the privacy profile of DP-SGD when sampling a batch of size $\Lambda$ uniformly without replacement can be tightly characterized as follows (Zhu et al., 2022):

**Proposition C.1.** *Consider datasets of size $N_{\text{total}}$ and subsampling without replacement with batch size $\Lambda \leq N_{\text{total}}$. Let $r = \frac{\Lambda}{N_{\text{total}}}$ be the probability of sampling a subsequence containing any specific element. Define $P(1) = MoG(\boldsymbol{\mu}, \boldsymbol{p}, \sigma)$ with means $\boldsymbol{\mu} = \begin{bmatrix} 0 & 2 \end{bmatrix}^T$ and weights $\boldsymbol{p} = \begin{bmatrix} 1-r & r \end{bmatrix}^T$. Further, define per-step privacy profile $H(\alpha) = \sup_{\tilde{x} \simeq_\Delta \tilde{x}'} H_\alpha(M_x \| M_{x'})$. Then,*

$$H(\alpha) = \begin{cases} H_\alpha(P(1) \| \mathcal{N}(0, \sigma)) & \text{if } \alpha \geq 1, \\ H_\alpha(\mathcal{N}(0, \sigma) \| P(1)) & \text{if } 0 \leq \alpha < 1. \end{cases}$$

But for event-level neighboring time series datasets $x \overset{1}{\simeq}_E x'$, there are up to $L_C + L_F$ subsequences that contain the sensitive element. Thus, the constructed datasets $\tilde{x}, \tilde{x}'$ would differ in up to $L_C + L_F$ records. More formally, they would be neighboring under $L_C + L_F$-substitution relation $\tilde{x} \simeq_{L_C+L_F,\Delta} \tilde{x}'$ which indicates that there is a subset $g \subseteq \tilde{x}$ with $|g| = L_C + L_F$ and a subset $g' \cap \tilde{x}' = \emptyset$ with $|g'| = L_C + L_F$ such that $\tilde{x}' = \tilde{x} \setminus g \cup g'$. Using an analogous construction to Theorem 4.3, one can easily show that the privacy profile is optimistically lower-bounded as follows:

**Proposition C.2.** *Consider datasets of size $N_{\text{total}}$ and subsampling without replacement with batch size $\Lambda \leq N_{\text{total}}$. Define $\underline{P}(\lambda) = MoG(\boldsymbol{\mu}, \boldsymbol{p}, \sigma)$ with means $\boldsymbol{\mu} \in \mathbb{N}_0^{L_C+L_F+1}$ and weights $\boldsymbol{p} \in [0,1]^{L_C+L_F+1}$ with $\mu_i = 2(i-1)$ and $p_i = \text{Hypergeom}(i \mid N_{\text{total}}, L_C + L_F, \Lambda)$. Further, define per-epoch privacy profile $H(\alpha) = \sup_{x \simeq_{L_C+L_F,\Delta} x'} H_\alpha(M_x \| M_{x'})$. Then,*

$$H(\alpha) \geq \begin{cases} H_\alpha(\underline{P}(\lambda) \| \mathcal{N}(0, \sigma)) & \text{if } \alpha \geq 1, \\ H_\alpha(\mathcal{N}(0, \sigma) \| \underline{P}(\lambda)) & \text{if } 0 \leq \alpha < 1. \end{cases}$$

In short: When applied to a dataset of subsequences $\tilde{x} \subseteq \mathbb{R}^{L_C+L_F}$ constructed from a time series dataset $x \subseteq \mathbb{R}^L$, DP-SGD is less private than the standard privacy guarantee would suggest. This is because there is a chance of including multiple subsequences with sensitive information, which will cause more privacy leakage, as is indicated by the mixture components with larger mean.

In contrast, structured subsampling as shown in Fig. 1 lets us control the maximum number of times that a sensitive element can appear in a batch. In particular, sampling one subsequence per sequence ($\lambda = 1$) ensures that a sensitive element appears at most once. In Fig. 13, we compare the optimistic lower bound from Proposition C.2 to the upper bound for structured subsampling with $\lambda = 1$ from Theorem 4.4. Specifically, we evaluate the bounds for $N_{\text{total}} \in \{10^4, 10^6\}$, $\frac{\Lambda}{N_{\text{total}}} \in \{0.1, 0.001\}$, and $L_C + L_F \in \{1, 2, 4, 8, 16, 32\}$ under varying $\alpha = e^\varepsilon$. Standard DP-SGD is less private than structured subsampling with $\lambda = 1$ for larger subsequence lengths $L_C + L_F$.

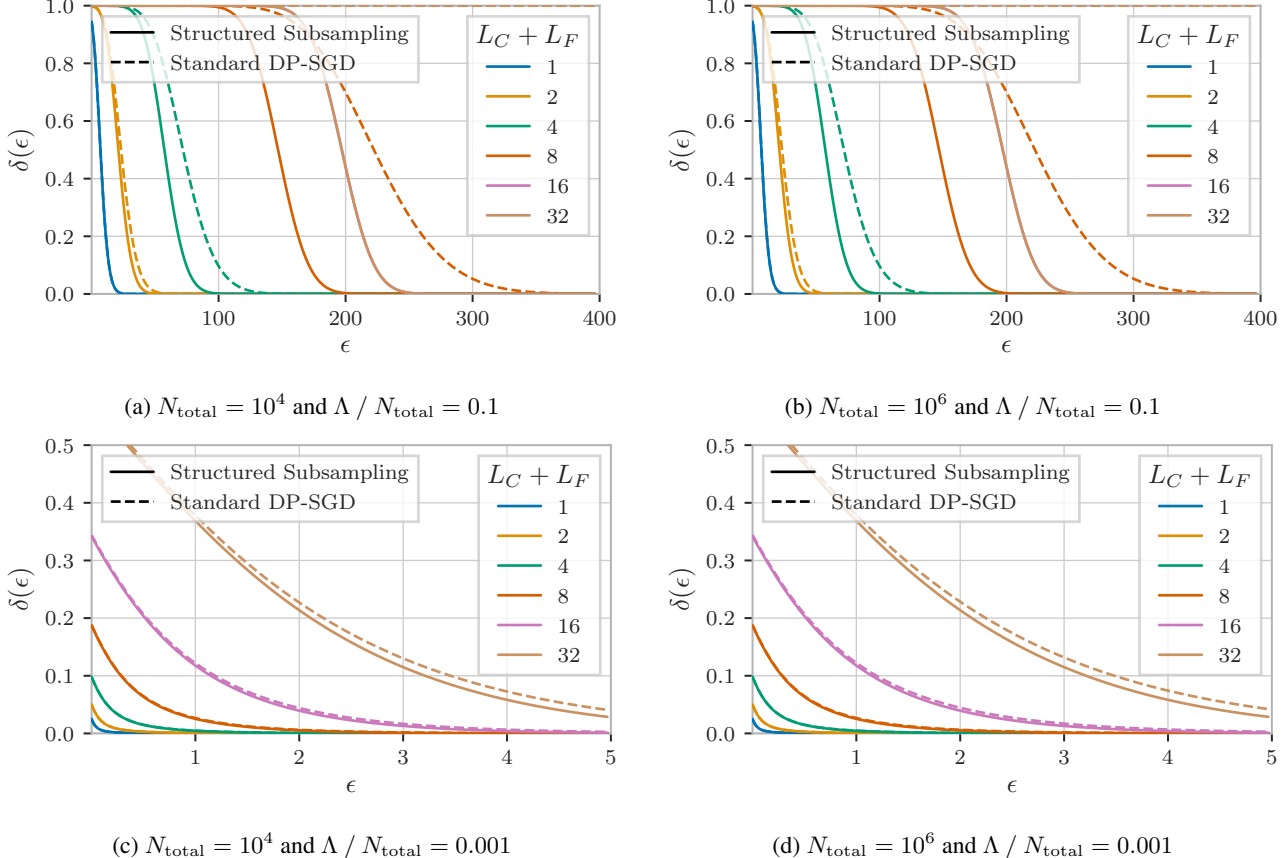

(a) $N_{\text{total}} = 10^4$ and $\Lambda / N_{\text{total}} = 0.1$

(b) $N_{\text{total}} = 10^6$ and $\Lambda / N_{\text{total}} = 0.1$

(c) $N_{\text{total}} = 10^4$ and $\Lambda / N_{\text{total}} = 0.001$

(d) $N_{\text{total}} = 10^6$ and $\Lambda / N_{\text{total}} = 0.001$

Figure 13: Comparison of an optimistic lower bound for standard DP-SGD to the tight upper bound for DP-SGD with structured subsampling and number of subsequences $\lambda = 1$ under varying overall number of subsequences $N_{\text{total}}$, batch size $\Lambda$, and subsequence length $L_C + L_F$. Standard DP-SGD, which first constructs a dataset of all possible subsequences and then samples without replacement, is less private for larger $L_C + L_F$.

# D. Additional Background

In the following, we provide additional background information on the toolset we use to to derive amplification bounds for structured subsampling of time series, determine the corresponding dominating pairs, and perform privacy accounting.

We recommend reading this section before proceeding to our derivations in Appendices E to G which rely on the Lemmata introduced here.

## D.1. Subsampling Analysis

Recall from Section 3 that our goal in subsampling analysis is to determine the privacy of a subsampled mechanism $M = B \circ S$, where $S : \mathbb{X} \to \mathbb{Y}$ is a subsampling scheme that maps from dataset space $\mathbb{X}$ to a space of batches $\mathbb{Y}$ and $B : \mathbb{Y} \to \mathbb{R}^D$ is a base mechanism that maps these batches to outputs. Further recall that the hockey-stick divergence of output distributions $M_x$ and $M_x$ for pairs of datasets $x, x' \in \mathbb{X}$ can be bounded via couplings:

**Definition 3.4.** A coupling $\Gamma$ between distributions $S_x, S_{x'}$ of randomly sampled batches $S(x), S(x') \in \mathbb{Y}$ is a joint distribution on $\mathbb{Y}^2$ whose marginals are $S_x$ and $S_{x'}$.

**Lemma D.1.** *Consider a subsampled mechanism $M = B \circ S$ and datasets $x \simeq x'$. Then, for any coupling $\Gamma$ of subsampling distributions $S_x, S_{x'}$,*

$$H_\alpha(M_x || M_{x'}) \leq \int_{\mathbb{Y}^2} H_\alpha(B_y || B_{y'}) \, \mathrm{d}\Gamma(y, y').$$

*Proof.* The proof is identical to that of Eq. 5 in (Balle et al., 2018). □

Balle et al. (2018) demonstrated that this tool can be combined with the *advanced joint convexity* property of hockey-stick divergences to derive provably tight privacy guarantees for insertion, removal, or substitution of a single element in a dataset:

**Lemma D.2** (Advanced joint convexity). *Consider mixture distributions $P = (1 - p)P_1 + pP_2$ and $Q = (1 - p)Q_1 + pQ_2$ with $P_1 = Q_1$ and some $w \in [0, 1]$. Given $\alpha \geq 1$, define $\alpha' = 1 + \frac{\alpha - 1}{w}$ and $\beta(\alpha) = \frac{\alpha}{\alpha'}$. Then,*

$$H_\alpha(P || Q) = p \cdot H_{\alpha'}(P_2 || (1 - \beta(\alpha)P_1 + \beta(\alpha)Q_2).$$

*Proof.* The proof is identical to that of Theorem 2 in (Balle et al., 2018). In our notation, we interchange $\alpha$ and $\alpha'$. □

More recently, Schuchardt et al. (2024) demonstrated that, when considering insertion and/or removal of multiple elements ("group privacy"), tighter bounds can be obtained via couplings between multiple subsampling distributions.

**Definition D.3.** A coupling $\Gamma$ between distributions $P_1, \ldots, P_N$ on $\mathbb{Y}$ is a distribution on $\mathbb{Y}^N$ with marginals $P_1, \ldots, P_N$.

Specifically, they propose to partition the support of subsampling distribution $S_x$ into events $A_1, \ldots, A_I$ and the support of subsampling distribution $S_{x'}$ into events $E_1, \ldots, E_J$, before defining a simultaneous coupling between the corresponding conditional distributions. For example, these events can indicate the number of group elements that appear in random batches $S(x)$ and $S(x')$.

**Lemma D.4.** *Consider a subsampled mechanism $M = B \circ S$ and datasets $x \simeq x'$, and let $S_x$ and $S_{x'}$ denote the distribution of random batches $S(x)$ and $S(x')$. Define two disjoint partitionings $\bigcup_{i=1}^I A_i = \mathrm{supp}(S_x)$ and $\bigcup_{j=1}^J E_j = \mathrm{supp}(S_{x'})$ of the support of $S_x$ and $S_{x'}$ such that $\forall i : S_x(A_i) > 0$ and $\forall j : S_{x'}(E_j) > 0$. Let $\Gamma$ be a coupling between the corresponding conditional subsampling distributions $S_x(\cdot|A_1), \ldots, S_x(\cdot|A_i), S_{x'}(\cdot|E_1), S_{x'}(\cdot|E_1) \ldots, S_{x'}(\cdot|E_J)$. Then, the hockey-stick divergence between subsampled output distributions $M_x$ and $M_{x'}$ is bounded via*

$$H_\alpha(M_x || M_{x'}) \leq \int_{\mathbb{Y}^{I+J}} c_\alpha(\boldsymbol{y}^{(1)}, \boldsymbol{y}^{(2)}) \, \mathrm{d}\Gamma((\boldsymbol{y}^{(1)}, \boldsymbol{y}^{(2)}))$$

*with cost function $c_\alpha : \mathbb{Y}^I \times \mathbb{Y}^J \to \mathbb{R}_+$ defined by*

$$c_\alpha(\boldsymbol{y}^{(1)}, \boldsymbol{y}^{(2)}) = H_\alpha \left( \sum_{i=1}^I B_{y_i^{(1)}} \cdot S(A_I) || \sum_{j=1}^J B_{y_j^{(2)}} \cdot S_{x'}(A_I) \right). \tag{2}$$

*Proof.* The proof is identical to that of Theorem 3.4 in (Schuchardt et al., 2024), since any distribution supported on a discrete, finite set has a mass function. □

While this bound is slightly more involved, it can be intuitively explained via comparison to Lemma D.1. In Lemma D.1, we couple subsampling distributions $S_x$ and $S'_x$. The resultant bound is a weighted sum of divergences between base mechanism distributions $B_y$ and $B_{y'}$ with batch $y$ from the support of $S_x$ and $y'$ from the support of $S_{x'}$. In Lemma D.4, we couple conditional subsampling distributions $S_x(\cdot|A_1), \ldots, S_x(\cdot|A_i)$ and $S_{x'}(\cdot|E_1), S_{x'}(\cdot|E_1) \ldots, S_{x'}(\cdot|E_J)$. The resultant bound is a weighted sum of divergences between two mixtures. The components of these mixtures are base mechanism distributions $B_{y_i^{(1)}}$ and $B_{y_j^{(2)}}$ with batch $y_i^{(1)}$ from the support of $S_x(\cdot \mid A_i)$ and batch $y_j^{(2)}$ from the support of $S_{x'}(\cdot \mid E_J)$.

The benefit of this formulation is that it allows us to prove that mechanisms are dominated by mixture distributions, rather than individual distributions. This enables the derivation of tight dominating pairs in scenarios where there are multiple possible levels of privacy leakage, such as in Algorithm 1 where a single sensitive element can appear in 0, 1, or multiple subsequences within a batch.

A problem with Eq. (2) is that it requires evaluating the base mechanism distribution $B_y$ for various batches $y \in \mathbb{Y}$, which may be defined by a complicated function, such as the noisy gradient descent update from Algorithm 2. Schuchardt et al. (2024) show that it can be sufficient to consider worst-case batches $\hat{y}_1^{(1)}, \ldots, \hat{y}_I^{(1)}$ and $\hat{y}_1^{(2)}, \ldots, \hat{y}_J^{(2)}$ that maximize the mixture divergences while retaining the pairwise distance between original batches $y_1^{(1)}, \ldots, y_I^{(1)}$ and $y_1^{(2)}, \ldots, y_J^{(2)}$.

**Definition D.5** (Induced distance). Consider an arbitrary neighboring relation $\simeq_\mathbb{Y}$ on batch space $\mathbb{Y}$. Then the corresponding induced distance $d : \mathbb{Y}^2 \to \mathbb{N}$ is a function such that $d(y, y') = K$ implies that there is a sequence of batches $y_1, \ldots, y_{K-1}$ such that $y \simeq_\mathbb{Y} y_1, \forall k : y_k \simeq_\mathbb{Y} y_{k+1}$, and $x_{K-1} \simeq_\mathbb{Y} y'$.

**Lemma D.6.** *Let $d : \mathbb{Y}^2 \to \mathbb{N}$ be the distance induced by a symmetric neighboring relation $\simeq_\mathbb{Y}$ on batch space $\mathbb{Y}$. Consider the tuples of batches $\boldsymbol{y}^{(1)} \in \mathbb{Y}^I$ and $\boldsymbol{y}^{(2)} \in \mathbb{Y}^J$, as well as cost function $c_\alpha$ from Eq. (2). Then,*

$$c_\alpha(\boldsymbol{y}^{(1)}, \boldsymbol{y}^{(2)}) \leq \sup_{\hat{\boldsymbol{y}}^{(1)}, \hat{\boldsymbol{y}}^{(2)}} c_\alpha(\hat{\boldsymbol{y}}^{(1)}, \hat{\boldsymbol{y}}^{(2)})$$

*subject to $\hat{\boldsymbol{y}}^{(1)} \in \mathbb{Y}^I, \hat{\boldsymbol{y}}^{(2)} \in \mathbb{Y}^J$ and*

$$d(\hat{\boldsymbol{y}}_t^{(1)}, \hat{\boldsymbol{y}}_u^{(1)}) \leq d(\boldsymbol{y}_t^{(1)}, \boldsymbol{y}_u^{(1)}) \qquad \forall t, u : 1 \leq t \leq I, 1 \leq u \leq I,$$

$$d(\hat{\boldsymbol{y}}_t^{(1)}, \hat{\boldsymbol{y}}_u^{(2)}) \leq d(\boldsymbol{y}_t^{(1)}, \boldsymbol{y}_u^{(2)}) \qquad \forall t, u : 1 \leq t \leq I, 1 \leq u \leq J,$$

$$d(\hat{\boldsymbol{y}}_t^{(2)}, \hat{\boldsymbol{y}}_u^{(2)}) \leq d(\boldsymbol{y}_t^{(2)}, \boldsymbol{y}_u^{(2)}) \qquad \forall t, u : 1 \leq t \leq J, 1 \leq u \leq J.$$

*Proof.* The proof is identical to that of Proposition 3.5 in (Schuchardt et al., 2024). □

A particular form of this optimization problem, which we will encounter in our later derivations, arises from analyzing sensitivity-bounded Gaussian mechanisms. For a specific set of constraints, it can be shown that the optimal solution is attained by a pair of univariate mixture-of-Gaussians mechanisms (recall Definition 3.3).

**Lemma D.7.** *Consider standard deviation $\sigma \in \mathbb{R}_+$ and mixture weights $\boldsymbol{w}^{(1)} \in [0,1]^I$ and $\boldsymbol{w}^{(2)} \in [0,1]^J$. Let $\Omega$ be the set of all pairs of Gaussian mixtures $(P, Q)$ with $P = \sum_i^I w_i^{(1)} \cdot \mathcal{N}(\boldsymbol{\mu}_i^{(1)}, \sigma^2 \boldsymbol{I})$ and $Q = \sum_j^J w_j^{(1)} \cdot \mathcal{N}(\boldsymbol{\mu}_j^{(2)}, \sigma^2 \boldsymbol{I})$ satisfying*

$$\boldsymbol{\mu}_1^{(1)} = \boldsymbol{\mu}_1^{(2)} = \boldsymbol{0}$$

$$||\boldsymbol{\mu}_t^{(1)} - \boldsymbol{\mu}_u^{(1)}||_2 \leq c \cdot |t - u| \qquad \forall t, u : 1 \leq t \leq I, 1 \leq u \leq I,$$

$$||\boldsymbol{\mu}_t^{(2)} - \boldsymbol{\mu}_u^{(2)}||_2 \leq c \cdot |t - u| \qquad \forall t, u : 1 \leq t \leq J, 1 \leq u \leq J.$$

*with some constant $c \in \mathbb{R}_+$. Then,*

$$\max_{P,Q \in \Omega} H_\alpha(P, Q) = H_\alpha \left( MoG(\boldsymbol{\mu}^{(1)*}, \boldsymbol{w}^{(1)}, \sigma) || MoG(\boldsymbol{\mu}^{(2)*}, \boldsymbol{w}^{(2)}, \sigma) \right)$$

*with univariate means $\boldsymbol{\mu}^{(1)*} = \begin{pmatrix} 0 & -1 & \cdots & -I \cdot c \end{pmatrix}^T$ and $\boldsymbol{\mu}^{(2)*} = \begin{pmatrix} 0 & 1 & \cdots & J \cdot c \end{pmatrix}^T$.*

*Proof.* After scaling the standard deviation by $c$, the proof is identical to that of Theorem 0.7 in (Schuchardt et al., 2024). □

## D.2. Dominating Pairs

In the following, we summarize multiple known results on dominating pairs due to Zhu et al. (2022) that we will need in our later derivations. Let us begin by recalling the definition of dominating pairs:

**Definition 3.2.** Distributions $(P, Q)$ are a *dominating pair* for component mechanism $M$ if $\sup_{x \simeq x'} H_\alpha(M_x || M_{x'}) \leq H_\alpha(P || Q)$ for all $\alpha \geq 0$. If the bound holds with equality for all $\alpha \geq 0$, then $(P, Q)$ are a *tight dominating pair*.

Further recall that a tight dominating pair is a pair of distributions $(P, Q)$ such that the bound in Definition 3.2 holds with equality. All mechanisms have tight dominating pairs:

**Lemma D.8.** *Any mechanism $M : \mathbb{X} \to \mathbb{R}^D$ has a tight dominating pair of distributions, i.e., a pair of distributions $(P, Q)$ such that $\sup_{x \simeq x'} H_\alpha(M_x || M_{x'}) = H_\alpha(P || Q)$ for all $\alpha \geq 0$.*

*Proof.* This result is a special case of Proposition 8 from (Zhu et al., 2022) for real-valued outputs. □

For privacy accounting, we will later need to evaluate $H_\alpha(P, Q)$ as a function of $\alpha$. Such functions are referred to as *privacy profiles* (Zhu et al., 2022):

**Definition D.9.** A privacy profile is a function $H : \mathbb{R}_+ \to \mathbb{R}$ such that there exists a pair of distributions $(P, Q)$ with $\forall \alpha \geq 0 : H(\alpha) = H_\alpha(P, Q)$.

Under symmetric neighboring relations (e.g. substitution of elements in a set), any privacy profile corresponding to a dominating pair enjoys a symmetry that allows us to focus many of our derivations on $\alpha \geq 1$:

**Lemma D.10.** *Let $\simeq$ be a symmetric neighboring relation on dataset space $\mathbb{X}$. It then holds that $\sup_{x \simeq x'} H_\alpha(M_x || M_{x'}) \leq H_\alpha(P, Q)$ for all $\alpha \geq 1$ if and only if $\sup_{x \simeq x'} H_\alpha(M_x || M_{x'}) \leq H_\alpha(Q, P)$ for all $0 < \alpha \leq 1$.*

*Proof.* This result corresponds to the third part of Lemma 31 from (Zhu et al., 2022). □

Later on, we will use the following properties of privacy profiles in identifying dominating pairs:

**Lemma D.11.** *A function $H : \mathbb{R}_+ \to \mathbb{R}$ is a privacy profile if and only if $H$ is convex, $H$ is decreasing, $H(0) = 1$, and $H(x) \geq \max\{1 - x, 0\}$.*

*Proof.* This result, in combination with Definition D.9, corresponds to the first part of Lemma 9 in (Zhu et al., 2022). □

Furthermore, the following result shows that for the purpose of deriving dominating pairs, it is in principle sufficient to derive privacy profiles:[2]

**Lemma D.12.** *Let $H : \mathbb{R}_+ \to \mathbb{R}$ be a privacy profile. Then there exists a pair of univariate distributions $(P, Q)$ with $H(\alpha) = H_\alpha(P, Q)$. Specifically, distribution $P$ is supported on $[0, 1)$ and has CDF $1 + H * (x - 1)$, where $H^*$ is the convex conjugate of privacy profile $H$. Distribution $Q$ is $\mathrm{Uniform}([0, 1])$.*

*Proof.* This result corresponds to the second part of Lemma 9 in (Zhu et al., 2022). □

## D.3. Privacy Accounting

Our ultimate objective is to provide privacy guarantees for an entire training run. The following result motivates why we can achieve this objective by deriving dominating pairs for each update step.

**Lemma D.13.** *Consider a component mechanism $M : \mathbb{X} \to \mathbb{R}^D$ and a component mechanism with auxiliary input $M' : \mathbb{X} \times \mathbb{R}^D \to \mathbb{R}^D$. Assume that $(P, Q)$ is a dominating of $M$ and $(P', Q')$ is a dominating pair of $x \mapsto M'(x, o)$ for all auxiliary inputs $o \in \mathbb{R}^D$. Then, the product measures $(P \times P', Q \times Q')$ supported on $\mathbb{R}^D \times \mathbb{R}^D$ are a dominating pair of composed mechanism $x \to M'(x, M(x))$.*

*Proof.* This result corresponds to Theorem 10 in combination with Footnote 3 from (Zhu et al., 2022) for the special case of real-valued co-domains. □

---

[2]Note that this result also implies that multiple dominating pairs may share the same privacy profile.

In the case of DP-SGD, the auxiliary inputs are the model parameters that resulted from the previous update step. This characterization of the composed mechanism's privacy in terms of product measures is in fact the tightest possible characterization given dominating pairs of the components (see discussion of Theorem 3.2 in (Dong et al., 2022)).

While Lemma D.13 lets us easily define dominating pairs for composed mechanisms, obtaining privacy parameters $(\varepsilon, \delta)$ by evaluating the privacy profile $\alpha \mapsto H_\alpha(P \times P' || Q \times Q')$ can be challenging. There exists a variety of solutions to this problem, such as moments accounting (Abadi et al., 2016; Mironov, 2017) or central limit theorems of composition (Sommer et al., 2019; Dong et al., 2022).

*Privacy loss distribution* (PLD) accounting is a family of SOTA approaches that enable tight numeric privacy accounting with arbitrary accuracy using the notion of privacy loss random variables introduced in (Dwork & Rothblum, 2016):

**Definition D.14.** Consider a pair of distributions $(P, Q)$. The corresponding privacy loss random variable $\mathcal{L}_{P,Q}$ is the random variable $\log\left(\frac{dP}{dQ}(o)\right)$ with $o \sim P$. Similarly, $\mathcal{L}_{Q,P}$ is the random variable $\log\left(\frac{dP}{dQ}(o)\right)$ with $o \sim Q$.

Note that one can easily convert between dominating pairs, privacy profiles, and distributions of privacy loss random variables (see Fig. 2 in (Zhu et al., 2022)). In particular, privacy profiles can be computed from privacy loss distributions as follows:

**Lemma D.15.** *Consider a pair of distributions $(P, Q)$ with corresponding privacy loss random variables $\mathcal{L}_{P,Q}$ and $\mathcal{L}_{Q,P}$. Then, $H_\alpha(P, Q) = \Pr[\mathcal{L}_{P,Q} > \log(\alpha)] - \alpha \Pr[\mathcal{L}_{Q,P} < \log(\alpha)]$ for all $\alpha \geq 0$.*

*Proof.* The proof is identical to that of Theorem 5 from (Balle & Wang, 2018), substituting $e^\varepsilon$ with $\alpha$. $\qquad\square$

Due to the logarithm in Definition D.14, the privacy loss random variable corresponding to the dominating pair $(P \times P', Q \times Q')$ of our composed mechanism is simply the sum $\mathcal{L}_{P,Q} + \mathcal{L}_{P',Q'}$ of the components' privacy loss random variables. A key insight underlying numeric privacy loss distribution accounting is that the density of the composed PLD density is thus simply a convolution of the components' PLD densitities (Meiser & Mohammadi, 2018; Sommer et al., 2019). The composed PLD can thus be efficiently computed using Fast Fourier Transforms, as proposed by Koskela et al. (2020).

A challenge in numerical PLD accounting is that it requires quantizing the distribution of $\mathcal{L}_{P,Q}$ while retaining sound privacy guarantees. Doroshenko et al. (2022) show how to optimally perform such a pessimistic quantization.

**Lemma D.16.** *Consider any pair of distributions $(P, Q)$ and finite set of quantization thresholds $\mathcal{E} = \{\varepsilon_0, \ldots, \varepsilon_K\}$ with $\varepsilon_k \in \mathbb{R} \cup \{+\infty, -\infty\}$ and $-\infty = \varepsilon_0 < \varepsilon_1 < \cdots < \varepsilon_k = +\infty$. There exists a pair of distributions $(\hat{P}^\uparrow \hat{Q}^\uparrow)$ such that the distribution of privacy loss $\mathcal{L}_{\hat{P}^\uparrow, \hat{Q}^\uparrow}$ is only supported on $\mathcal{E}$ and $(\hat{P}^\uparrow \hat{Q}^\uparrow)$ dominates $(P, Q)$. Furthermore, $(\hat{P}^\uparrow \hat{Q}^\uparrow)$ is dominated by any other distribution supported on $\mathcal{E}$ and is uniquely defined by $\{(\varepsilon, H_{e^\varepsilon}(P, Q)) \mid \varepsilon \in \mathcal{E}\}$.*

*Proof.* This result is an immediate consequence of Lemma 4.1 from (Doroshenko et al., 2022) and the fact that their Algorithm 1 takes only $\{H_{e^\varepsilon}(P, Q) \mid \varepsilon \in \mathcal{E}\}$ as an input. $\qquad\square$

Note that the last part of Lemma D.16 means that constructing a pessimistic dominating pair for PLD accounting with quantized distributions only requires access to privacy profile $\alpha \mapsto H_\alpha(P, Q)$ at some finite set of points, rather than to $(P, Q)$ themselves. This is similar to Lemma D.12, which showed that constructing a tight dominating pair only requires access to privacy profile $\alpha \mapsto H_\alpha(P, Q)$ at arbitrary $\alpha \geq 0$.

**Summary:** To summarize, we can perform optimal numeric privacy accounting for a composed mechanism by: (1) Determining dominating pairs of the component mechanisms, (2) optimally quantizing the resultant privacy loss distribution using the "connecting the dots" method from (Doroshenko et al., 2022) (3) computing the composed privacy loss distribution using the Fast Fourier method of (Koskela et al., 2020), and (4) determining privacy parameters $(\varepsilon, \delta)$ of the composed mechanism via Lemma D.15 derived by Balle et al. (2018).

As steps 2-4 are standard algorithms implemented in libraries like Google's `privacy_accounting` library (Google Differential Privacy Team, 2024), we can focus our analysis on determining dominating pairs for the component mechanisms, i.e., the training steps or epochs of DP-SGD for time series forecasting.

# E. Proofs from Section 4.1 (Bottom-Level Sampling)

In the following, we derive (tight) pessimistic guarantees, as well as optimistic lower bounds for epoch-level accounting when using top-level deterministic iteration and bottom-level sampling with replacement (Appendix E.1) or bottom-level Poisson sampling (Appendix E.2). In addition, we prove that there is an optimal choice for number of subsequences $\lambda \in \mathbb{N}$ (Appendix E.3):

**Theorem 4.1.** *Let $P^*(\lambda)$, $Q^*(\lambda)$ be a tight dominating pair of epoch $M$ for bottom-level Poisson sampling or sampling with replacement and $\lambda \in \mathbb{N}$ (expected) subsequences. Then $H_\alpha(P^*(\lambda), Q^*(\lambda))$ is minimized by $\lambda = 1$ for all $\alpha \geq 0$.*

Note that all proofs and statements of tightness assume learning without hidden states — just like other works on privacy accounting for DP-SGD (e.g. (Abadi et al., 2016; Wang et al., 2019; Koskela et al., 2020; Gopi et al., 2021)). That is, each epoch of length $K$ applied to a model $f_\theta$ with parameters $\theta \in \mathbb{R}^D$ is a mechanism $M : \mathbb{X} \to \mathbb{R}^{K \times D}$ that does not simply release the final updated model parameters, but the gradient used in each update step. By the post-processing property of differential privacy (Dwork, 2006; Dong et al., 2022), releasing gradients at each step is at least as private as releasing the updated model parameters at each step and vice-versa (assuming a fixed and known learning rate).

Our privacy guarantees can likely be tightened if we were to assume learning with hidden states (Ye & Shokri, 2022), i.e., only release the final updates model parameter via a mechanism $M : \mathbb{X} \to \mathbb{R}^D$. However, such analyses based on amplification by iteration (Feldman et al., 2018) generally require some assumptions about convexity and/or smoothness of the loss landscape, which we cannot make in our general treatment of deep time series forecasting.

### E.1. Sampling With Replacement

Our main objective for top-level deterministic iteration (see Algorithm 3) and bottom-level sampling with replacement (see Algorithm 4) will be proving the the following tight pessimistic bound:

**Theorem 4.2.** *Consider the number of sampled subsequences $\lambda = 1$, and let $r = \frac{L_C + L_F}{L - L_F + 1}$ be the probability of sampling a subsequence containing any specific element. Define $P(1) = MoG(\boldsymbol{\mu}, \boldsymbol{p}, \sigma)$ with means $\boldsymbol{\mu} = \begin{bmatrix} 0 & 2 \end{bmatrix}^T$ and weights $\boldsymbol{p} = \begin{bmatrix} 1 - r & r \end{bmatrix}^T$. Further, define per-epoch privacy profile $H(\alpha) = \sup_{x \simeq_E x'} H_\alpha(M_x || M_{x'})$. Then,*

$$H(\alpha) = \begin{cases} H_\alpha(P(1) || \mathcal{N}(0, \sigma)) & \text{if } \alpha \geq 1, \\ H_\alpha(\mathcal{N}(0, \sigma) || P(1)) & \text{if } 0 \leq \alpha < 1. \end{cases}$$

We shall further prove the following pessimistic and optimistic bounds for $\lambda \geq 1$:

**Theorem E.1.** *Consider number of sampled subsequences $\lambda \geq 1$, and let $r = \frac{L_C + L_F}{L - L_F + 1}$ be the probability of sampling a subsequence containing any specific element. Define $\overline{P}(\lambda) = MoG(-1 \cdot \boldsymbol{\mu}, \boldsymbol{p}, \sigma)$ and $\overline{Q}(\lambda) = MoG(\boldsymbol{\mu}, \boldsymbol{p}, \sigma)$ with means $\boldsymbol{\mu} = \begin{bmatrix} 0 & 2 & 4 & \cdots & 2 \cdot \lambda \end{bmatrix}^T$ and weights $\boldsymbol{p} \in [0, 1]^{\lambda+1}$ with $p_i = \text{Binomial}(i - 1 \mid \lambda, r)$. Further define per-epoch privacy profile $H(\alpha) = \sup_{x \simeq_E x'} H_\alpha(M_x || M_{x'})$. Then,*

$$H(\alpha) \leq H_\alpha(\overline{P}(\lambda) || \overline{Q}(\lambda))$$

**Theorem 4.3.** *Consider the number of subsequences $\lambda \geq 1$, and let $r = \frac{L_C + L_F}{L - L_F + 1}$. Define $\underline{P}(\lambda) = MoG(\boldsymbol{\mu}, \boldsymbol{p}, \sigma)$ with means $\boldsymbol{\mu} \in \mathbb{N}_0^{\lambda+1}$ and weights $\boldsymbol{p} \in [0, 1]^{\lambda+1}$ with $\mu_i = 2(i - 1)$ and $p_i = \text{Binomial}(i \mid \lambda, r)$. Further, define per-epoch privacy profile $H(\alpha) = \sup_{x \simeq_E x'} H_\alpha(M_x || M_{x'})$. Then,*

$$H(\alpha) \geq \begin{cases} H_\alpha(\underline{P}(\lambda) || \mathcal{N}(0, \sigma)) & \text{if } \alpha \geq 1, \\ H_\alpha(\mathcal{N}(0, \sigma) || \underline{P}(\lambda)) & \text{if } 0 \leq \alpha < 1. \end{cases}$$

To this end, we will

1. Prove that, under top-level deterministic iteration, the privacy of our epoch-level mechanism can be upper-bounded by analyzing a training step for a single batch,
2. prove that the privacy of this training step can be upper-bounded by analyzing sampling with replacement under group substitution,

3. derive pessimistic upper bounds for sampling with replacement under group substitution,
4. determine optimistic lower bounds that coincide with the upper bound for $\lambda = 1$ by constructing worst-case time series datasets,
5. determine dominating pairs corresponding to our pessimistic upper bounds.

### E.1.1. REDUCTION FROM EPOCH- TO STEP-LEVEL PRIVACY

Consider two sets of sequences $x = \{x_1, \ldots, x_N\}$ and $x = \{x'_1, \ldots, x'_N\}$ with $x \overset{1}{\simeq}_E$, i.e., $x_n \neq x'_n$ for exactly one $n$. If we partition these sets into subsets of size $N' = \lfloor \Lambda / \lambda \rfloor$ in a data-independent manner and use each subset for exactly one training step, we know that only one of these steps will access the modified sequence $x_n$ and potentially leak its sensitive information.

The following statement formalizes this idea, which essentially corresponds to an adaptive form of the parallel composition property of differential privacy (McSherry, 2009), expressed in the language of dominating pairs.

**Lemma E.2** (Adaptive parallel composition.). *Consider some data space $\mathbb{A}$ and a dataset space $\mathbb{X} = \{x \in \mathcal{P}(\mathbb{A}) \mid |a| = N\}$ with $N = K \cdot N'$ for some $K, N' \in \mathbb{N}$. Further consider a sequence of $K$ adaptive mechanisms $M^{(1)}, \ldots, M^{(K)}$ with $M^{(k)} : \mathbb{A}^{N'} \times \mathbb{R}^{(k-1) \times D} \to \mathbb{R}^D$. Define the adaptively composed mechanism*

$$M(x) = M^{(K)}\left(\{x_{N-N'+1}, \ldots, x_N\}, M^{(K-1)}\left(\{x_{N-2N'+1}, \ldots, x_{N-N'}\}, M^{(K-2)}(\ldots)\right)\right)$$

*that lets the component mechanisms only access disjoint subsets of input dataset $x$. Let $\simeq_\Delta$ be the substitution relation. Let $P^{(k)}, Q^{(k)}$ be a dominating pair of $M^{(k)}(\cdot, z)$ under $\simeq_\Delta$ for all size-$N'$ subsets $z$ of $\mathbb{A}$. Then, for all $\alpha \geq 0$,*

$$\sup_{x \simeq_\Delta x'} H_\alpha(M_x || M_{x'}) \leq \max_k H_\alpha(P^{(k)} || Q^{(k)}). \tag{3}$$

*Proof.* With a slight abuse of notation, let us in the following write $M^{(k)}(y, z)$ for the output distribution $M_{y,z}$ of any component mechanism $M^{(k)}$ given input $y$ and auxiliary input $z$, instead of its output random variable. Consider any pair of datasets $x \simeq_\Delta x'$. Assume w.l.o.g. that the single pair of elements $x_n$ and $x'_n$ with $x_n \neq x'_n$ is accessed by the $k$th mechanisms. Since $P^{(k)}, Q^{(k)}$ is a dominating pair under substitution for all size-$N'$ auxiliary inputs $z$, we know that

$$\max_z H_\alpha\left(M^{(k)}(\{x_{N-(K-k+1)N'+1}, \ldots, x_{N-(K-k)N'}\}, z), M^{(k)}(\{x'_{N-(K-k+1)N'+1}, \ldots, x'_{N-(K-k)N'}\}, z)\right)$$
$$\leq H_\alpha(P^{(k)} || Q^{(k)})$$

Since all component mechanisms access disjoint subsets, we further know for any $l \neq k$ that

$$\max_z H_\alpha\left(M^{(l)}(\{x_{N-(K-l+1)N'+1}, \ldots, x_{N-(K-l)N'}\}, z), M^{(l)}(\{x'_{N-(K-l+1)N'+1}, \ldots, x'_{N-(K-l)N'}\}, z)\right)$$
$$= \max\{1 - \alpha, 0\}$$
$$\leq H_\alpha(N(0, \sigma), N(0, \sigma))$$

for some $\sigma \in \mathbb{R}_+$.

Define $P = P^{(k)} \times (N(0, \sigma) \times \cdots \times N(0, \sigma))$ and $Q = P^{(k)} \times (N(0, \sigma) \times \cdots \times N(0, \sigma))$. From the above two inequalities and the composition theorem for dominating pairs (see Lemma D.13, and see proof of Theorem 27 in (Zhu et al., 2022) for intermediate steps), we know that $H_\alpha(M_x || M_{x'}) \leq H_\alpha(P, Q)$. Thus, by definition of hockey stick divergence $H_\alpha$,

$$H_\alpha(M_x || M_{x'})$$
$$\leq \int_{\mathbb{R}^{K \times D}} \max\left\{0, \frac{dP}{dQ}(\boldsymbol{O}) - \alpha\right\} dQ(\boldsymbol{O})$$
$$= \int_{\mathbb{R}^{K \times D}} \max\left\{0, \frac{dP^{(k)}}{dQ^{(k)}}(\boldsymbol{O}_k) - \alpha\right\} dQ(\boldsymbol{O})$$
$$= \int_{\mathbb{R}^{K \times D}} \max\left\{0, \frac{dP^{(k)}}{dQ^{(k)}}(\boldsymbol{O}_k) - \alpha\right\} dQ^{(k)}(\boldsymbol{O}_k)$$
$$= H_\alpha(P^{(k)} || Q^{(k)}),$$

where the second-to-last equality follows from marginalizing the output of all mechanisms $M^{(l)}$ with $l \neq k$. Our result from Eq. (3) then follows immediately from taking the supremum over all pairs of datasets $x \simeq_\Delta x'$ and the maximum over all steps $k$ in which the substituted elements $x_n \neq x'_n$ can appear. $\qquad\square$

### E.1.2. REDUCTION TO SUBSAMPLED GROUP PRIVACY

Let us assume w.l.o.g. that $x_1 \neq x'_1$ for our two datasets $x \overset{1}{\simeq}_E x'$. That is, the single modified time series contributes to the first pair of top-level batches $x^{(1)} = \{x_1, x_2, \ldots, x_{N'}\}$ and $x'^{(1)} = \{x'_1, x_2, \ldots, x_{N'}\}$ with $N' = \lfloor \Lambda / \lambda \rfloor$.

Let $M^{(k)}$ be the mechanism that yields bottom-level subsampled, clipped, summed, and noised gradients for the $k$th top-level batch. Since all top-level batches are disjoint, we can apply the parallel composition lemma from Lemma E.2. Since all top-level batches $x^{(k)}, x^{(k)}$ are identical for $k > 1$, we already know that the $k$th step is perfectly private, i.e., $\forall k > 1 : H_\alpha(M^{(k)}_{x^{(k)}} || M^{(k)}_{x^{(k)}}) = \max\{1 - \alpha, 0\}$. Thus, the maximum in Eq. (3) will be attained by $M^{(1)}$ and we can focus on analyzing the privacy of the first gradient step.

For the following proof, let $\check{S} : \mathbb{R}^L \to \mathcal{P}(\mathbb{R}^{L_C + L_F})$ be the bottom-level subsampling function described by Algorithm 4 that takes a single sequence and yields multiple subsequences of length $L_C + L_F$. Let $G : \mathcal{P}(\mathbb{R}^{L_C + L_F}) \to \mathbb{R}$ be the function that yields clipped and summed per-subsequence gradients for a set of subsequences, i.e., Algorithm 2 without adding Gaussian noise.

The next result shows that we only need to analyze the privacy of the gradients for modified time series $x_1$ and $x'_1$, rather than the entire top-level batch $x^{(1)}$:

**Lemma E.3.** *Let $M^{(1)} : \mathcal{P}(\mathbb{R}^L) \to \mathcal{P}(\mathbb{R}^D)$ be the mechanism that yields bottom-level subsampled, clipped, summed, and noised gradients for the first top-level batch. Let $\check{M} : \mathbb{R}^L \to \mathbb{R}^D$ with $\check{M}(x_n) = Z + (G \circ \check{S})(x_n)$ and $Z \simeq \mathcal{N}(0, \sigma^2 C^2 I)$ be the mechanism that yields bottom-level subsampled, clipped, summed, and noised gradients for a single sequence. Consider an arbitrary pair of top-level batches $x^{(1)} \overset{1}{\simeq}_E x'^{(1)}$ with $x_1 \neq x'_1$ where $x_1 \in x^{(1)}$ and $x'_1 \in x'^{(1)}$. Then, for all $\alpha \geq 0$,*

$$H_\alpha(M_{x^{(1)}} || M_{x'^{(1)}}) \leq \sup_{x_1, x'_1} H_\alpha(\check{M}_{x_1} || \check{M}_{x'_1}) \quad s.t. \ \{x_1\} \overset{1}{\simeq}_E \{x'_1\}.$$

*Proof.* Consider a pair of top-level batches $x^{(1)} = \{x_1, x_2, \ldots, x_{N'}\}$ and $x'^{(1)} = \{x'_1, x_2, \ldots, x_{N'}\}$.

Since addition of Gaussian gradient noise commutes with summing over per-sequence gradients, we can write $M^{(1)}(x^{(1)})$ via

$$M^{(1)}(x^{(1)}) = Z + \sum_{n=1}^{N'} (G \circ \check{S})(x_i) = Z + (G \circ \check{S})(x_1) + \sum_{n=2}^{N'} (G \circ \check{S})(x_i) = \check{M}(x_1) + \sum_{n=2}^{N'} (G \circ \check{S})(x_i).$$

By the post-processing property of differential privacy (Dwork, 2006; Dong et al., 2022), this sum over per-sequence gradients, in which only the first one is randomly perturbed, is at least as private as releasing the per-sequence gradients individually. That is, the following mechanism attains greater or equal hockey stick divergence:

$$x^{(1)} \mapsto \left( \check{M}(x_1), (G \circ \check{S})(x_2), \ldots, (G \circ \check{S})(x_{N'}) \right).$$

This is a (non-)adaptive parallel composition, i.e., we can apply Lemma E.2. Since $x_n = x'_n$ for all $n > 1$, we already know that all outputs except the first one are perfectly private, i.e., have privacy profile $\alpha \mapsto \max\{1 - \alpha, 0\}$. Thus, the maximum in Eq. (3) must be attained by $H_\alpha(\check{M}_{x_1} || \check{M}_{x'_1})$ for some worst-case choice of sequences $x_1, x'_1$ that differ in one element, i.e., $\{x_1\} \overset{1}{\simeq}_E \{x'_1\}$. $\qquad\square$

Finally, recall from Algorithm 4 that $\check{S} : \mathbb{R}^L \to \mathcal{P}(\mathbb{R}^{L_C + L_F})$ samples $\lambda$ subsequences of length $L_C + L_F$ with replacement from the $T = L - L_F + 1$ available subsequences. Furthermore, exactly $L_C + L_F$ such subsequences differ between sequence $x_1$ and $x'_1$. Abstracting away from our time series context, this is equivalent to privacy under group substitution:

**Definition E.4.** Consider a dataset space $\mathbb{X} = \mathcal{P}(\mathbb{A})$ with underlying set $\mathbb{A}$. Two datasets $x, x' \in \mathbb{X}$ of size $T$ are $k$-group-substitution neighboring ($x \simeq_{k,\Delta} x'$) if there are groups $g \subseteq x$ and $g' \subseteq x'$ with $|g| = |g'| = k$ and $x \setminus g = x' \setminus g'$.

Furthermore, abstracting away from our machine learning context, our gradient mechanism $G(\cdot) + Z$ with $Z \simeq \mathcal{N}(0, \sigma^2 C^2 I)$ is simply a calibrated Gaussian mechanism with an underlying function of bounded sensitivity $\Delta_2 = C$ (Song et al., 2013):

**Definition E.5.** Consider a batch space $\mathbb{Y} = \mathcal{P}(\mathbb{A})$ or $\mathbb{Y} = \mathcal{P}_{\mathrm{multi}}(\mathbb{A})$, where $\mathcal{P}_{\mathrm{multi}}(\mathbb{A})$ is the set of all multisets that can be constructed from underlying set $\mathbb{A}$. A function $f : \mathbb{Y} \to \mathbb{R}^D$ has $\ell_2$-sensitivity $\Delta_2$ under insertion/removal ($\simeq_\pm$) if $\forall y, y' \in \mathbb{Y} : y \simeq_\pm y' \implies \|f(y) - f(y')\| \leq \Delta_2$.

Based on our results from this and the previous section, we can focus on analyzing the privacy of such mechanisms under group substitution.

### E.1.3. Sampling with Replacement under Group Substitution

In the following, we apply the conditional coupling approach from (Schuchardt et al., 2024) (recall Lemma D.4). That is, we partition the support of our subsampling distributions into events and define a joint coupling between the corresponding subsampling distributions.

**Lemma E.6.** *Consider a dataset space $\mathbb{X} = \mathcal{P}(\mathbb{A})$ with underlying set $\mathbb{A}$. and batch space $\mathbb{Y} = \mathcal{P}_{\mathrm{multi}}(\mathbb{A})$. Let $S : \mathbb{X} \to \mathbb{Y}$ be subsampling with replacement with batch size $\lambda$. Let base mechanism $B : \mathbb{Y} \to \mathbb{R}^D$ be a Gaussian mechanism $B(y) = f + Z$ with $Z \sim \mathcal{N}(0, \Delta_2 \sigma^2 \boldsymbol{I})$, where $\Delta_2$ is the $\ell_2$-sensitivity of underlying function $f : \mathbb{Y} \to \mathbb{R}^D$ under insertion/removal ($\simeq_\pm$). Define subsampled mechanism $M = B \circ S$ and consider $k$-group-substitution neighboring datasets $x \simeq_{k,\Delta} x'$ of size $T$. Then, for all $\alpha \geq 0$,*

$$H_\alpha(M_x \| M_{x'}) \leq \max_{P,Q \in \Omega} H_\alpha(P \| Q), \tag{4}$$

*where $\Omega$ is set of all pairs of multivariate Gaussian mixtures $(P, Q)$ with $P = \sum_{i=1}^{\lambda+1} w_i \cdot \mathcal{N}(\boldsymbol{\mu}_i^{(1)}, \sigma^2 \boldsymbol{I})$ and $Q = \sum_{j=1}^{\lambda+1} w_j \cdot \mathcal{N}(\boldsymbol{\mu}_j^{(2)}, \sigma^2 \boldsymbol{I})$ satisfying*

$$
\begin{aligned}
\|\boldsymbol{\mu}_i^{(1)} - \boldsymbol{\mu}_j^{(1)}\|_2 &\leq 2 \cdot |i - j| & \forall i, j : 1 \leq i \leq \lambda + 1, 1 \leq u \leq \lambda + 1 \\
\|\boldsymbol{\mu}_i^{(2)} - \boldsymbol{\mu}_j^{(2)}\|_2 &\leq 2 \cdot |i - j| & \forall i, j : 1 \leq i \leq \lambda + 1, 1 \leq u \leq \lambda + 1 \\
\|\boldsymbol{\mu}_i^{(1)} - \boldsymbol{\mu}_j^{(2)}\|_2 &\leq 2 \cdot \max\{i, j\} & \forall i, j : 1 \leq i \leq \lambda + 1, 1 \leq u \leq \lambda + 1,
\end{aligned} \tag{5}
$$

*with $\forall i : \boldsymbol{\mu}_i^{(1)} \in \mathbb{R}^D$ and $w_i = \mathrm{Binomial}(i - 1 \mid \lambda, r)$ and $r = \frac{k}{T}$.*

*Proof.* Since $x \simeq_{k,\Delta} x'$, there must be two groups $g = \{a_1, \ldots, a_k\}, g' = \{a'_1, \ldots, a'_k\}$ with $|g| = |g'| = k$ and $x \setminus g = x \setminus g'$ whose elements we assign an arbitrary ordering. Let $A_i = \{y \subseteq x \mid y \cap g = i\}$ and $E_j = \{y \subseteq x' \mid y \cap g' = j\}$ be the events that $i$ and $j$ group elements are sampled from $x$ and $x'$, respectively. Note that $\subseteq$ refers to sub-multisets, where a single element can be sampled multiple times. As can be easily verified, $S_x(A_i) = \mathrm{Binomial}(i \mid \lambda, r)$ and $S_{x'}(E_j) = \mathrm{Binomial}(j \mid \lambda, r)$

Let $s_x : \mathbb{Y} \to [0, 1]$ and $s_{x'} : \mathbb{Y} \to [0, 1]$ be the densities of subsampling distributions $S_x$ and $S_{x'}$. Since any batch that contains $i$ or $j$ group elements under condition $A_i$ or $A_j$ is equally likely we have

$$s_x(y \mid A_i) \propto \mathbb{1}\left[y \in A_i\right], \qquad s_x(y \mid E_i) \propto \mathbb{1}\left[y \in E_J\right].$$

We can thus define a joint coupling $\Gamma$ of $S_x(\cdot \mid A_0), \ldots, S_x(\cdot \mid A_\lambda), S_{x'}(\cdot \mid E_0), \ldots, S_{x'}(\cdot \mid E_\lambda)$ via the following mass function $\gamma : \mathbb{Y}^{2 \cdot (\lambda+1)} \to [0, 1]$:

$$\gamma(\boldsymbol{y}^{(1)}, \boldsymbol{y}^{(2)}) \propto s_x(y^{(1)} \mid A_0) \cdot \prod_{i=0}^{\lambda-1} \gamma(y_{i+1}^{(1)} \mid y_i^{(1)}) \cdot \prod_{i=0}^{\lambda} \gamma(y_i^{(2)} \mid y_i^{(1)})$$

with

$$\gamma(y_{i+1}^{(1)} \mid y_i^{(1)}) \propto \mathbb{1}\left[\exists a \in y_i^{(1)}, a' \in g : a \notin g \wedge y_{i+1}^{(1)} = y_i^{(1)} \setminus \{a\} \cup \{a'\}\right]$$

$$\gamma(y_i^{(2)} \mid y_i^{(1)}) = \mathbb{1}\left[y_i^{(2)} \setminus g' = y_i^{(1)} \setminus g\right] \cdot \mathbb{1}\left[\forall 1 \leq l \leq k : a_l \in y_i^{(1)} \implies a'_l \in y_i^{(2)}\right]$$

In short, we sample uniformly at random with replacement a multiset that does not contain any elements from group $g$ to obtain $y_0^{(1)}$. Then, we iteratively construct $y_{i+1}^{(1)}$ from $y_i^{(1)}$ by replacing a non-group element uniformly at random with a

group element. Finally, we construct $y_i^{(2)}$ from $y_i^{(1)}$ by replacing all elements from group $g$ with their counterpart from $g'$. By construction, all marginals are uniformly supported on the support of their corresponding distributions, i.e., we have a valid coupling.

Now, let $d(y, y')$ be the induced distance under *insertion/removal* between multiset-batches $y$ and $y'$, i.e., the number of insertions or removals needed to construct one from the other. By definition, we have for the entire support of the coupling:

$$d(y_i^{(1)} - y_j^{(1)}) \leq 2 \cdot |i - j| \qquad \forall i, j : 1 \leq i \leq \lambda, 1 \leq u \leq \lambda,$$
$$d(y_i^{(2)} - y_j^{(2)}) \leq 2 \cdot |i - j| \qquad \forall i, j : 1 \leq i \leq \lambda, 1 \leq u \leq \lambda,$$
$$d(y_i^{(1)} - y_j^{(2)}) \leq 2 \cdot \max\{i, j\} \qquad \forall i, j : 1 \leq i \leq \lambda, 1 \leq u \leq \lambda.$$

This is because $y_j^{(1)}$ with $j \geq i$ is iteratively constructed from $y_1^{(1)}$ via $j - i$ substitutions, i.e., $2 \cdot |i - j|$ insertions/removals. Furthermore, $y_j^{(2)}$ with $j \geq i$ is constructed from $y_i^{(1)}||_2$ by substituting $i$ elements from group $i$, inserting their $i$ counterparts from group $g'$, and then substituting an additional $j - i$ elements for a total of $(j - i) + i = j$ substitutions. The case $j \leq i$ is analogous.

The result then immediately follows from considering worst-case datasets (recall Lemma D.6) fulfilling these distance constraints, and the fact that base mechanism $B : \mathbb{Y} \to \mathbb{R}^D$ is a Gaussian mechanism with covariance $\sigma^2 \Delta_2 I$ whose underlying function $f : \mathbb{Y} \to \mathbb{R}^D$ has $\ell_2$ sensitivity $\Delta_2$ under insertion/removal. $\square$

Maybe somewhat surprisingly, the bound is identical to sampling with replacement under a single substition (see Theorem L.3 in (Schuchardt et al., 2024)), except for a change of Binomial distribution parameter $r$ from $\frac{1}{T}$ to $\frac{k}{T}$.

Next, we can solve a relaxed form of the optimization problem in Lemma E.6 to obtain our pessimistic upper bound for arbitrary $\lambda$.

**Theorem E.1.** *Consider number of sampled subsequences $\lambda \geq 1$, and let $r = \frac{L_C + L_F}{L - L_F + 1}$ be the probability of sampling a subsequence containing any specific element. Define $\overline{P}(\lambda) = MoG(-1 \cdot \mu, p, \sigma)$ and $\overline{Q}(\lambda) = MoG(\mu, p, \sigma)$ with means $\mu = \begin{bmatrix} 0 & 2 & 4 & \cdots & 2 \cdot \lambda \end{bmatrix}^T$ and weights $p \in [0, 1]^{\lambda+1}$ with $p_i = \text{Binomial}(i - 1 \mid \lambda, r)$. Further define per-epoch privacy profile $H(\alpha) = \sup_{x \simeq_E x'} H_\alpha(M_x || M_{x'})$. Then,*

$$H(\alpha) \leq H_\alpha(\overline{P}(\lambda) || \overline{Q}(\lambda))$$

*Proof.* As discussed in the previous two, the privacy of $M$ can be upper-bounded by analyzing group privacy under substitution using sampling with replacement. Specifically, we can instantiate Lemma E.6 with sensitivity $\Delta_2$ equal to clipping constant $C$, group size $k = L_C + L_F$ and dataset size $T = L - L_F + 1$. Due to translation equivariance of hockey stick divergences between Gaussians, we can assume w.l.o.g. that $\mu_1^{(1)} = \mu_1^{(2)} = 0$. We can further discard $||\mu_i^{(1)} - \mu_j^{(2)}||_2 \leq 2 \cdot \max\{i, j\}$ to obtain a relaxed optimization problem.

The optimal solution to the relaxed optimization problem is known from (Schuchardt et al., 2024) (see Lemma D.7) and corresponds exactly to our result. $\square$

For the special case of $\lambda = 1$, we can solve the optimization problem exactly to obtain a tight upper bound (we will later prove tightness, i.e., the $\geq$ part of Theorem 4.2, by constructing a pessimistic lower bound in the next section):

**Lemma E.7.** *Consider number of sampled subsequences $\lambda = 1$, and let $r = \frac{L_C + L_F}{L - L_F + 1}$ be the probability of sampling a subsequence containing any specific element. Define $P(1) = MoG(\mu, p, \sigma)$ with means $\mu = \begin{bmatrix} 0 & 2 \end{bmatrix}^T$ and weights $p = \begin{bmatrix} 1 - r & r \end{bmatrix}^T$. Further define per-epoch privacy profile $H(\alpha) = \sup_{x \simeq_E x'} H_\alpha(M_x || M_{x'})$. Then,*

$$H(\alpha) \leq \begin{cases} H_\alpha(P(1) || \mathcal{N}(0, \sigma)) & \text{if } \alpha \geq 0, \\ H_\alpha(\mathcal{N}(0, \sigma) || P(1)) & \text{if } 0 \leq \alpha < 1. \end{cases}$$

*Proof.* We begin with the case $\alpha \geq 1$. As before, we can instantiate Lemma E.6 with sensitvity $\Delta_2$ equal to clipping constant $C$, group size $k = L_C + L_F$ and dataset size $T = L - L_F + 1$. Due to translation equivariance of hockey stick divergences between Gaussians, we can assume w.l.o.g. that $\mu_2^{(1)} = 0$.

Thus, our optimization problem becomes

$$\max_{\boldsymbol{\mu}^{(1)}, \boldsymbol{\mu}^{(2)}} H_\alpha \left( (1-r) \cdot \mathcal{N}(\boldsymbol{\mu}_1^{(1)}, \sigma^2 \boldsymbol{I}) + r \cdot \mathcal{N}(\boldsymbol{\mu}_2^{(1)}, \sigma^2 \boldsymbol{I})) \| (1-r) \cdot \mathcal{N}(\boldsymbol{\mu}_1^{(2)}, \sigma^2 \boldsymbol{I}) + r \cdot \mathcal{N}(\boldsymbol{\mu}_2^{(2}, \sigma^2 \boldsymbol{I}) \right)$$

subject to $\boldsymbol{\mu}_2^{(1)} = 0$, $\boldsymbol{\mu}_1^{(1)} = \boldsymbol{\mu}_1^{(2)}$, $\|\boldsymbol{\mu}_1^{(2)} - \boldsymbol{\mu}_2^{(2)}\| \leq 2$, and $\|\boldsymbol{\mu}_2^{(1)} - \boldsymbol{\mu}_2^{(2)}\| \leq 2$.

Since the first two mixture components are identical and have identical weights, we can apply the advanced joint convexity property of hockey stick divergences (Balle et al. (2018), see Lemma D.2) to eliminate $\boldsymbol{\mu}_1^{(1)}$ and rewrite our objective as

$$\max_{\boldsymbol{\mu}^{(1)}, \boldsymbol{\mu}^{(2)}} r \cdot H_{\alpha'} \left( \mathcal{N}(\boldsymbol{\mu}_1^{(1)}, \sigma^2 \boldsymbol{I})) \| (1 - \beta(\alpha)) \cdot \mathcal{N}(\boldsymbol{\mu}_1^{(2)}, \sigma^2 \boldsymbol{I}) + \beta(\alpha) \cdot \mathcal{N}(\boldsymbol{\mu}_2^{(2}, \sigma^2 \boldsymbol{I}) \right)$$

with some $\alpha' \geq \alpha$ and $\beta(\alpha) \in [0, 1]$. Since we eliminated one variable, we are left with constraints $\boldsymbol{\mu}_2^{(1)} = 0$, $\|\boldsymbol{\mu}_1^{(2)} - \boldsymbol{\mu}_2^{(2)}\| \leq 2$, and $\|\boldsymbol{\mu}_2^{(1)} - \boldsymbol{\mu}_2^{(2)}\| \leq 2$. Since we now have only distance constraints to the origin, we can apply Lemma D.7 to arrive at the optimal value

$$r \cdot H_{\alpha'} \left( \mathcal{N}(0, \sigma) \right) \| (1 - \beta(\alpha)) \cdot \mathcal{N}(2, \sigma) + \beta(\alpha) \cdot \mathcal{N}(2, \sigma))$$
$$= H_\alpha \left( (1 - r) \cdot \mathcal{N}(2, \sigma) + r \cdot \mathcal{N}(0, \sigma)) \| \mathcal{N}(2, \sigma) \right)$$

where the equality follows from reverse application of the advanced joint convexity property, and the fact that the two components of the second distribution are identical. The result for $\alpha \geq 1$ then follows from rotating and translating the coordinate system such that the second distribution has its mean at the origin.

For the case $0 \leq \alpha < 1$, we can use the following fact: If $P, Q$ is dominating for $\alpha \geq 1$ under a symmetric neighboring relation, then $Q, P$ is dominating for $0 \leq \alpha < 1$ (Zhu et al. (2022), see Lemma D.10). $\qquad\square$

### E.1.4. OPTIMISTIC LOWER BOUNDS

Next, we construct optimistic lower bounds by constructing a worst-case gradient function and a pair of datasets for each $\alpha \geq 0$.

**Theorem 4.3.** *Consider the number of subsequences $\lambda \geq 1$, and let $r = \frac{L_C + L_F}{L - L_F + 1}$. Define $\underline{P}(\lambda) = MoG(\boldsymbol{\mu}, \boldsymbol{p}, \sigma)$ with means $\boldsymbol{\mu} \in \mathbb{N}_0^{\lambda+1}$ and weights $\boldsymbol{p} \in [0,1]^{\lambda+1}$ with $\mu_i = 2(i-1)$ and $p_i = \text{Binomial}(i \mid \lambda, r)$. Further, define per-epoch privacy profile $H(\alpha) = \sup_{x \simeq_E^1 x'} H_\alpha(M_x \| M_{x'})$. Then,*

$$H(\alpha) \geq \begin{cases} H_\alpha(\underline{P}(\lambda) \| \mathcal{N}(0, \sigma)) & \text{if } \alpha \geq 1, \\ H_\alpha(\mathcal{N}(0, \sigma) \| \underline{P}(\lambda)) & \text{if } 0 \leq \alpha < 1. \end{cases}$$

*Proof.* Since our model $f_\theta$ is an arbitrary parametric function, we can assume that it is the anti-derivative of any desired gradient function $g : \mathbb{R}^{L_C + L_F} \to \mathbb{R}^D$. We choose the following function:

$$g(a) = \begin{cases} C \cdot \boldsymbol{e}_1 & \text{if } \exists l \in \{1, \ldots, L_C + L_F\} : a_l = 1, \\ -C & \boldsymbol{e}_1 \text{otherwise.} \end{cases}$$

where $\boldsymbol{e}_1$ is the indicator vector that is non-zero in its first component, and $C$ is the clipping constant (i.e., our per-sequence gradients will never be affected by clipping).

**Case 1 ($\alpha \geq 1$):** Consider sequences $x_1, x_1' \in \mathbb{R}^L$ with $x_1 = \begin{bmatrix} 1 & 0 & \cdots & 0 \end{bmatrix}$ and $x_1' = \begin{bmatrix} 0 & 0 & \cdots & 0 \end{bmatrix}$ that differ in their first element. Note that the first element can appear in $L_C + L_F$ different subsequences in different positions because Algorithm 4 zero-pads the sequence before sampling with replacement. Further consider sequences $x_2, \ldots, x_N \in \mathbb{R}^L \setminus \{\mathbf{1}\}$ such that $\forall m > n > 1 : x_m \neq x_n \wedge x_1 \neq x_n \neq x_1'$ (so that our dataset is a proper set, i.e., does not have duplicates). Define datasets $x = \{x_1, x_2, \ldots, x_N\}$ and $x' = \{x_1', x_2, \ldots, x_N\}$.

By construction, $x \simeq_E^1 x'$. For every sequence in a top-level batch, we sample $\lambda$ subsequences with replacement, there are $L - L_F + 1$ subsequences in total, and there are exactly $L_C + L_F$ subsequences in $x_1$ that contain 1. These $L_C + L_F$ subsequences will add $C$ to our summed gradient when sampled, while all other gradients subtract $C$. Due to translation

invariance of hockey stick divergences between Gaussian mixtures (i.e., we can translate $-C \cdot \lambda$ into the origin) and after marginalizing out all but the first dimension, we exactly attain our desired bound. The subsequent training steps operate on identical data and will thus not contribute to the hockey stick divergence attained by the epoch-level mechanism (same argument as in Lemma E.2).

**Case 2** ($0 \leq \alpha < 1$)**:** For this case, we define $x = \{x_1', x_2, \ldots, x_N\}$ and $x' = \{x_1, x_2, \ldots, x_N\}$, i.e., interchange the first sequence between $x$ and $x'$. The remaining proof is symmetric to the previous case. $\qquad \square$

Note that, for $\lambda = 1$, Theorem 4.3 coincides with Lemma E.7, i.e., our pessimistic bound is tight. Thus, this also concludes our proof of Theorem 4.2.

### E.1.5. DOMINATING PAIRS.

Next, let us discuss how to construct dominating pairs corresponding to our bounds.

In the case of our generic pessimistic bound for $\lambda \geq 1$ (see Theorem E.1), we simply have $\sup_{x \simeq_E^1 x'} H_\alpha(M_x || M_{x'}) \leq H_\alpha(\overline{P}(\lambda) || \overline{Q}(\lambda))$, where $\overline{P}(\lambda)$ and $\overline{Q}(\lambda)$ are univariate mixtures of Gaussians. Evidently, these two mixtures are a dominating pair.

In the case of our tight bound for $\lambda \geq 1$, we have

$$\sup_{x \simeq_E^1 x'} H_\alpha(M_x || M_{x'}) = \begin{cases} H_\alpha(P(1) || \mathcal{N}(0, \sigma)) & \text{if } \alpha \geq 1, \\ H_\alpha(\mathcal{N}(0, \sigma) || P(1)) & \text{if } 0 \leq \alpha < 1. \end{cases}$$

Due to equality, the r.h.s. term is a valid privacy profile by definition (cf. Lemma D.11). For the purpose of further analysis, one can construct a dominating pair via convex conjugation (Zhu et al., 2022):

**Lemma D.12.** *Let $H : \mathbb{R}_+ \to \mathbb{R}$ be a privacy profile. Then there exists a pair of univariate distributions $(P, Q)$ with $H(\alpha) = H_\alpha(P, Q)$. Specifically, distribution $P$ is supported on $[0, 1)$ and has CDF $1 + H * (x - 1)$, where $H^*$ is the convex conjugate of privacy profile $H$. Distribution $Q$ is $\mathrm{Uniform}([0, 1])$.*

For the purpose of numerical privacy accounting, we need to construct a quantized dominating pair. As is known from (Doroshenko et al., 2022), the best possible quantized dominating pair only requires access to a privacy profile:

**Lemma D.16.** *Consider any pair of distributions $(P, Q)$ and finite set of quantization thresholds $\mathcal{E} = \{\varepsilon_0, \ldots, \varepsilon_K\}$ with $\varepsilon_k \in \mathbb{R} \cup \{+\infty, -\infty\}$ and $-\infty = \varepsilon_0 < \varepsilon_1 < \cdots < \varepsilon_k = +\infty$. There exists a pair of distributions $(\hat{P}^\uparrow \hat{Q}^\uparrow)$ such that the distribution of privacy loss $\mathcal{L}_{\hat{P}^\uparrow, \hat{Q}^\uparrow}$ is only supported on $\mathcal{E}$ and $(\hat{P}^\uparrow \hat{Q}^\uparrow)$ dominates $(P, Q)$. Furthermore, $(\hat{P}^\uparrow \hat{Q}^\uparrow)$ is dominated by any other distribution supported on $\mathcal{E}$ and is uniquely defined by $\{(\varepsilon, H_{e^\varepsilon}(P, Q)) \mid \varepsilon \in \mathcal{E}\}$.*

Thus, we can algorithmically generate a dominating pair using the "connect-the-dots" (Doroshenko et al., 2022) method without explicitly needing to construct a non-quantized dominating pair. We use the latter approach for all our experiments.

### E.2. Poisson Sampling

---

**Algorithm 6** Bottom-Level Poisson Sampling

---

**Input:** Sequence $x_n$, context length $L_C$, forecast length $L_F$, expected subsequences $\lambda$
$x_n' \leftarrow \text{prepend\_zeros}(x_n, L_C)$  {padding}
$T \leftarrow L - L_F + 1$  {maximum start index}
$r \leftarrow \min\{1, \lambda / T\}$  {Rate to sample $\lambda$ in expectation.}
**for** $t \leftarrow 1$ **to** $T$ **do**
  **if** $\mathrm{Uniform}[0, 1] \leq r$ **then**
    **yield** $x_n'[t : t + L_C + L_F - 1]$  {cropping}

---

Our main objective for top-level deterministic iteration (see Algorithm 3) and bottom-level Poisson sampling (see Algorithm 6) will be proving the the following tight pessimistic bound for arbitrary $\lambda \in \mathbb{N}$:

**Theorem E.8.** *Consider an expected number of subsequences* $\lambda \in \mathbb{N}$ *and let* $r = \min\{1, \lambda / (L - L_F + 1)\}$ *be the resultant sampling rate from Algorithm 6. Then* $P(\lambda) = MoG(-1 \cdot \boldsymbol{\mu}, \boldsymbol{p}, \sigma)$ *and* $Q(\lambda) = MoG(\boldsymbol{\mu}, \boldsymbol{p}, \sigma)$ *with means* $\mu_i = (i - 1)$ *and weights* $p_i = \text{Binomial}(i - 1 \mid L_C + L_F, r)$ *are a tight dominating pair of epoch* $M$ *under* $\overset{1}{\simeq}_E$.

To this end, we can

1. Prove that, under top-level deterministic iteration, the privacy of our epoch-level mechanism can be upper-bounded by analyzing a training step for a single batch,
2. prove that the privacy of this training step can be upper-bounded by analyzing Poisson sampling under group substitution,
3. derive pessimistic upper bounds for sampling with replacement under group substitution,
4. determine optimistic lower bounds that coincide with the upper bound for all $\lambda \in \mathbb{N}$ by constructing worst-case time series datasets,
5. determine tight dominating pairs corresponding to our tight pessimistic bounds.

The first two steps are identical to the pervious section, since they do not depend on the distribution of the bottom-level subsampling procedure (see Appendices E.1.1 and E.1.2). For the third step, we can use the following known result from (Schuchardt et al., 2024) for group insertion/removal.

**Definition E.9.** Consider a dataset space $\mathbb{X} = \mathcal{P}(\mathbb{A})$ with underlying set $\mathbb{A}$. Two datasets $x, x' \in \mathbb{X}$ of size $T$ are $(k_+, k_-)$-group-insertion/removal neighboring $(x \simeq_{k_+, k_-, \pm} x')$ if there are groups $g_- \subseteq x$ and $g_+ \subseteq x'$ with $|g_-| = k_-$ and $|g_+| = k_+$ such that $x' = x \setminus g_- \cup g_+$.

**Lemma E.10** ((Schuchardt et al., 2024)). *Consider a dataset space* $\mathbb{X} = \mathcal{P}(\mathbb{A})$ *with underlying set* $\mathbb{A}$. *and batch space* $\mathbb{Y} = \mathcal{P}(\mathbb{A})$. *Let* $S : \mathbb{X} \to \mathbb{Y}$ *be Poisson sampling with rate* $r$ *Let base mechanism* $B : \mathbb{Y} \to \mathbb{R}^D$ *be a Gaussian mechanism* $B(y) = f + Z$ *with* $Z \sim \mathcal{N}(0, \Delta_2 \sigma^2 \boldsymbol{I})$, *where* $\Delta_2$ *is the* $\ell_2$-*sensitivity of underlying function* $f : \mathbb{Y} \to \mathbb{R}^D$ *under insertion/removal* $(\simeq_\pm)$. *Define subsampled mechanism* $M = B \circ S$. *Further define mixture distributions* $P(k) = MoG(-1 \cdot \boldsymbol{\mu}_-, \boldsymbol{p}_-, \sigma)$ *and* $Q(k) = MoG(\boldsymbol{\mu}_+, \boldsymbol{p}_+, \sigma)$ *with means* $\boldsymbol{\mu}_- = \begin{bmatrix} 0 & 1 & 2 & \cdots & k_- \end{bmatrix}^T$ *and* $\boldsymbol{\mu}_+ = \begin{bmatrix} 0 & 1 & 2 & \cdots & k_+ \end{bmatrix}^T$, *as well a weights weights* $\boldsymbol{p}_- \in [0, 1]^{k_- + 1}$ *with* $p_{-i} = \text{Binomial}(i - 1 \mid k_-, r)$. *and* $\boldsymbol{p}_+ \in [0, 1]^{k_+ + 1}$ *with* $p_{-i} = \text{Binomial}(i - 1 \mid k_+, r)$. *Then, for all* $\alpha \geq 0$,

$$\sup_{x \simeq_{k_+, k_-, \pm} x'} H_\alpha(M_x \| M_{x'}) \leq H_\alpha(P(k) \| Q(k)). \tag{6}$$

From our derivations in Appendices E.1.1 and E.1.2, the following result immediately follows via a seemingly naïve reduction from group substitution to group insertion/removal:

**Lemma E.11.** *Consider an expected number of subsequences* $\lambda \in \mathbb{N}$ *and let* $r = \min\{1, \lambda / (L - L_F + 1)\}$ *be the resultant sampling rate from Algorithm 6. Then* $P(\lambda) = MoG(-1 \cdot \boldsymbol{\mu}, \boldsymbol{p}, \sigma)$ *and* $Q(\lambda) = MoG(\boldsymbol{\mu}, \boldsymbol{p}, \sigma)$ *with means* $\mu_i = (i - 1)$ *and weights* $p_i = \text{Binomial}(i - 1 \mid L_C + L_F, r)$ *are a dominating pair of epoch* $M$ *under* $\overset{1}{\simeq}_E$, *i.e., for all* $\alpha \geq 0$:

$$\sup_{x \overset{1}{\simeq}_E x'} H_\alpha(M_x \| M_{x'}) \leq H_\alpha(P(\lambda) \| Q(\lambda)). \tag{7}$$

*Proof.* Any group-substitution with group size $k$ is equivalent to a group insertion of size $k$, followed by a group removal of size $k$. Due to this observation and Lemmas E.2 and E.3, we can instantiate Lemma E.10 with with sensitivity $\Delta_2$ equal to clipping constant $C$, and number of insertions/removals $k_+ = k_- = L_C + L_F$. $\square$

Perhaps somewhat surprisingly, our next step will show that this naïve reduction does in fact yield a tight dominating pair for our epoch-level mechanism.

### E.2.1. OPTIMISTIC LOWER BOUNDS

As with sampling with replacement, we construct optimistic lower bounds by constructing a worst-case gradient function and a pair of datasets for each $\alpha \geq 0$.

**Lemma E.12.** *Consider an expected number of subsequences* $\lambda \in \mathbb{N}$ *and let* $r = \min\{1, \lambda / (L - L_F + 1)\}$ *be the resultant sampling rate from Algorithm 6. Then* $P(\lambda) = MoG(-1 \cdot \boldsymbol{\mu}, \boldsymbol{p}, \sigma)$ *and* $Q(\lambda) = MoG(\boldsymbol{\mu}, \boldsymbol{p}, \sigma)$ *with means* $\mu_i = (i - 1)$

*and weights $p_i = \text{Binomial}(i - 1 \mid L_C + L_F, r)$ fulfill for all $\alpha \geq 0$:*

$$\sup_{x \overset{1}{\simeq}_E x'} H_\alpha(M_x \| M_{x'}) \geq H_\alpha(P(\lambda) \| Q(\lambda)). \tag{8}$$

*Proof.* Since our model $f_\theta$ is an arbitrary parametric function, we can assume that it is the anti-derivative of any desired gradient function $g : \mathbb{R}^{L_C + L_F} \to \mathbb{R}^D$. We choose the following function:

$$g(a) = \begin{cases} C \cdot \boldsymbol{e}_1 & \text{if } \exists l \in \{1, \ldots, L_C + L_F\} : a_l = 1, \\ -C \cdot \boldsymbol{e}_1 & \text{else if } \exists l \in \{1, \ldots, L_C + L_F\} : a_l = -1, \\ 0 & \text{otherwise.} \end{cases}$$

where $\boldsymbol{e}_1$ is the indicator vector that is non-zero in its first component, and $C$ is the clipping constant (i.e., our per-sequence gradients will never be affected by clipping).

Next, consider sequences $x_1, x_1' \in \mathbb{R}^L$ with $x_1 = \begin{bmatrix} 1 & 0 & \cdots & 0 \end{bmatrix}$ and $x_1' = \begin{bmatrix} -1 & 0 & \cdots & 0 \end{bmatrix}$ that differ in their first element. Note that the first element can appear in $L_C + L_F$ different subsequences in different positions because Algorithm 4 zero-pads the sequence before Poisson sampling. Further consider sequences $x_2, \ldots, x_N$ such that $\forall m > n > 1 : x_m \neq x_n \wedge x_1 \neq x_n \neq x_1'$ (so that our dataset is a proper set, i.e., does not have duplicates) and $\forall n : 1 \notin x_n \wedge -1 \notin x_n$ (so that the gradients for all subsequences are always zero). Define datasets $x = \{x_1, x_2, \ldots, x_N\}$ and $x' = \{x_1', x_2, \ldots, x_N\}$.

By construction, $x \overset{1}{\simeq}_E x'$. For every sequence in a top-level batch, we Poisson sample up to $L + L_F - 1$ subsequences, there are $L - L_F + 1$ subsequences in total. There are exactly $L_C + L_F$ subsequences in $x_1$ that contain 1 and $L_C + L_F$ subsequences in $x_1'$ that contain $-1$. These $L_C + L_F$ subsequences will, respectively, add $+C$ and $-C$ to our summed gradient when sampled, while all other gradients are 0. Thus, we exactly attain our desired bound in the first training step. The subsequent training steps operate on identical data and will thus not contribute to the hockey stick divergence attained by the epoch-level mechanism (same argument as in Lemma E.2). $\square$

From the fact that our upper bound (Lemma E.11) and our lower bound (Lemma E.12) coincide, the tightness of our dominating pairs (Theorem E.8) immediately follows.

**Other contributions.** The fact that we have been able to derive tight guarantees for what is essentially group substitution using the sensitivity of the base mechanism w.r.t. insertion/removal showcases the benefit of analyzing hybrid neighboring relations (Balle et al., 2018). Furthermore, it showcases that certain subsampling schemes are more compatible with certain neighboring relations (Lebeda et al., 2024) — even in the group privacy setting. [3]

### E.3. Optimal Number of Subsequences

Finally, we can prove that $\lambda = 1$ minimizes the privacy profile for all $\alpha$ under both sampling with replacement and Poisson sampling. To this end, recall the following fact about stochastic dominance between Binomial distributions (see, e.g., Example 4.2.4. from (Roch, 2024)):

**Lemma E.13.** *Consider distributions $D_{n,p} = \text{Binomial}(N, p)$ and $D_{M,q} = \text{Binomial}(m, q)$ with $N \geq M$ and $p \geq q$. Then $D_{N,p}$ stochastically dominates $D_{M,q}$, i.e, there exists a monotonic coupling $\Pi$ of $D_{N,p}$ and $D_{M,q}$, with mass function $\pi : \mathbb{N}_0^2 \to [0, 1]$ such that $\pi(k, i) > 0 \iff k \geq i$.*

In other words, we construct $D_{N,p}$ from $D_{M,q}$ by shifting probability mass towards larger values.

In addition to stochastic dominance, we will make use of the following monotonicity result for hockey stick divergences between pairs of Gaussian mixtures (special case of Theorem O.6 from (Schuchardt et al., 2024) for univariate mixtures):

**Lemma E.14.** *Consider any pair of Gaussian mixtures $P = MoG(-\boldsymbol{\mu}, \boldsymbol{p}, \sigma)$ and $Q = MoG(\boldsymbol{\nu}, \boldsymbol{q}, \sigma)$ with non-negative means $\boldsymbol{\mu} \in \mathbb{R}_+^N, \boldsymbol{\nu} \in \mathbb{R}_+^M$ and weights $\boldsymbol{p} \in [0, 1]^N, \boldsymbol{q} \in [0, 1]^M$. Then, the (sub-)derivative of $H_\alpha(P \| Q)$ w.r.t. any $\mu_i$ or $\nu_j$ is non-negative.*

In other words: Moving mixture components farther from the origin (integrating over the non-negative (sub-)derivatives) only ever increases divergence.

---

[3]Trying to use the sensitivity w.r.t. substitution would induce much more complicated distance constraints after coupling.

Let us now begin with proving optimality for Poisson sampling:

**Lemma E.15.** *Let $P^*(\lambda)$, $Q^*(\lambda)$ be a tight dominating pair of epoch $M$ for bottom-level Poisson sampling and $\lambda \in \mathbb{N}$ expected subsequences. Then $H_\alpha(P^*(\lambda), Q^*(\lambda))$ is minimized by $\lambda = 1$ for all $\alpha \geq 0$.*

*Proof.* By definition, all tight dominating pairs exactly match the privacy profile of mechanism $M$, i.e., attain identical hockey stick divergence for all $\alpha \geq 0$. We can thus focus our discussion on the tight dominating pairs from Theorem E.8, which are mixtures of Gaussians.

Consider an arbitrary $\lambda \geq 1$ and let $P(\lambda) = MoG(-1 \cdot \boldsymbol{\mu}, \boldsymbol{p}, \sigma)$ and $Q(\lambda) = MoG(\boldsymbol{\mu}, \boldsymbol{p}, \sigma)$ with means $\mu_i = (i-1)$ and weights $p_i = \text{Binomial}(i-1 \mid L_C + L_F, r(\lambda))$ and sampling rate $r(\lambda) = \min\{1, \lambda / (L - L_F + 1)\}$.

Since $r(\lambda)$ is increasing in $\lambda$, we know from Lemma E.13 that $\text{Binomial}(L_C + L_F, r(\lambda))$ stochastically dominates $\text{Binomial}(L_C + L_F, r(1))$. We can thus use a monotonic coupling of the two weight distributions with coupling mass function $\pi : \mathbb{N}_0^2 \to [0, 1]$ to restate $P(1), Q(1)$ as follows:

$$P(1) = \sum_{i=0}^{L_C+L_F} \sum_{k=0}^{L_C+L_F} \pi(k, i) \cdot \mathcal{N}(-1 \cdot i, \sigma^2 \boldsymbol{I}), \qquad Q(1) = \sum_{i=0}^{L_C+L_F} \sum_{k=0}^{L_C+L_F} \pi(k, i) \cdot \mathcal{N}(i, \sigma^2 \boldsymbol{I}),$$

i.e., we split up the $i$th (with zero-based indexing) mixture component into $L_C + L_F - i + 1$ mixture components with identical means. Similarly, we can restate $P(\lambda), Q(\lambda)$ as

$$P(\lambda) = \sum_{i=0}^{L_C+L_F} \sum_{k=0}^{L_C+L_F} \pi(k, i) \cdot \mathcal{N}(-1 \cdot k, \sigma^2 \boldsymbol{I}), \qquad Q(\lambda) = \sum_{i=0}^{L_C+L_F} \sum_{k=0}^{L_C+L_F} \pi(k, i) \cdot \mathcal{N}(k, \sigma^2 \boldsymbol{I}).$$

Since $\pi(k, i) \geq 0 \iff k \geq i$ (Lemma E.13), and divergence increases when increasing the norm of the mixture means (Lemma E.14), we know that $H_\alpha(P(\lambda) \| Q(\lambda)) \geq H_\alpha(P(1) \| Q(1))$. $\qquad \square$

Next, let us consider sampling with replacement. The proof is slightly more involved, since we only have optimistic lower bounds for $\lambda > 1$, as opposed to an exact characterization of the privacy profile.

**Lemma E.16.** *Let $P^*(\lambda)$, $Q^*(\lambda)$ be a tight dominating pair of epoch $M$ for bottom-level sampling with replacement and $\lambda \in \mathbb{N}$ expected subsequences. Then $H_\alpha(P^*(\lambda), Q^*(\lambda))$ is minimized by $\lambda = 1$ for all $\alpha \geq 0$.*

*Proof.* **Case 1** ($\alpha \geq 1$): Consider any tight dominating pair $P^*(\lambda), Q^*(\lambda)$ for $\lambda \geq 1$, as well as a tight dominating pair $P^*(1), Q^*(1)$ for $\lambda = 1$. We know from Theorem 4.3 and the definition of tight dominating pairs that

$$H_\alpha(P^*(\lambda) \| Q^*(\lambda)) \geq H_\alpha(\overline{P}(\lambda) \| N(0, \sigma)),$$

while we know from Theorem 4.2 that

$$H_\alpha(\overline{P}(1) \| \mathcal{N}(0, \sigma)) = H_\alpha(P^*(1) \| Q^*(1)),$$

with $\overline{P}(\lambda) = MoG(\boldsymbol{\mu}, \boldsymbol{p}, \sigma)$, $\boldsymbol{\mu} \in \mathbb{N}_0^{\lambda+1}$ and weights $\boldsymbol{p} \in [0, 1]0^{\lambda+1}$ with $\mu_i = 2(i-1)$ and $p_i = \text{Binomial}(i \mid \lambda, r)$, where $r$ is constant in $\lambda$.

We can thus prove $H_\alpha(P^*(\lambda) \| Q^*(\lambda)) \geq H_\alpha(P^*(1) \| Q^*(1))$ by proving $H_\alpha(\overline{P}(\lambda) \| \mathcal{N}(0, \sigma)) \geq H_\alpha(\overline{P}(1) \| \mathcal{N}(0, \sigma))$.

To this end, recall from Lemma E.13 that $\text{Binomial}(\lambda, r)$ stochastically dominates $\text{Binomial}(1, r)$. We can thus apply exactly the same monotonic-coupling-based proof as with Lemma E.15.

**Case 2** ($0 \leq \alpha < 1$): This case is fully analogous, except that we need to use monotonic couplings to prove $H_\alpha(\overline{P}(\mathcal{N}(0, \sigma) \| \lambda)) \geq H_\alpha(\mathcal{N}(0, \sigma) \| \overline{P}(1))$ $\qquad \square$

# F. Proofs from Section 4.2 (Bi-Level Sampling)

The main purpose of this section is to prove the following tight pessimistic guarantee for top-level sampling without replacement and bottom-level sampling with replacement:

**Theorem 4.4.** *Consider the number of subsequences $\lambda = 1$ and batch size $\Lambda$. Let $r = \frac{L_C + L_F}{L - L_F + 1}$ and let $\rho = \lfloor \Lambda / \lambda \rfloor / N$ be the probability of sampling any specific sequence. Define $\tilde{P}(1) = MoG(\boldsymbol{\mu}, \tilde{\boldsymbol{p}}, \sigma)$ with $\tilde{\boldsymbol{\mu}} = \begin{bmatrix} 0 & 2 \end{bmatrix}^T$ and $\tilde{\boldsymbol{p}} = \begin{bmatrix} (1 - \rho) + \rho \cdot (1 - r) & \rho \cdot r \end{bmatrix}^T$. Then, per-step privacy profile $\tilde{H}(\alpha) = \sup_{x \simeq_E^1 x'} H_\alpha(\tilde{M}_x || \tilde{M}_{x'})$ fulfills*

$$\tilde{H}(\alpha) = \begin{cases} H_\alpha(\tilde{P}(1) || \mathcal{N}(0, \sigma)) & \text{if } \alpha \geq 1, \\ H_\alpha(\mathcal{N}(0, \sigma) || \tilde{P}(1)) & \text{if } 0 \leq \alpha < 1. \end{cases}$$

We further want to prove the following (not necessarily tight) pessimistic guarantees:

**Theorem F.1.** *Consider bottom-level sampling with replacement, number of subsequences $\lambda = 1$ and batch size $\Lambda$. Let $r = \frac{L_C + L_F}{L - L_F + 1}$ and let $\rho = \lfloor \Lambda / \lambda \rfloor / N$ be the probability of sampling any specific sequence. Like in Theorem E.1, define $\overline{P}(\lambda) = MoG(-1 \cdot \boldsymbol{\mu}, \boldsymbol{p}, \sigma)$ and $\overline{Q}(\lambda) = MoG(\boldsymbol{\mu}, \boldsymbol{p}, \sigma)$ with means $\boldsymbol{\mu} = \begin{bmatrix} 0 & 2 & 4 & \cdots & 2 \cdot \lambda \end{bmatrix}^T$ and weights $\boldsymbol{p} \in [0, 1]^{\lambda + 1}$ with $p_i = \text{Binomial}(i - 1 \mid \lambda, r)$. Then, per-step privacy profile $\tilde{H}(\alpha) = \sup_{x \simeq_E^1 x'} H_\alpha(\tilde{M}_x || \tilde{M}_{x'})$ fulfills*

$$\tilde{H}(\alpha) \leq (1 - \rho) \cdot \max\{0, 1 - \alpha\} + \rho \cdot H_\alpha(\overline{P}(\lambda) || \overline{Q}(\lambda))$$

**Theorem F.2.** *Consider bottom-level Poisson sampling, number of subsequences $\lambda = 1$ and batch size $\Lambda$. Let $r = \min\{1, \lambda / (L - L_F + 1)\}$ be the resulting Poisson sampling rate, and let $\rho = \lfloor \Lambda / \lambda \rfloor / N$ be the probability of sampling any specific sequence. Like in Theorem E.8, define $P(\lambda) = MoG(-1 \cdot \boldsymbol{\mu}, \boldsymbol{p}, \sigma)$ and $Q(\lambda) = MoG(\boldsymbol{\mu}, \boldsymbol{p}, \sigma)$ with means $\mu_i = (i - 1)$ and weights $p_i = \text{Binomial}(i - 1 \mid L_C + L_F, r)$. Then, per-step privacy profile $\tilde{H}(\alpha) = \sup_{x \simeq_E^1 x'} H_\alpha(\tilde{M}_x || \tilde{M}_{x'})$ fulfills*

$$\tilde{H}(\alpha) \leq (1 - \rho) \cdot \max\{0, 1 - \alpha\} + \rho \cdot H_\alpha(P(\lambda) || Q(\lambda)).$$

We further want to determine pessimistic lower bounds to serve as baselines for $\lambda = 1$ in our numerical experiments.

To derive these results, we proceed similar to Appendix E. Specifically, we:

1. Prove that, the privacy of our bi-level mechanism can be bounded by considering a fixed set of top-level batches,
2. from this result, derive pessimistic upper bounds via joint convexity,
3. derive tighter upper bounds for $\lambda = 1$ and bottom-level sampling with replacement by focusing on our analysis on a single per-subsequence gradient via the parallel composition property,
4. determine optimistic lower bounds that coincide with the upper bound for $\lambda = 1$ and bottom-level sampling with replacement by constructing worst-case time series datasets,
5. determine dominating pairs corresponding to our pessimistic upper bounds.

## F.1. Reduction to Fixed Set of Top-Level Batches

In the following, we use conditional couplings to eliminate the randomness inherent to top-level sampling from our analysis. The proof is largely identical to that for sampling without replacement and substitution in sets (e.g., Theorem 11.2 in (Zhu et al., 2022)), except that we sometimes only substitute with sequences that differ in a single element (1-event-level privacy).

**Lemma F.3.** *Consider the space $\mathbb{A} = \mathbb{R}^L$ of all length-$L$ sequences. Let $\mathbb{X} \subseteq \mathcal{P}(\mathbb{A})$ be the space of all size-$N$ datasets of sequences. Further consider batch size $\Lambda$ and bottom-level number of instances $\lambda$ with resultant top-level batch size $N' = \lfloor \Lambda / \lambda \rfloor$. Let $\mathbb{Y} = \mathcal{P}(\mathbb{A})$ be the space of all size-$N'$ top-level batches. Let $\hat{S} : \mathbb{X} \to \mathbb{Y}$ be top-level sampling without replacement, as defined in Algorithm 5, and $\rho = \lfloor \Lambda / \lambda \rfloor / N$ be the probability of sampling any specific sequence. Finally, let $\hat{B} : \mathbb{Y} \to \mathbb{R}^D$ be an arbitrary mechanism that maps top-level batches to gradients and define $\tilde{M} = \hat{B} \circ \hat{S}$. Then, for all $\alpha \geq 0$,*

$$\sup_{x \simeq_E^1 x'} H_\alpha(\tilde{M}_x || \tilde{M}_{x'}) \leq \sup_{y, y', y'' \in \mathbb{Y}} H_\alpha\left((1 - \rho) \cdot \hat{B}_y + \rho \cdot \hat{B}_{y'} || (1 - \rho) \cdot \hat{B}_y + \rho \cdot \hat{B}_{y''}\right)$$

*subject to $y \simeq_\Delta y'$, $y \simeq_\Delta y''$, $y' \simeq_E^1 y''$, where $\simeq_\Delta$ indicates arbitrary substitution of a single sequence, while $\simeq_E^1$ indicates substitution while only changing a single element of the sequence.*

*Proof.* Consider arbitrary $x \overset{1}{\simeq}_E x'$. By definition, there must be some $x_n \in x$ and $x'_n \in x'$ such that $x \setminus \{x'_n\} = x' \setminus \{x'_n\}$ and sequences $x_n$ differ in exactly one sequence.

Define events $A_0 = \{y \subseteq \mid x_n \notin x\}$, $A_1 = \{y \subseteq \mid x_n \in x\}$, Define events $E_0 = \{y \subseteq \mid x'_n \notin x'\}$, $E_1 = \{y \subseteq \mid x'_n \in x'\}$. We naturally have $\hat{S}_x(A_0) = \hat{S}_{x'}(E_0) = 1 - \rho$ and $\hat{S}_x(A_1) = \hat{S}_{x'}(E_1) = \rho$.

We can then define a coupling $\Gamma$ of $\hat{S}_x(\cdot \mid A_0), \hat{S}_x(\cdot \mid A_1), \hat{S}_{x'}(\cdot \mid E_0), \hat{S}_{x'}(\cdot \mid E_0)$ via the following mass function:

$$\gamma(y_0^{(1)}, y_1^{(1)}, y_0^{(2)}, y_1^{(2)})$$

$$\propto s_x(y_0^{(1)} \mid A_0) \cdot \mathbb{1}\left[\exists x_m \in y_0^{(1)} : y_1^{(1)} = y_0^{(1)} \setminus \{x_m\} \cup \{x_n\}\right] \cdot \prod_{i=0}^{1} \mathbb{1}\left[y_i^{(2)} = y_i^{(1)} \setminus \{x_n\} \cup \{x'_n\}\right].$$

By construction, the entire support of our couplings fulfills $y_0^{(1)} = y_0^{(2)}$, $y_0^{(1)} \simeq_\Delta y_1^{(1)}$, $y_0^{(1)} \simeq_\Delta y_1^{(2)}$, and $y_1^{(1)} \overset{1}{\simeq}_E y_1^{(2)}$.

The result then immediately follows from considering worst-case datasets given these constraints (Lemma D.6). □

The distinction between the two different types of substitution is critically important for our later analysis of amplification-by-augmentation, as we may otherwise drastically underestimate the difference of specific Gaussian distributions.

## F.2. Pessimistic Upper Bounds via Joint Convexity

Via the next lemma, we can immediately prove our (not necessarily tight) pessimistic upper bounds for top-level sampling without replacement and bottom-level Poisson sampling.

**Lemma F.4.** *Consider the space $\mathbb{A} = \mathbb{R}^L$ of all length-$L$ sequences. Let $\mathbb{X} \subseteq \mathcal{P}(\mathbb{A})$ be the space of all size-$N$ datasets of sequences. Further consider batch size $\Lambda$ and bottom-level number of instances $\lambda$ with resultant top-level batch size $N' = \lfloor \Lambda / \lambda \rfloor$. Let $\mathbb{Y} = \mathcal{P}(\mathbb{A})$ be the space of all size-$N'$ top-level batches. Let $\hat{S} : \mathbb{X} \to \mathbb{Y}$ be top-level sampling without replacement, as defined in Algorithm 5, and $\rho = \lfloor \Lambda / \lambda \rfloor / N$ be the probability of sampling any specific sequence. Finally, let $\hat{B} : \mathbb{Y} \to \mathbb{R}^D$ be an arbitrary mechanism that maps top-level batches to gradients and define $\tilde{M} = \hat{B} \circ \hat{S}$. Then, for all $\alpha \geq 0$,*

$$\sup_{x \overset{1}{\simeq}_E x'} H_\alpha(\tilde{M}_x || \tilde{M}_{x'}) \leq (1 - \rho) \cdot \max\{0, 1 - \alpha\} + \rho \cdot \sup_{y', y''} H_\alpha\left(\hat{B}_{y'} || \hat{B}_{y''}\right)$$

*subject to $y' \overset{1}{\simeq}_E y''$.*

*Proof.* The result immediately follows from Lemma F.3 and joint convexity of hockey stick divergences ((Balle et al., 2020)), i.e.,

$$H_\alpha\left((1 - \rho) \cdot \hat{B}_y + \rho \cdot \hat{B}_{y'} || (1 - \rho) \cdot \hat{B}_y + \rho \cdot \hat{B}_{y''}\right)$$

$$\leq (1 - \rho) \cdot H_\alpha(\hat{B}_y || \hat{B}_y) + \rho \cdot H_\alpha\left(\hat{B}_{y'} || \hat{B}_{y''}\right)$$

$$= (1 - \rho) \cdot \max\{0, 1 - \alpha\} + \rho \cdot H_\alpha\left(\hat{B}_{y'} || \hat{B}_{y''}\right).$$

□

Analyzing the privacy of bottom-level subsampling when two top-level batches differ in one element of one sequence, i.e., bounding $\sup_{y', y''} H_\alpha\left(\hat{B}_{y'} || \hat{B}_{y''}\right)$ is exactly what we have already done in Appendix E. Specifically Theorem F.1 follows directly from Lemma F.4 and Theorem E.1, while Theorem F.2 follows directly from Lemma F.4 and Theorem E.8.

## F.3. Tighter Upper Bounds for Sampling With Replacement

Instead of directly applying joint convexity, we can derive tighter bounds on the following optimization problem from Lemma F.3 via conditional coupling:

$$\sup_{y, y', y'' \in \mathbb{Y}} H_\alpha\left((1 - \rho) \cdot \hat{B}_y + \rho \cdot \hat{B}_{y'} || (1 - \rho) \cdot \hat{B}_y + \rho \cdot \hat{B}_{y''}\right) \tag{9}$$

Due to permutation invariance of bottom-level subsampling and gradient computation, we can define an arbitrary ordering within the top-level batches such that $y = \{y_1, y_2, \ldots, y_{N'}\}$, $y' = \{y_1', y_2, \ldots, y_{N'}\}$, $y'' = \{y_1'', y_2, \ldots, y_{N'}\}$ differ in their first sequence, and $y_1'$ and $y_1''$ only differ in a single element. We observe that the distribution over gradients for $y_2, \ldots, y_{N'}$ are identical under both mixture mechanisms in Eq. (9) and are computed (before summation) completely independently of $y_1, y_1', y_1''$. We can thus make precisely the same post-processing and parallel composition argument as in Lemma E.3 from Appendix E.1.2 to show that

$$\sup_{y, y', y'' \in \mathbb{Y}} H_\alpha \left( (1 - \rho) \cdot \hat{B}_y + \rho \cdot \hat{B}_{y'} \| (1 - \rho) \cdot \hat{B}_y + \rho \cdot \check{M}_{y''} \right) \tag{10}$$

$$\leq \sup_{y_1, y_1', y_1'' \in \mathbb{Y}} H_\alpha \left( (1 - \rho) \cdot \check{M}_{y_1} + \rho \cdot \check{M}_{y_1'} \| (1 - \rho) \cdot \check{M}_{y_1} + \rho \cdot \check{M}_{y_1''} \right), \tag{11}$$

with $\{y_1\} \simeq_\Delta \{y_1'\}$, $\{y_1\} \simeq_\Delta \{y_1''\}$, and $\{y_1'\} \overset{1}{\simeq}_E \{y_1''\}$. Again, note the distinction between arbitrary substitution of the sequence and substitution of a single element. In the above inequality, $\check{M} : \mathbb{R}^L \to \mathbb{R}^D$ with $\check{M}(y_1) = Z + (G \circ \check{S})(y_1) := (\check{B} \circ \check{S})(y_1)$ and $Z \simeq \mathcal{N}(0, \sigma^2 C^2 \mathbf{I})$ is the mechanism that yields bottom-level subsampled, clipped, summed, and noised gradients for a single sequence. We can further upper-bound Eq. (11) via conditional coupling to prove the following result for bottom-level sampling with replacement and number of instances $\lambda = 1$:

**Lemma F.5.** *Consider arbitrary sequences $y_1, y_1', y_1'' \in \mathbb{R}^L$ such that $\{y_1\} \simeq_\Delta \{y_1'\}$, $\{y_1\} \simeq_\Delta \{y_1''\}$, $\{y_1\} \overset{1}{\simeq}_E \{y_1''\}$, i.e, all sequences are different, but $y_1'$ and $y_1''$ only differ in a single element. Let $\check{S} : \mathbb{R}^L \to \mathbb{R}^{L_C + L_F}$ be bottom-level sampling with replacement and number of subsequences $\lambda = 1$, as defined in Algorithm 4. Let $r = \frac{L_C + L_F}{L - L_F + 1}$ be the probability of sampling a subsequence containing any specific element. Further define forecast-to-context ratio $\phi = \frac{L_F}{L_C + L_F}$. Let $\check{B} : \mathbb{R}^{L_C + L_F} \to \mathbb{R}^D$ be an arbitrary mechanism that maps a single subsequence to a gradient and define bottom-level subsampled mechanism $\check{M} = \check{B} \circ \check{S}$. Then,*

$$H_\alpha \left( (1 - \rho) \cdot \check{M}_{y_1} + \rho \cdot \check{M}_{y_1'} \| (1 - \rho) \cdot \check{M}_{y_1} + \rho \cdot \check{M}_{y_1''} \right) \leq H_\alpha \left( P \| Q \right)$$

*with*

$$P = (1 - \rho) \cdot \check{B}_a + \rho \cdot (1 - r)\check{B}_{a'} + \rho \cdot r \cdot \phi \cdot \check{B}_{a_F'} + \rho \cdot r \cdot (1 - \phi) B_{a_C'},$$
$$Q = (1 - \rho) \cdot \check{B}_a + \rho \cdot (1 - r)\check{B}_{a'} + \rho \cdot r \cdot \phi \cdot \check{B}_{a_F''} + \rho \cdot r \cdot (1 - \phi) B_{a_C''},$$

*for some worst-case tuple of subsequences $a, a', a_F', a_C' \in \mathbb{R}^{L_C + L_F}$ where $\{a_F'\} \overset{1}{\simeq}_E \{a_F''\}$, $\{a_C'\} \overset{1}{\simeq}_E \{a_C''\}$, subsequences $a_F, a_F'$ only differ in their last $L_F$ elements (forecast window), subsequences $a_C, a_C'$ only differ in their first $L_C$ elements (context window), and the second mixture components are identical.*

*Proof.* For the following proof, we define an ordering on the support of our subsampling distributions $\check{S}_{y_1}, \check{S}_{y_1'}, \check{S}_{y_1''}$ with mass functions $\check{s}_{y_1}, \check{s}_{y_1'}, \check{s}_{y_1''}$ such that the earliest subsequence comes first, i.e.,

$$\text{supp}(\check{S}_{y_1}) = \{a_1, a_2, \ldots, a_{L-L_F+1}\},$$
$$\text{supp}(\check{S}_{y_1'}) = \{a_1, a_2, \ldots, a_{L-L_F+1}\},$$
$$\text{supp}(\check{S}_{y_1''}) = \{a_1'', a_2'', \ldots, a_{L-L_F+1}''\}.$$

We further partition $\text{supp}(\check{S}_{y_1'})$ and $\text{supp}(\check{S}_{y_1''})$ based on whether and where the single modified element appears in the subsequence, i.e.,

$$A' = \{a_i' \in \text{supp}(\check{S}_{y_1'}) \mid a_i' = a_i''\}$$
$$A_C' = \{a_i' \in \text{supp}(\check{S}_{y_1'}) \mid a_i'[1 : L_C] \neq\mid a_i''[1 : L_C]\}$$
$$A_F' = \{a_i' \in \text{supp}(\check{S}_{y_1'}) \mid a_i'[L_C + 1 :] \neq\mid a_i''[L_C + 1 :]\}$$
$$A'' = \{a_i'' \in \text{supp}(\check{S}_{y_1''}) \mid a_i' = a_i''\}$$
$$A_C'' = \{a_i'' \in \text{supp}(\check{S}_{y_1''}) \mid a_i'[1 : L_C] \neq\mid a_i''[1 : L_C]\}$$
$$A_F'' = \{a_i'' \in \text{supp}(\check{S}_{y_1''}) \mid a_i'[L_C + 1 :] \neq\mid a_i''[L_C + 1 :]\}$$

By definition of the bottom-level subsampled mechanism $\check{M}$, we have for the the components of the first mixture distribution:

$$\check{M}_{y_1} = \sum_{a \in \text{supp}(\check{S}_{y_1})} \check{B}_a \check{s}_{y_1}(a)$$

$$\check{M}_{y_1'} = \check{S}_{y_1'}(A') \sum_{a' \in A'} \check{B}_{a'} \check{s}_{y_1'}(a' \mid A') + \check{S}_{y_1'}(A'_C) \sum_{a'_C \in A'_C} \check{B}_{a'_C} \check{s}_{y_1'}(a'_C \mid A'_C), + \check{S}_{y_1'}(A'_F) \sum_{a'_F \in A'_F} \check{B}_{a'_F} \check{s}_{y_1'}(a'_F \mid A'_F).$$

Analogously, we have for the components of the second mixture distribution:

$$\check{M}_{y_1} = \sum_{a \in \text{supp}(\check{S}_{y_1})} \check{B}_a \check{s}_{y_1}(a)$$

$$\check{M}_{y_1''} = \check{S}_{y_1''}(A'') \sum_{a'' \in A''} \check{B}_{a''} \check{s}_{y_1''}(a'' \mid A'') + \check{S}_{y_1''}(A''_C) \sum_{a''_C \in A''_C} \check{B}_{a''_C} \check{s}_{y_1''}(a''_C \mid A''_C), + \check{S}_{y_1''}(A''_F) \sum_{a''_F \in A''_F} \check{B}_{a''_F} \check{s}_{y_1''}(a''_F \mid A''_F).$$

Here, the probability of not sampling, sampling in the context window, and sampling in the forecast window are $\check{S}_{y_1'}(A') = \check{S}_{y_1''}(A'') = (1-r)$, $\check{S}_{y_1'}(A') = \check{S}_{y_1''}(A'') = r \cdot (1-\phi)$, and $\check{S}_{y_1'}(A') = \check{S}_{y_1''}(A'') = r \cdot \phi$, respectively, with forecast-to-context-ratio $\phi = \frac{L_F}{L_C + L_F}$.

We can now define a coupling $\Gamma$ of conditional subsampling distributions $\check{S}_{y_1}(\cdot), \check{S}_{y_1'}(\cdot \mid A'), \check{S}_{y_1'}(\cdot \mid A'_C), S_{y_1'}(\cdot \mid A'_F)$, as well as their counterpart from the second mixture $\check{S}_{y_1}(\cdot), \check{S}_{y_1''}(\cdot \mid A''), \check{S}_{y_1''}(\cdot \mid A''_C), S_{y_1''}(\cdot \mid A''_F)$. Specifically, the coupling will match any subsequence from $y_1'$ with its counterpart from $y_1''$ that covers the same time range:

$$\gamma(a^{(1)}, a', a'_F, a'_C, a^{(2)}, a'', a''_F, a''_C)$$
$$= \left( \check{s}_{y_1}(a^{(1)}) \cdot \check{s}_{y_1'}(a' \mid A') \cdot \check{s}_{y_1'}(a'_F \mid A'_F) \cdot \check{s}_{y_1'}(a'_C \mid A'_C) \right) \cdot \left( \gamma(a^{(2)} \mid a^{(1)}) \cdot \gamma(a'' \mid a') \cdot \gamma(a''_F \mid a'_F) \cdot \gamma(a''_C \mid a'_C) \right)$$

with intra-distribution components

$$\gamma(a^{(2)} \mid a^{(1)}) = \mathbb{1}[a^{(1)} = a^{(2)}],$$
$$\gamma(a'' \mid a') = \mathbb{1}[a' = a''],$$
$$\gamma(a''_F \mid a'_F) = \mathbb{1}[a'_F = a'_i \implies a''_F = a''_i],$$
$$\gamma(a''_C \mid a'_C) = \mathbb{1}[a'_C = a'_i \implies a''_C = a''_i].$$

By construction of the coupling, all $(a^{(1)}, a', a'_F, a'_C, a^{(2)}, a'', a''_F, a''_C)$ with non-zero measure fulfill the following constraints: We have $\{a'_F\} \overset{1}{\simeq}_E \{a''_F\}$, $\{a'_C\} \overset{1}{\simeq}_E \{a''_C\}$, subsequences $a_F, a'_F$ only differ in their last $L_F$ elements (forecast window), subsequences $a_C, a'_C$ only differ in their first $L_C$ elements (context window). Furthermore, we have $a^{(1)} = a^{(2)}$ and $a' = a'$.

The result then follows from rewriting the two mixtures as marginals of the coupling and moving the coupling outside the hockey stick divergence using joint convexity, i.e., the usual conditional coupling procedure from Lemma D.4. $\qquad\square$

Finally, we can determine worst-case mixture components given worst-case subsequences to obtain a bound in terms of Gaussian mixtures:

**Lemma F.6.** *Consider the number of subsequences $\lambda = 1$ and batch size $\Lambda$. Let $r = \frac{L_C + L_F}{L - L_F + 1}$ and let $\rho = \lfloor \Lambda / \lambda \rfloor / N$ be the probability of sampling any specific sequence. Let $\check{B} : \mathbb{R}^{L_C + L_F} \to \mathbb{R}^D$ be a Gaussian mechanism $B(a) = g(a) + Z$ with $Z \sim \mathcal{N}(0, \sigma^2 C^2 \boldsymbol{I})$ and underlying function $g$ with maximum norm $C$ that maps a single subsequence to a gradient. Consider arbitrary subsequences $a, a', a'_F, a'_C, a''_F, a''_C \in \mathbb{R}^{L_C + L_F}$ where $\{a'_F\} \overset{1}{\simeq}_E \{a''_F\}$, $\{a'_C\} \overset{1}{\simeq}_E \{a''_C\}$, subsequences $a_F, a'_F$ only differ in their last $L_F$ elements (forecast window), subsequences $a_C, a'_C$ only differ in their first $L_C$ elements (context window). Define corresponding output distributions*

$$P = (1-\rho) \cdot \check{B}_a + \rho \cdot (1-r) \check{B}_{a'} + \rho \cdot r \cdot \phi \cdot \check{B}_{a'_F} + \rho \cdot r \cdot (1-\phi) B_{a'_C},$$
$$Q = (1-\rho) \cdot \check{B}_a + \rho \cdot (1-r) \check{B}_{a'} + \rho \cdot r \cdot \phi \cdot \check{B}_{a''_F} + \rho \cdot r \cdot (1-\phi) B_{a''_C}.$$

*Further define* $\tilde{P}(1) = MoG(\boldsymbol{\mu}, \tilde{\boldsymbol{p}}, \sigma)$ *with* $\tilde{\boldsymbol{\mu}} = \begin{bmatrix} 0 & 2 \end{bmatrix}^T$ *and* $\tilde{\boldsymbol{p}} = \begin{bmatrix} (1-\rho) + \rho \cdot (1-r) & \rho \cdot r \end{bmatrix}^T$. *Then, for all* $\alpha \geq 0$,

$$H_\alpha(P||Q) \leq \begin{cases} H_\alpha(\tilde{P}(1)||\mathcal{N}(0,\sigma)) & \text{if } \alpha \geq 1, \\ H_\alpha(\mathcal{N}(0,\sigma)||\tilde{P}(1)) & \text{if } 0 \leq \alpha < 1. \end{cases}$$

*Proof.* **Case 1** ($\alpha \geq 1$): Since the hockey stick divergence is jointly convex, it is also jointly quasi-convex. That is, the value attained by any interpolated argument is l.e.q. one of the arguments it is interpolating between. In the context of our mixture distributions, this means that

$$H_\alpha(P||Q)$$
$$\leq \max\{H_\alpha\left((1-\rho)\cdot\check{B}_a + \rho\cdot(1-r)\check{B}_{a'} + \rho\cdot r\cdot\check{B}_{a'_F}||(1-\rho)\cdot\check{B}_a + \rho\cdot(1-r)\check{B}_{a'} + \rho\cdot r\cdot\check{B}_{a''_F}\right),$$
$$H_\alpha\left((1-\rho)\cdot\check{B}_a + \rho\cdot(1-r)\check{B}_{a'} + \rho\cdot r\cdot B_{a'_C}||(1-\rho)\cdot\check{B}_a + \rho\cdot(1-r)\check{B}_{a'} + \rho\cdot r\cdot B_{a''_C}\right)\}.$$

Since the bound we shall derive shortly will lead to identical results for both terms, let us focus w.l.o.g. on bounding the first one, i.e.,

$$H_\alpha\left((1-\rho)\cdot\check{B}_a + \rho\cdot(1-r)\check{B}_{a'} + \rho\cdot r\cdot\check{B}_{a'_F}||(1-\rho)\cdot\check{B}_a + \rho\cdot(1-r)\check{B}_{a'} + \rho\cdot r\cdot\check{B}_{a''_F}\right)$$

Since the inputs $a$ and $a'$ to our mechanism in the first two mixture components are identical between both distributions, we can use advanced joint convexity (Lemma D.2) to restate our mixture divergence via

$$H_\alpha\left((1-\rho)\cdot\check{B}_a + \rho\cdot(1-r)\check{B}_{a'} + \rho\cdot r\cdot\check{B}_{a'_F}||(1-\rho)\cdot\check{B}_a + \rho\cdot(1-r)\check{B}_{a'} + \rho\cdot r\cdot\check{B}_{a''_F}\right)$$
$$=\rho\cdot r\cdot H_{\alpha'}\left(\check{B}_{a'_F}||(1-\beta(\alpha))\cdot\left((1-\rho)\cdot\check{B}_a + \rho\cdot(1-r)\check{B}_{a'}\right) + \beta(\alpha)\cdot\rho\cdot r\cdot\check{B}_{a''_F}\right)$$

with some $\alpha' \geq \alpha$ and some $\beta(\alpha) \in [0,1]$.

Recall that $\check{B}$ is a Gaussian mechanism. Since underlying function $g$ has maximum norm $C$, we know that the maximum $\ell_2$ distance of any two means is $2 \cdot C$. Due to translation equivariance of hockey stick divergences between Gaussian mixtures, we can assume w.l.o.g. that $\check{B}_{a'_F} = \mathcal{N}(\boldsymbol{0}, \sigma^2 C^2 \boldsymbol{I})$. We can thus define a constrained optimization problem over Gaussian mixture means to upper-bound our divergence:

$$\rho\cdot r\cdot H_{\alpha'}\left(\check{B}_{a'_F}||(1-\beta(\alpha))\cdot\left((1-\rho)\cdot\check{B}_a + \rho\cdot(1-r)\check{B}_{a'}\right) + \beta(\alpha)\cdot\rho\cdot r\cdot\check{B}_{a''_F}\right)$$
$$\leq \max_{\boldsymbol{\mu}^{(1)},\boldsymbol{\mu}^{(2)},\boldsymbol{\mu}^{(3)}} \rho\cdot r\cdot H_{\alpha'}\left(\mathcal{N}(\boldsymbol{0})||(1-\beta(\alpha))\cdot\left((1-\rho)\cdot\mathcal{N}(\boldsymbol{\mu}^{(1)}) + \rho\cdot(1-r)\mathcal{N}(\boldsymbol{\mu}^{(2)})\right) + \beta(\alpha)\cdot\rho\cdot r\cdot\mathcal{N}(\boldsymbol{\mu}^{(3)})\right),$$

subject to $\forall i : \boldsymbol{\mu}^{(1)} \in \mathbb{R}^D$ and $||\boldsymbol{\mu}^{(i)}|| \leq C$, where we omitted the covariance matrix $\sigma^2 C^2 \boldsymbol{I}$ for brevity. As is known from (Schuchardt et al., 2024) (see Lemma D.7), the maximum is attained at $\boldsymbol{\mu}^{(1)}, \boldsymbol{\mu}^{(2)}, \boldsymbol{\mu}^{(3)} = 2C \cdot \boldsymbol{e}_1$, where $\boldsymbol{e}_1$ has 1 in the first component and 0 everywhere else. In other words, all components in the second mixture are identical. Finally, we can apply advanced joint convexity in reverse order (note that it is an equality, not an inequality) to conclude that

$$H_\alpha\left((1-\rho)\cdot\check{B}_a + \rho\cdot(1-r)\check{B}_{a'} + \rho\cdot r\cdot\check{B}_{a'_F}||(1-\rho)\cdot\check{B}_a + \rho\cdot(1-r)\check{B}_{a'} + \rho\cdot r\cdot\check{B}_{a''_F}\right)$$
$$\leq H_\alpha\left(((1-\rho) + \rho\cdot(1-r))\cdot\mathcal{N}(2C\cdot\boldsymbol{e}_1, \sigma^2 C^2\boldsymbol{I}) + \rho\cdot r\cdot\mathcal{N}(\boldsymbol{0}, \sigma^2 C^2\boldsymbol{I})||\mathcal{N}(2C\cdot\boldsymbol{e}_1, \sigma^2 C^2\boldsymbol{I})\right)$$

Marginalizing out all but the first dimension, and scaling and rotating the coordinate system appropriately, concludes our proof for this case.

**Case 2** ($0 \leq \alpha < 1$): For the case $0 \leq \alpha < 1$, we can use the following fact: If $P, Q$ is dominating for $\alpha \geq 1$ under a symmetric neighboring relation, then $Q, P$ is dominating for $0 \leq \alpha < 1$ (Zhu et al. (2022), see Lemma D.10). $\square$

In conjunction with the previous steps and lemmata, this concludes our proof of the "$\leq$" part of Theorem 4.4. Next, let us show that this bound coincides with an optimistic upper bound, i.e., is tight.

### F.4. Optimistic Lower Bounds

Next, we can again construct worst-case datasets and gradient functions. As we shall see, we can conveniently use the same worst-case construction as for our bottom-level analysis from Appendix E.

**Theorem F.7.** *Consider top-level sampling without replacement and bottom-level sampling with replacement. Further consider any number of subsequences $\lambda \in \mathbb{N}$ and batch size $\Lambda$. Let $r = \frac{L_C + L_F}{L - L_F + 1}$ and let $\rho = \lfloor \Lambda / \lambda \rfloor / N$ be the probability of sampling any specific sequence. Define $\underline{P}(\lambda) = MoG(\boldsymbol{\mu}, \boldsymbol{p}, \sigma)$ with means $\boldsymbol{\mu} \in \mathbb{N}_0^{\lambda+1}$ and weights $\boldsymbol{p} \in [0, 1]^{\lambda+1}$ with $\mu_i = 2(i - 1)$ and $p_i = \mathrm{Binomial}(i \mid \lambda, r)$. Further define per-step privacy profile $H(\alpha) = \sup_{x \simeq_E x'} H_\alpha(\tilde{M}_x || \tilde{M}_{x'})$. Then,*

$$H(\alpha) \geq \begin{cases} H_\alpha((1 - \rho) \cdot \mathcal{N}(0, \sigma) + \rho \cdot \underline{P}(\lambda) || \mathcal{N}(0, \sigma)) & \text{if } \alpha \geq 1, \\ H_\alpha(\mathcal{N}(0, \sigma) || (1 - \rho) \cdot \mathcal{N}(0, \sigma) + \rho \cdot \underline{P}(\lambda)) & \text{if } 0 \leq \alpha < 1. \end{cases}$$

*Proof.* Exactly like in our proof of Theorem 4.3, we can construct the following gradient function

$$g(a) = \begin{cases} C \cdot \boldsymbol{e}_1 & \text{if } \exists l \in \{1, \dots, L_C + L_F\} : a_l = 1, \\ -C & \boldsymbol{e}_1 \text{otherwise.} \end{cases}$$

where $\boldsymbol{e}_1$ is the indicator vector that is non-zero in its first component, and $C$ is the clipping constant.

**Case 1 ($\alpha \geq 1$):** In this case, we can construct sequences $x_1, x_1' \in \mathbb{R}^L$ with $x_1 = \begin{bmatrix} 1 & 0 & \cdots & 0 \end{bmatrix}$ and $x_1' = \begin{bmatrix} 0 & 0 & \cdots & 0 \end{bmatrix}$ that differ in their first element. We can further construct sequences $x_2, \dots, x_N \in \mathbb{R}^L \setminus \{1\}$ such that $\forall m > n > 1 : x_m \neq x_n \wedge x_1 \neq x_n \neq x_1'$ (so that our dataset is a proper set, i.e., does not have duplicates). Finally, we can define datasets $x = \{x_1, x_2, \dots, x_N\}$ and $x' = \{x_1', x_2, \dots, x_N\}$.

Due to top-level subsampling, the chance of $a_1$ appearing in any sampled subsequence reduces by a factor of $1 - \rho$.

**Case 2 ($\alpha \geq 1$):** Here, we can interchange datasets $x$ and $x'$. The proof is analogous otherwise. $\square$

Note that the lower bound coincides with the bound from Theorem 4.4 for $\lambda = 1$, which concludes our proof of tightness.

**Theorem F.8.** *Consider top-level sampling without replacement, bottom-level Poisson sampling, number of subsequences $\lambda = 1$ and batch size $\Lambda$. Let $r = \min\{1, \lambda / (L - L_F + 1)\}$ be the resulting Poisson sampling rate, and let $\rho = \lfloor \Lambda / \lambda \rfloor / N$ be the probability of sampling any specific sequence. Like in Theorem E.8, define $\underline{P}(\lambda) = MoG(-1 \cdot \boldsymbol{\mu}, \boldsymbol{p}, \sigma)$ and $\underline{Q}(\lambda) = MoG(\boldsymbol{\mu}, \boldsymbol{p}, \sigma)$ with means $\mu_i = (i - 1)$ and weights $p_i = \mathrm{Binomial}(i - 1 \mid L_C + L_F, r)$. Then, per-step privacy profile $\tilde{H}(\alpha) = \sup_{x \simeq_E x'} H_\alpha(\tilde{M}_x || \tilde{M}_{x'})$ fulfills*

$$H(\alpha) \geq \begin{cases} H_\alpha((1 - \rho) \cdot \mathcal{N}(0, \sigma) + \rho \cdot \underline{P}(\lambda) || \mathcal{N}(0, \sigma)) & \text{if } \alpha \geq 1, \\ H_\alpha(\mathcal{N}(0, \sigma) || (1 - \rho) \cdot \mathcal{N}(0, \sigma) + \rho \cdot \underline{P}(\lambda)) & \text{if } 0 \leq \alpha < 1. \end{cases}$$

*Proof.* The proof is, again, fully analogous for our optimistic lower bound for bottom-level subsampling, i.e., Lemma E.12. Top-level sampling reduces the chance of sampling a non-zero element by a factor of $(1 - \rho)$. $\square$

### F.5. Dominating Pairs

In this section, we have derived multiple bounds of the form

$$(1 - \rho) \cdot \max\{0, 1 - \alpha\} + \rho H_\alpha(P || Q).$$

Since this is a weighted sum of two valid privacy profiles, it naturally fulfills all necessary and sufficient conditions for privacy profiles (Zhu et al., 2022), i.e.,

**Lemma D.11.** *A function $H : \mathbb{R}_+ \to \mathbb{R}$ is a privacy profile if and only if $H$ is convex, $H$ is decreasing, $H(0) = 1$, and $H(x) \geq \max\{1 - x, 0\}$.*

Thus, we can use the same toolset for constructing corresponding dominating pairs as discussed in Appendix E.1.5, i.e., convex conjugation ((Zhu et al., 2022), see Lemma D.12) or "connect-the-dots" ((Doroshenko et al., 2022), see Lemma D.16).

# G. Proofs from Section 4.3 (Context–Forecast Split)

For this section, we focus exclusively on top-level sampling without replacement and bottom-level sampling with replacement. Our goal is to prove the following statement for $(1, v)$-event-level privacy, where a single element can change its value by at most $v$:

**Theorem G.1.** *Consider top-level sampling without replacement and bottom-level sampling with replacement with $\lambda = 1$, batch size $\Lambda$, as well as context and forecast standard deviations $\sigma_C, \sigma_F \in \mathbb{R}_+$ (see Eq. (1)). Let $r = \frac{L_C + L_F}{L - L_F + 1}$ and $\rho = \lfloor \Lambda / \lambda \rfloor / N$. Further define forecast-to-context ratio $\phi = \frac{L_F}{L_C + L_F}$. Define $\hat{P}(1) = MoG(\boldsymbol{\mu}, \tilde{\boldsymbol{p}}, \sigma)$ with means $\hat{\boldsymbol{\mu}} = \begin{bmatrix} 0 & 2 \end{bmatrix}^T$ and weights $\tilde{\boldsymbol{p}} \in [0, 1]^2$ with $p_1 = 1 - p_2$ and*

$$p_2 = \rho \cdot r \cdot \phi \cdot \text{TVD}\left(\mathcal{N}(0, \sigma_F), \mathcal{N}(1, \sigma_F)\right) + \rho \cdot r \cdot (1 - \phi) \cdot \text{TVD}\left(\mathcal{N}(0, \sigma_C), \mathcal{N}(1, \sigma_C)\right)$$

*Then, the augmented per-step privacy profile $\hat{H}(\alpha) = \sup_{x \overset{1,v}{\simeq}_E x'} H_\alpha(\hat{M}_x || \hat{M}_{x'})$ fulfills*

$$\hat{H}(\alpha) \leq \begin{cases} H_\alpha(\hat{P}(1) || \mathcal{N}(0, \sigma)) & \text{if } \alpha \geq 1, \\ H_\alpha(\mathcal{N}(0, \sigma) || \hat{P}(1)) & \text{if } 0 \leq \alpha < 1. \end{cases}$$

The following special case from Section 4.3 immediately follows from setting $\sigma_C = \sigma_F$:

**Theorem 4.5.** *Consider $\lambda = 1$, batch size $\Lambda$, as well as context and forecast standard deviations $\sigma_C, \sigma_F \in \mathbb{R}_+$ with $\sigma_C = \sigma_F$. Let $r = \frac{L_C + L_F}{L - L_F + 1}$ and $\rho = \lfloor \Lambda / \lambda \rfloor / N$. Define $\hat{P}(1) = MoG(\boldsymbol{\mu}, \tilde{\boldsymbol{p}}, \sigma)$ with means $\hat{\boldsymbol{\mu}} = \begin{bmatrix} 0 & 2 \end{bmatrix}^T$ and weights $\tilde{\boldsymbol{p}} \in [0, 1]^2$, with $p_1 = 1 - p_2$ and*

$$p_2 = \rho \cdot r \cdot \text{TVD}\left(\mathcal{N}(0, \sigma_F), \mathcal{N}(1, \sigma_F)\right),$$

*with total variation distance $\text{TVD}(P, Q) = H_1(P || Q)$. Then, $\hat{H}(\alpha) = \sup_{x \overset{1,v}{\simeq}_E x'} H_\alpha(\hat{M}_x || \hat{M}_{x'})$ fulfills*

$$\hat{H}(\alpha) \leq \begin{cases} H_\alpha(\hat{P}(1) || \mathcal{N}(0, \sigma)) & \text{if } \alpha \geq 1, \\ H_\alpha(\mathcal{N}(0, \sigma) || \hat{P}(1)) & \text{if } 0 \leq \alpha < 1. \end{cases}$$

To prove these results, let us

1. Bound the privacy profile of $\hat{M}$ via constructing a conditional coupling between mixture decompositions of maximal couplings, leading to yet another divergence maximization problem involving multivariate Gaussian mixtures,

2. and solve the resultant optimization problem using joint quasi-convexity of the hockey stick divergence.

## G.1. Bound via Conditional Coupling and Maximal Couplings

For this section, we will use the following properties of maximal couplings, taken from (Balle et al., 2018) and Section 2.5 of (Den Hollander, 2012):

**Proposition G.2.** *Consider arbitrary distributions $P, Q$. There exists a coupling $\Pi^*$ of $P, Q$, referred to as* maximal coupling*, such that*

1. *$\Pi^*$ is an optimum of $\sup_{\Pi \in \Phi(P,Q)} \Pr(X, Y) \sim \Pi[X = Y]$, where $\Phi(P, Q)$ is the space of all couplings of $P, Q$,*

2. *$\Pi^*$ has marginals that decompose as $P = (1 - \tau)P^{(0)} + \tau P^{(1)}$ and $Q = (1 - \tau)P^{(0)} + \tau Q^{(1)}$ with $\tau = \text{TVD}(P, Q) = H_1(P, Q)$.*

As such, maximal coupling exactly correspond to our intuition of trying to determine the probability that we sample the same context and ground-truth forecast when a single sequence element changes its value by $v$.

**Lemma G.3.** *Consider top-level sampling without replacement and bottom-level sampling with replacement with $\lambda = 1$, batch size $\Lambda$, as well as context and forecast standard deviations $\sigma_C, \sigma_F \in \mathbb{R}_+$ (see Eq. (1)). Let $r = \frac{L_C + L_F}{L - L_F + 1}$ and*

$\rho = \lfloor \Lambda / \lambda \rfloor / N$. *Further define forecast-to-context ratio* $\phi = \frac{L_F}{L_C + L_F}$. *Then, the augmented per-step privacy profile* $\hat{H}(\alpha) = \sup_{x \overset{1,v}{\simeq}_E x'} H_\alpha(\hat{M}_x || \hat{M}_{x'})$ *fulfills*

$$\hat{H}(\alpha) \leq \max_{P,Q \in \Omega} H_\alpha(P||Q) \tag{12}$$

*where* $\Omega$ *is set of all pairs of multivariate Gaussian mixtures* $(P,Q)$ *with* $P = \sum_{i=1}^{6} w_i \cdot \mathcal{N}(\boldsymbol{\mu}_i^{(1)}, \sigma^2 \boldsymbol{I})$ *and* $Q = \sum_{j=1}^{6} w_j \cdot \mathcal{N}(\boldsymbol{\mu}_j^{(2)}, \sigma^2 \boldsymbol{I})$ *satisfying*

$$
\begin{aligned}
||\boldsymbol{\mu}_i^{(1)} - \boldsymbol{\mu}_j^{(1)}||_2 &\leq 2 &&\forall i,j \in \{1,\ldots,6\} \\
||\boldsymbol{\mu}_i^{(2)} - \boldsymbol{\mu}_j^{(2)}||_2 &\leq 2 &&\forall i,j \in \{1,\ldots,6\} \\
||\boldsymbol{\mu}_i^{(1)} - \boldsymbol{\mu}_j^{(2)}||_2 &\leq 2 &&\forall i,j \in \{1,\ldots,6\} \\
\boldsymbol{\mu}_i^{(1)} &= \boldsymbol{\mu}_i^{(2)} &&\forall i \in \{1,\ldots,4\},
\end{aligned}
\tag{13}
$$

*with* $\forall i : \boldsymbol{\mu}_i^{(1)} \in \mathbb{R}^D$ *and*

$$
\begin{aligned}
w_1 &= (1-\rho) \\
w_2 &= \rho \cdot (1-r) \\
w_3 &= \rho \cdot r \cdot \phi \cdot (1 - \mathrm{TVD}\left(\mathcal{N}(0,\sigma_F), \mathcal{N}(1,\sigma_F)\right)) \\
w_4 &= \rho \cdot r \cdot (1-\phi) \cdot (1 - \mathrm{TVD}\left(\mathcal{N}(0,\sigma_C), \mathcal{N}(1,\sigma_C)\right)) \\
w_5 &= \rho \cdot r \cdot \phi \cdot \mathrm{TVD}\left(\mathcal{N}(0,\sigma_F), \mathcal{N}(1,\sigma_F)\right) \\
w_6 &= \rho \cdot r \cdot (1-\phi) \cdot \mathrm{TVD}\left(\mathcal{N}(0,\sigma_C), \mathcal{N}(1,\sigma_C)\right).
\end{aligned}
$$

*Proof.* Let $\check{B} : \mathbb{R}^{L_C + L_F} \to \mathbb{R}^D$ be the mechanism that adds isotropic Gaussian noise to a pair of context and forecast windows, computes the resultant gradient, and adds isotropic Gaussian noise to the elements of the output gradient. Note that this mechanism is just another subsampled mechanism, where the subsampling distribution happens to be continuous. We can thus apply the usual approach based on joint couplings of conditional subsampling distributions. To this end, let us decompose the mechanism via $\check{B} = \underline{B} \circ \underline{S}$, where

$$\underline{S}(a) = \left[ (a[1 : L_C] + Z_C)^T \quad (a[L_C + 1 :] + Z_F)^T \right]^T$$

with $Z_F \sim \mathcal{N}(\mathbf{0}, \sigma_F^2 \cdot v^2 \cdot \boldsymbol{I})$, $Z_F \sim \mathcal{N}(\mathbf{0}, \sigma_F^2 \cdot v^2 \cdot \boldsymbol{I})$.

We know from our derivations in Appendix F and specifically Lemma F.5 that $\hat{H}(\alpha) \leq H_\alpha(P||Q)$

$$
\begin{aligned}
P &= (1-\rho) \cdot \check{B}_a + \rho \cdot (1-r)\check{B}_{a'} + \rho \cdot r \cdot \phi \cdot \check{B}_{a'_F} + \rho \cdot r \cdot (1-\phi)B_{a'_C}, \\
Q &= (1-\rho) \cdot \check{B}_a + \rho \cdot (1-r)\check{B}_{a'} + \rho \cdot r \cdot \phi \cdot \check{B}_{a''_F} + \rho \cdot r \cdot (1-\phi)B_{a''_C},
\end{aligned}
$$

for some worst-case tuple of subsequences $a, a', a'_F, a'_C \in \mathbb{R}^{L_C + L_F}$ where $\{a'_F\} \overset{1,v}{\simeq}_E \{a''_F\}$, $\{a'_C\} \overset{1,v}{\simeq}_E \{a''_C\}$, subsequences $a_F, a'_F$ only differ in their last $L_F$ elements (forecast window), subsequences $a_C, a'_C$ only differ in their first $L_C$ elements (context window), and the first two mixture components are identical between the two mixture distributions.

We know that $||a'_F - a''_F|| \leq v$ and that they only differ in one element in the forecast window. Thus, $\mathrm{TVD}(\underline{S}_{a'_F} || \underline{S}_{a''_F}) \leq \mathrm{TVD}(\mathcal{N}(0,\sigma_F), \mathcal{N}(1,\sigma_F))$. For brevity, let us define $\tau_F = \mathrm{TVD}(\mathcal{N}(0,\sigma_F), \mathcal{N}(1,\sigma_F))$. Using maximal couplings, we can restate $\check{B}_{a'_F}$ and $\check{B}_{a''_F}$ as sums of two mixtures where the first summand is identical, i.e.,

$$
\begin{aligned}
\check{B}_{a'_F} &= (1-\tau_F) \cdot \int_{\mathbb{R}^{L_C+L_F}} \underline{B}(z) \, \mathrm{d} \, \underline{S}_{a'_F}^{(0)}(z) + \tau_F \cdot \int_{\mathbb{R}^{L_C+L_F}} \underline{B}(z) \, \mathrm{d} \, \underline{S}_{a'_F}^{(1)}(z), \\
\check{B}_{a''_F} &= (1-\tau_F) \cdot \int_{\mathbb{R}^{L_C+L_F}} \underline{B}(z) \, \mathrm{d} \, \underline{S}_{a''_F}^{(0)}(z) + \tau_F \cdot \int_{\mathbb{R}^{L_C+L_F}} \underline{B}(z) \, \mathrm{d} \, \underline{S}_{a''_F}^{(1)}(z).
\end{aligned}
$$

Similarly, we know that $||a'_C - a''_C|| \leq v$ and that they only differ in one element in the context window. Thus, $\mathrm{TVD}(\underline{S}_{a'_C} || \underline{S}_{a''_C}) \leq \tau_C$ with $\tau_C = \mathrm{TVD}(\mathcal{N}(0,\sigma_C), \mathcal{N}(1,\sigma_C))$. Using maximal couplings, we can thus also restate

$\check{B}_{a'_C}$ and $\check{B}_{a''_C}$ as sums of two mixtures where the first summand is identical, i.e.,

$$\check{B}_{a'_C} = (1 - \tau_C) \cdot \int_{\mathbb{R}^{L_C + L_C}} \underline{B}(z) \, \mathrm{d}\, \underline{S}_{a'_C}^{(0)}(z) + \tau_C \cdot \int_{\mathbb{R}^{L_C + L_C}} \underline{B}(z) \, \mathrm{d}\, \underline{S}_{a'_C}^{(1)}(z),$$

$$\check{B}_{a''_C} = (1 - \tau_C) \cdot \int_{\mathbb{R}^{L_C + L_C}} \underline{B}(z) \, \mathrm{d}\, \underline{S}_{a''_C}^{(0)}(z) + \tau_C \cdot \int_{\mathbb{R}^{L_C + L_C}} \underline{B}(z) \, \mathrm{d}\, \underline{S}_{a''_C}^{(1)}(z).$$

Using the usual coupling toolkit, we can now define a continuous coupling between the six subsampling distributions $\underline{S}_a, \underline{S}_{a'}, \underline{S}_{a'_F}^{(0)}, \underline{S}_{a'_C}^{(0)}, \underline{S}_{a'_F}^{(1)}, \underline{S}_{a'_C}^{(1)}$ from the first mixture distribution and the six subsampling distributions $\underline{S}_a, \underline{S}_{a'}, \underline{S}_{a'_F}^{(0)}, \underline{S}_{a'_C}^{(0)}, \underline{S}_{a''_F}^{(1)}, \underline{S}_{a''_C}^{(1)}$. Since the first to fourth mixture component are identical, we can trivially construct a coupling $\Gamma$ such that the first to fourth element are identical for all pairs of tuples of subsequences in the support of $\Gamma$.

We can then rewrite all subsampling distributions as marginals of the coupling and move the coupling outside the hockey stick divergence using joint convexity (i.e., the approach from (Schuchardt et al., 2024)). Finally, we can conclude our proof by recalling that the gradient function $g : \mathbb{R}^{L_C + L_F} \to \mathbb{R}^D$ underlying $\underline{B}(z)$ is clipped to a norm of $C$ and yields identical results for identical inputs. $\qquad\square$

### G.2. Worst-Case Mixture Components

Next, we can solve the optimization problem in Lemma G.3 for $\alpha \geq 1$.

**Lemma G.4.** *For $\alpha \geq 1$, an optimal solution to the optimization problem in Lemma G.3 is given by $\boldsymbol{\mu}_5^{(1)} = \boldsymbol{\mu}_6^{(1)} = 2\boldsymbol{e}_1$, $\forall 1 \leq i \leq 4 : \boldsymbol{\mu}_i^{(1)} = \boldsymbol{0}$, and $\forall 1 \leq i \leq 4 : \boldsymbol{\mu}_i^{(2)} = \boldsymbol{0}$, where $\boldsymbol{e}_1$ is the first canonical unit vector.*

*Proof.* Consider any feasible solution to our optimization problem $P = \sum_{i=1}^{6} w_i \cdot \mathcal{N}(\boldsymbol{\mu}_i^{(1)}, \sigma^2 \boldsymbol{I})$ and $Q = \sum_{j=1}^{6} w_j \cdot \mathcal{N}(\boldsymbol{\mu}_j^{(2)}, \sigma^2 \boldsymbol{I})$ that fulfills

$$
\begin{aligned}
\|\boldsymbol{\mu}_i^{(1)} - \boldsymbol{\mu}_j^{(1)}\|_2 &\leq 2 && \forall i,j \in \{1, \dots, 6\} \\
\|\boldsymbol{\mu}_i^{(2)} - \boldsymbol{\mu}_j^{(2)}\|_2 &\leq 2 && \forall i,j \in \{1, \dots, 6\} \\
\|\boldsymbol{\mu}_i^{(1)} - \boldsymbol{\mu}_j^{(2)}\|_2 &\leq 2 && \forall i,j \in \{1, \dots, 6\} \\
\boldsymbol{\mu}_i^{(1)} &= \boldsymbol{\mu}_i^{(2)} && \forall i \in \{1, \dots, 4\},
\end{aligned}
\tag{14}
$$

with $\forall i, j : \boldsymbol{\mu}_i^{(1)}, \boldsymbol{\mu}_j^{(2)} \in \mathbb{R}^D$.

Since $P$ and $Q$ are identical in their first four components, we have via the advanced joint convexity property (Lemma D.2):

$$H_\alpha(P\|Q) = (w_5 + w_6) H_{\alpha'}(P'\|Q')$$

with some $\alpha' \geq \alpha$ and some mixture weights $w_5'^{(1)}, w_6'^{(1)}, w_1'^{(2)}, \dots, w_6'^{(2)}$ and

$$P' = w_5'^{(1)} \mathcal{N}(\boldsymbol{\mu}_5^{(1)}, \sigma^2 \boldsymbol{I}) + w_6'^{(1)} \mathcal{N}(\boldsymbol{\mu}_6^{(1)}, \sigma^2 \boldsymbol{I}),$$

$$Q' = \sum_{i=1}^{6} w_i'^{(2)} \mathcal{N}(\boldsymbol{\mu}_i^{(2)}, \sigma^2 \boldsymbol{I}),$$

Since hockey stick divergences are jointly convex, they are also jointly quasi-convex and thus

$$
\begin{aligned}
&(w_5 + w_6) \cdot H_{\alpha'}(P'\|Q') \\
\leq &(w_5 + w_6) \cdot \max_{i \in \{5,6\}} \max_{j \in \{1,\dots,6\}} H_{\alpha'}(\mathcal{N}(\boldsymbol{\mu}_i^{(1)}, \sigma^2 \boldsymbol{I}) \| \mathcal{N}(\boldsymbol{\mu}_i^{(2)}, \sigma^2 \boldsymbol{I})) \\
\leq &(w_5 + w_6) \cdot H_{\alpha'}(\mathcal{N}(2\boldsymbol{e}_1, \sigma^2 \boldsymbol{I}) \| \mathcal{N}(\boldsymbol{0}, \sigma^2 \boldsymbol{I}))
\end{aligned}
$$

The last inequality follows from our distance constraints and the fact that the hockey stick divergence between two Gaussians is monotonically increasing with the distance of their means (see Lemma D.7 for formal statement).

The result finally follows from applying advanced joint convexity in reverse order, i.e.,

$$(w_5 + w_6) \cdot H_{\alpha'}(\mathcal{N}(2\boldsymbol{e}_1, \sigma^2 \boldsymbol{I}) || \mathcal{N}(\boldsymbol{0}, \sigma^2 \boldsymbol{I})$$

$$=(w_5 + w_6) \cdot H_{\alpha'}\left(\sum_{i=5}^{6} w_i'^{(1)} \mathcal{N}(2\boldsymbol{e}_1, \sigma^2 \boldsymbol{I}) || \sum_{j=1}^{6} w_j'^{(2)} \mathcal{N}(\boldsymbol{0}, \sigma^2 \boldsymbol{I})\right)$$

$$=H_{\alpha}\left(\sum_{i=1}^{4} w_i^{(1)} \mathcal{N}(\boldsymbol{0}, \sigma^2 \boldsymbol{I}) + \sum_{i=5}^{6} w_i'^{(1)} \mathcal{N}(2\boldsymbol{e}_1, \sigma^2 \boldsymbol{I}) || \sum_{j=1}^{6} w_j^{(2)} \mathcal{N}(\boldsymbol{0}, \sigma^2 \boldsymbol{I})\right).$$

$\square$

Our main result Theorem G.1 then follows immediately from marginalization and the following lemma ((Zhu et al., 2022)):

**Lemma D.10.** *Let $\simeq$ be a symmetric neighboring relation on dataset space $\mathbb{X}$. It then holds that $\sup_{x \simeq x'} H_{\alpha}(M_x || M_{x'}) \leq H_{\alpha}(P, Q)$ for all $\alpha \geq 1$ if and only if $\sup_{x \simeq x'} H_{\alpha}(M_x || M_{x'}) \leq H_{\alpha}(Q, P)$ for all $0 < \alpha \leq 1$.*

# H. Inference Privacy

As discussed in Appendix H, our work and other works on DP-SGD (e.g. (Abadi et al., 2016)) focus on ensuring privacy of parameters $\theta$ to guarantee that information from any training sample $x_n$ (here: sequences) does not leak when releasing model $f_\theta$ or making predictions $f_\theta(x_m)$ (here: forecasts) for other data $x_m$. However, as our data evolves over time, there may also be scenarios where one wants to release a forecast $f_\theta(x_n)$ while simultaneously ensuring that no sensitive information from $x_n$ is leaked. Of course, the context window $L_C$ for generating this forecast will be much smaller than the size of an entire training set, i.e., it will be harder to obfuscate any individual element with noise while retaining high utility.

A direct approach to the problem, which has already been explored in (Li et al., 2019; Arcolezi et al., 2022) is adding calibrated Gaussian noise to time series $x_n$ to ensure $(w, v)$-event- or $(w, v)$-user-level privacy when releasing $f_\theta(\tilde{x}_n)$ with noised time series $x_n$.

In the following, we explore whether we can improve the privacy–utility trade-off of random input perturbations using amplification-by-subsampling. For this purpose, we apply Theorem 3.2 from (Koga et al., 2022), i.e., privacy amplification for time series release via Poisson subsampling. Rather than using this mechanism for downstream analysis, we use their mechanism to subsample our time series, and then impute the missing values with the remaining average, and then apply our model $f_\theta$ to the subsampled, noised, and reconstructed time series.

**Experimental Setup.** We use the same hyperparameters as in Appendix B, except for two changes: First, we omit DP-SGD training since we already enforce privacy at inference time. Second, since our privacy is now independent of context length and batch size, we use we use $L_C = 8, \Lambda = 64$ for `traffic`, $L_C = 2, \Lambda = 64$ for `electricity`, and $L_C = 2, \Lambda = 128$ for `solar_10_minutes`, which we found to lead to much better CRPS for non-private training during hyperparameter search on the validation set.

Tables 11 to 13 show the resultant test CRPS with varying Poisson subsampling rate $r$ and Gaussian noise calibrated such that we attain the desired $\varepsilon, \delta$. On `traffic` and `solar_10_minutes` subsampling with $r = 0.5$ or $r = 0.75$ yields better CRPS for small $\varepsilon$, while there is no improvement on `electricity`. This confirms that subsampling at inference time can in some circumstances improve utility while enforcing inference-time privacy for sequence $x_n$ when releasing forecast $f_\theta(x_n)$. Of course, there are various opportunities for further improving utility, e.g., via neural denoising or imputation.

Again, *this is not the primary focus of our work* (or any other work on DP-SGD), and we only included this discussion for completeness.

Table 11: CRPS for `traffic` when enforcing inference-time privacy with $v = 0.1, \delta = 1e^{-4}$. Bold font indicates the best Poisson subsampling rate $r$ per model (collection of three rows).

| MODEL | $\varepsilon = 0.5$ | $\varepsilon = 1$ | $\varepsilon = 2$ | $\varepsilon = 4$ | $\varepsilon = 8$ |
|---|---|---|---|---|---|
| SIMPLEFF ($r = 1.0$) | 7.611 $\pm$1.714 | 2.630 $\pm$0.311 | 1.030 $\pm$0.029 | 0.543 $\pm$0.017 | **0.348** $\pm$0.006 |
| SIMPLEFF ($r = 0.75$) | 6.367 $\pm$1.393 | 2.164 $\pm$0.244 | 0.880 $\pm$0.022 | 0.502 $\pm$0.011 | 0.352 $\pm$0.004 |
| SIMPLEFF ($r = 0.5$) | **5.521** $\pm$1.430 | **1.799** $\pm$0.260 | **0.748** $\pm$0.020 | **0.479** $\pm$0.005 | 0.381 $\pm$0.002 |
| DEEPAR ($r = 1.0$) | 1.754 $\pm$0.268 | 1.026 $\pm$0.091 | 0.685 $\pm$0.037 | 0.522 $\pm$0.035 | **0.425** $\pm$0.034 |
| DEEPAR ($r = 0.75$) | 1.376 $\pm$0.173 | 0.856 $\pm$0.055 | 0.622 $\pm$0.033 | 0.512 $\pm$0.040 | 0.445 $\pm$0.042 |
| DEEPAR ($r = 0.5$) | **1.019** $\pm$0.095 | **0.709** $\pm$0.035 | **0.575** $\pm$0.032 | **0.510** $\pm$0.043 | 0.463 $\pm$0.047 |
| DLINEAR ($r = 1.0$) | 4.262 $\pm$0.061 | 2.251 $\pm$0.031 | 1.221 $\pm$0.017 | 0.707 $\pm$0.009 | 0.442 $\pm$0.005 |
| DLINEAR ($r = 0.75$) | 3.473 $\pm$0.046 | 1.863 $\pm$0.025 | 1.044 $\pm$0.014 | 0.637 $\pm$0.008 | **0.430** $\pm$0.005 |
| DLINEAR ($r = 0.5$) | **2.650** $\pm$0.037 | **1.463** $\pm$0.020 | **0.870** $\pm$0.011 | **0.580** $\pm$0.007 | 0.438 $\pm$0.004 |
| SEASONAL ($r = 1.0$) | 7.871 $\pm$0.010 | 4.273 $\pm$0.006 | 2.358 $\pm$0.003 | 1.350 $\pm$0.002 | 0.827 $\pm$0.001 |

Table 12: CRPS for `electricity` when enforcing inference-time privacy with $v = 10, \delta = 1e^{-4}$. Bold font indicates the best Poisson subsampling rate $r$ per model (collection of three rows).

| MODEL | $\varepsilon = 0.5$ | $\varepsilon = 1$ | $\varepsilon = 2$ | $\varepsilon = 4$ | $\varepsilon = 8$ |
|---|---|---|---|---|---|
| SIMPLEFF ($r = 1.0$) | **0.060** $\pm 0.060$ | **0.058** $\pm 0.058$ | **0.057** $\pm 0.057$ | **0.056** $\pm 0.056$ | **0.056** $\pm 0.056$ |
| SIMPLEFF ($r = 0.75$) | 0.118 $\pm 0.118$ | 0.117 $\pm 0.117$ | 0.116 $\pm 0.116$ | 0.116 $\pm 0.116$ | 0.116 $\pm 0.116$ |
| SIMPLEFF ($r = 0.5$) | 0.179 $\pm 0.179$ | 0.178 $\pm 0.178$ | 0.178 $\pm 0.178$ | 0.178 $\pm 0.178$ | 0.177 $\pm 0.177$ |
| DEEPAR ($r = 1.0$) | **0.055** $\pm 0.055$ | **0.053** $\pm 0.053$ | **0.052** $\pm 0.052$ | **0.051** $\pm 0.051$ | **0.051** $\pm 0.051$ |
| DEEPAR ($r = 0.75$) | 0.112 $\pm 0.112$ | 0.111 $\pm 0.111$ | 0.111 $\pm 0.111$ | 0.110 $\pm 0.110$ | 0.110 $\pm 0.110$ |
| DEEPAR ($r = 0.5$) | 0.169 $\pm 0.169$ | 0.168 $\pm 0.168$ | 0.168 $\pm 0.168$ | 0.168 $\pm 0.168$ | 0.168 $\pm 0.168$ |
| DLINEAR ($r = 1.0$) | **0.063** $\pm 0.063$ | **0.059** $\pm 0.059$ | **0.058** $\pm 0.058$ | **0.057** $\pm 0.057$ | **0.057** $\pm 0.057$ |
| DLINEAR ($r = 0.75$) | 0.125 $\pm 0.125$ | 0.122 $\pm 0.122$ | 0.122 $\pm 0.122$ | 0.121 $\pm 0.121$ | 0.121 $\pm 0.121$ |
| DLINEAR ($r = 0.5$) | 0.186 $\pm 0.186$ | 0.184 $\pm 0.184$ | 0.184 $\pm 0.184$ | 0.183 $\pm 0.183$ | 0.183 $\pm 0.183$ |
| SEASONAL ($r = 1.0$) | 0.079 $\pm 0.000$ | 0.074 $\pm 0.000$ | 0.071 $\pm 0.000$ | 0.070 $\pm 0.000$ | 0.070 $\pm 0.000$ |

Table 13: CRPS for `solar_10_minutes` when enforcing inference-time privacy with $v = 1, \delta = 1e^{-4}$. Bold font indicates the best Poisson subsampling rate $r$ per model (collection of three rows).

| MODEL | $\varepsilon = 0.5$ | $\varepsilon = 1$ | $\varepsilon = 2$ | $\varepsilon = 4$ | $\varepsilon = 8$ |
|---|---|---|---|---|---|
| SIMPLEFF ($r = 1.0$) | 3.999 $\pm 2.246$ | 1.791 $\pm 0.588$ | 1.102 $\pm 0.163$ | 0.897 $\pm 0.076$ | **0.836** $\pm 0.079$ |
| SIMPLEFF ($r = 0.75$) | 3.280 $\pm 1.701$ | 1.457 $\pm 0.401$ | **0.971** $\pm 0.093$ | **0.868** $\pm 0.069$ | 0.850 $\pm 0.073$ |
| SIMPLEFF ($r = 0.5$) | **3.022** $\pm 1.723$ | **1.441** $\pm 0.386$ | 1.032 $\pm 0.097$ | 0.936 $\pm 0.045$ | 0.921 $\pm 0.034$ |
| DEEPAR ($r = 1.0$) | 2.483 $\pm 1.071$ | **1.361** $\pm 0.299$ | **0.892** $\pm 0.078$ | **0.712** $\pm 0.024$ | **0.641** $\pm 0.020$ |
| DEEPAR ($r = 0.75$) | 2.257 $\pm 0.734$ | 1.405 $\pm 0.321$ | 1.021 $\pm 0.146$ | 0.873 $\pm 0.107$ | 0.804 $\pm 0.087$ |
| DEEPAR ($r = 0.5$) | **2.123** $\pm 0.681$ | 1.565 $\pm 0.403$ | 1.304 $\pm 0.277$ | 1.206 $\pm 0.228$ | 1.167 $\pm 0.212$ |
| DLINEAR ($r = 1.0$) | 2.792 $\pm 0.154$ | 1.783 $\pm 0.059$ | **1.289** $\pm 0.022$ | **1.043** $\pm 0.017$ | **0.918** $\pm 0.021$ |
| DLINEAR ($r = 0.75$) | 2.443 $\pm 0.157$ | 1.663 $\pm 0.085$ | 1.305 $\pm 0.061$ | 1.150 $\pm 0.049$ | 1.088 $\pm 0.041$ |
| DLINEAR ($r = 0.5$) | **2.187** $\pm 0.123$ | **1.640** $\pm 0.089$ | 1.419 $\pm 0.073$ | 1.340 $\pm 0.062$ | 1.315 $\pm 0.057$ |
| SEASONAL ($r = 1.0$) | 9.461 $\pm 0.113$ | 5.485 $\pm 0.059$ | 3.433 $\pm 0.029$ | 2.371 $\pm 0.014$ | 1.816 $\pm 0.008$ |

# I. Generalizations

For the sake of exposition, and to spare our readers from even further complicating the already involved notation in some of our later proofs, we made certain simplifying assumptions about the shape of the time series, and focused on analyzing 1-event and $(1, v)$-event level privacy.

In the following, we discuss how these restrictions can (except for one case involving amplification-by-augmentation with $\sigma_C \neq \sigma_F$) be easily lifted.

## I.1. Shape of Time Series

**Minimum Length.** The first simplifying assumption we made was that $L - L_F + 1 \geq L_C + L_F$, which means that the first element $x_n[1]$ of a time series $x_n$ appears (after zero-padding with length $L_C$) in $L_C + L_F$ different time series at different positions. If $L - L_F + 1 < L_C + L_F$, then all context windows will always contain some of the padding elements. Thus, the maximum number of subsequences an element can contribute to is $\max\{0, \min\{L_C + L_F, L - L_F + 1\}\}$. Since all our proofs are only dependent on number of subsequences that contain a specific sensitive element, and the fact that there are no duplicate appearances in a subsequence, one can replace $L_C + L_F$ with $\max\{0, \min\{L_C + L_F, L - L_F + 1\}\}$ in all our guarantees to cover this edge case.

**Variable length.** Using the same stochastic dominance argument as in our proof of Appendix E.3, one can show that all our upper- and lower bounds for bottom-level sampling with replacement are monotonically increasing in $\frac{\max\{0,\min\{L_C+L_F,L-L_F+1\}\}}{L-L_F+1}$. Thus, given a dataset composed of variable-length sequences, one can evaluate our bounds using whatever length maximizes the above two functions. The tight bounds will remain tight, since we have to make the worst-case assumption that exactly this optimal-length sequence is changed under our neighboring relation.

**Dimensionality.** None of our proofs use the fact that our dataset space is $\mathbb{X} = \mathcal{P}(\mathbb{R}^L)$ as opposed to $\mathbb{X} = \mathcal{P}(\mathbb{R}^{L \times D_{\text{in}}})$ — except for our amplification-by-augmentation proof from Lemma G.3 for $(1, v)$-event privacy. If we interpret $v$ as implying that $\forall n, \forall t : ||x_n[t] - x_n[t]||_2 \leq v$, then the proof are identical. If we interpret $v$ as implying that $\forall n, \forall t, \forall d : |x_n[t, d] - x_n[t, d]|$, then the maximum total variation distance between two subsequences that differ in the same element increases to $\text{TVD}\left(\mathcal{N}(0, \sigma), \mathcal{N}(\sqrt{D_{\text{in}}}, \sigma)\right)$ and we need to adjust the bound accordingly. Thus, our methods can in principle also be applied to highly-dimensional time series such as videos.

## I.2. Neighboring Relations

Finally, we can generalize our bounds to $(w, v)$-event and $(w, v)$-user privacy with $v \in \mathbb{R}_+ \cup \{\infty\}$.

**Bottom- and Top-Level Subsampling.** Our proofs for bottom- and top-level sampling only depend on the number of subsequences that contain at least one element of any specific individual (see, e.g., Appendix E.1.2 where we abstract our analysis $k$-group-substitutions). Under the $(w, v)$-event neighboring relation, all elements are adjacent and thus this number is given by $\max\{0, \min\{L_C + L_F + w - 1, L - L_F + 1\}\}$. Under the $(w, v)$-user neighboring relation, all elements can be placed arbitrarily far from each other. One can thus place them such that every subsequence only ever contains a single element, i.e., $\max\{0, \min\{w \cdot (L_C + L_F), L - L_F + 1\}\}$.

**Amplification by Data Augmentation.** In the case of $\sigma_C = \sigma_F$ (Theorem 4.5), our guarantee is only dependent on the maximum magnitude of change between two time series that differ in the same indices. We can thus adjust our bound by using $\text{TVD}\left(\mathcal{N}(0, \sigma), \mathcal{N}(\sqrt{w}, \sigma)\right)$. Only our guarantee for $\sigma_c \neq \sigma_F$ cannot be easily generalized, because our proof of Lemma F.5 explicitly makes a case distinction based on whether an element appears in the context window or the forecast window. Future work would need to generalize our proof make a fine-grained case distinction over the number of elements that appear in the context window and in the forecast window.

