# OpenReview forum: "Privacy Amplification by Structured Subsampling for Deep Differentially Private Time Series Forecasting"
_ICML.cc/2025/Conference — ICML 2025 spotlightposter_

### Official Review · Reviewer_gQDa · 2025-03-12

**Overall Recommendation:** 4

**Summary:**

This paper studies privacy amplification with structured subsampling, for applications in DP-SGD training on forecasting models. The sampling considered works by first selecting a subset of time series (top-level sample), then one or more contiguous subsequences per sample (bottom-level sample), and finally splitting each subsequence into a context window and ground-truth forecast. The authors derive event- and user-level privacy guarantees for sampling contiguous subsequences. They also study the tradeoffs between the choice of parameters for the top or bottom level sampling, and show that composing batches of many top-level sequences is optimal for the privacy guarantees. To handle the privacy of the context/forecast windows, the authors propose data augmentation with Gaussian noise.

**Claims And Evidence:**

Yes.

**Essential References Not Discussed:**

N/A

**Experimental Designs Or Analyses:**

- The authors mainly study the theoretical privacy guarantees of the different subsampling parameters. However, it would also be informative to empirically compare how the subsampling affects the utility of the model, independent of the privacy guarantees, so that we can know whether improvements are due to the privacy analysis or the choice of subsampling parameters. This would be a useful baseline to know how much the new privacy analysis improves on the results.
- It might be good to compare against a baseline using DP-SGD in a black-box manner, to better demonstrate the improvements from the new analysis.

**Methods And Evaluation Criteria:**

- It might be good to include results on a wider range of epsilon in Table 1, or to include a plot.

**Other Comments Or Suggestions:**

N/A

**Other Strengths And Weaknesses:**

Strengths:
- Privacy analysis is thorough and the tradeoffs between the subsampling parameters are studied in detail.

Weaknesses:
- The analysis seems very specific to the application in forecasting models, and it is unclear whether their results will be useful in other applications.
- The techniques used are not novel, and the main novelty is applying existing techniques to a new application.
- It would be good to have more experimental baselines to determine the impact of the new analysis.

**Questions For Authors:**

- How do the subsampling parameters affect the model utility, independent of the privacy analysis?

**Relation To Broader Scientific Literature:**

The authors are the first to study the privacy analysis for this particular type of structured subsampling used in forecasting, though the techniques used are not new.

**Theoretical Claims:**

I did not check the proofs in the appendix.

---

> ### Author Rebuttal · Authors · 2025-03-31
>
> Thank you for your review and great suggestions for further expanding our experimental evaluation!
> Please note that we cannot update the manuscript during this phase, but will integrate your suggestions as soon as possible.
>
> ## Additional experiments/baselines
>
> ### Effect of subsampling parameters on utility, independent of privacy
> There are three key parameters: How we select sequences ("top-level"), how many subsequences $\lambda$ we sample per sequence, and batch size $\Lambda$.
>
> In the following, we first let $\lambda=1$ (like in our other experiments), and vary the top-level scheme and $\Lambda$:
> https://figshare.com/s/490812bd1b089d5c1dc5
> Larger batch sizes $\Lambda$ improve utility on `solar_10_minutes` for some models, potentially due to its sparsity. However, the top-level scheme has no significant effect on utility.
> Based on this observation, we choose large batch sizes $\Lambda=128$ and vary $\lambda$:
> https://figshare.com/s/c4068a78b2d4f06aee25
> There is no significant difference between $\lambda=1$ and $\lambda=16$, and no top-level scheme consistently outperforms any other.
>
> We will add these results to Appendix C.2 and reference them from 5.2 to motivate our parameter choices for DP training.
>
> ### Black-Box DP-SGD Baseline
> Thank you for this suggestion, which will let us better explain to readers why structured subsampling is preferable to standard DP-SGD.
>
> With standard fixed-batchsize DP-SGD, we would directly sample a batch of size $\Lambda$ from the $N_\mathrm{total} = N \cdot (L-L_F+1)$ subsequences of which $L_C+L_F$ contain sensitive information, with $L_C$ and $L_F$ being the context and forecast length.
> For $L_C+L_F=1$, one can use known tight bounds in a black-box manner (see Theorem 11.2 from Zhu et al. (2022)). However, they assume that only $1$ sensitive subsequence can appear in a batch, **which makes them invalid when there are multiple sensitive subsequences  ($L_C+L_F>1$)**
>
> Like in our Theorem 4.3, we know that with $L_C + L_F > 1$ the privacy of standard DP-SGD is in fact optimistically lower-bounded by
> $H_{e^\varepsilon}(\sum_{i=0}^{L_C+L_F} \gamma_i \cdot N(2i, \sigma)||N(0,\sigma))$, where $\gamma_i$ is the probability of sampling $i$ substituted subsequences, i.e., there is a risk of multiple leakage.
>
> The following compares this lower bound to our upper bound for structured subsampling with $\lambda=1$ under  $N_\mathrm{total}=10^6$ and varying $\Lambda,L_C+L_F$:
> https://figshare.com/s/c061340645d5132c4a7e
> **Standard DP-SGD is equally private for $L_C+L_F=1$ and less private for $L_C+L_F>1$**, precisely because it cannot prevent multiple leakage.
>
> We will include this discussion and reference it from ll.90-92 in Section 1.1: "This risk of multiple leakage is underestimated if we apply guarantees for standard DP-SGD in a black-box manner (see Appendix C.3), [...]".
>
> ### Wider range of $\varepsilon$ in Table 1
> Thank you! We expanded the set of $\varepsilon$:
> https://figshare.com/s/bbef7e6e8ad6ce500bf1
>
> ---
>
> ## Novelty of Techniques
> Since our work is the first to analyze multi-level subsampling and privacy amplification for sequential data, we assume that you are referring to the fact that we use couplings to bound divergences in our derivations (please let us know if we misunderstood you).
>
> Please note that *defining couplings only makes up a part of our overall proof strategy*. For example, of our derivations for bottom-level sampling in on pp. 28-37, only Appendices E.1.3 and E.2.0 on pp.31-32;35 make use of this technique, whereas the remaining proofs introduce novel techniques and results (e.g. adaptive parallel composition for dominating pairs in E.1.1).
>
> Furthermore, applying couplings still requires non-trivial efforts (defining partitionings, proving validity of couplings, identifying distance constraints, deriving worst-case components given constraints,...)
>
> ## Applications Beyond Forecasting
> While we focus on forecasting, the underlying approach generalizes to multiple domains:
>
> 1. Our method can be directly applied to **private self-supervised training for language models**, where one also predicts "ground truth sequences" from a "context window" (see Section 4.5).
> 2. Our bounds directly apply to **user-level privacy for arbitrary data**, where $K$ records from one of $N$ data holders are substituted (see ll. 709-711).
> 3. Our novel **amplification-by-augmentation** method from Section 4.3 applies to any neighboring relations that enforces a bounded $\ell_2$ distance between datasets (e.g. privacy for patches in images).
>
> Based on your feedback, we will mention all applications at the end of Section 4.5 so that the broader impact of our work is highlighted in one specific paragraph and not scattered over the manuscript.
>
> ---
>
> Thank you again for your efforts!
> Please let us know if you have any additional questions during the discussion period.
>
> ---
>
> Zhu et al. Optimal Accounting of Differential Privacy via Characteristic Function. AISTATS 2022

---

> > ### Comment · Reviewer_gQDa · 2025-04-04
> >
> > Thank you for all the clarifications! I have increased my score accordingly.

---

### Official Review · Reviewer_GCw3 · 2025-03-17

**Overall Recommendation:** 4

**Summary:**

This paper investigates the problem of training forecasting models (specifically those that leverage the temporal structure directly in their architecture) on both univariate and multivariate time series data. As is present most prior work on DP-SGD, amplificaton is a key result necessary for achieving realistic privacy-utility tradeoffs on deep learning tasks. The authors focus their work in this paper on deriving a structured subsampling procedure that provides similar amplificaton guarantees for these forecasting models. Finally, they demonstrate the new privacy utility tradeoffs when using this subsampling procedure.

### Update after rebuttal ###

The authors have addressed my concerns and provided sufficient detail about how the accounting was implemented. Thus I feel confident about this utility of this new subsampling mechanism for practice. Thus, I have maintained my score at Accept.

**Claims And Evidence:**

The claims in the submission are supported with theoretical analysis, which is necessary for amplifcation results. Their empirical results are less focused on the main claim of the paper and an investigation of the impact of the main contribution of the paper.

**Essential References Not Discussed:**

None.

**Experimental Designs Or Analyses:**

Yes, I checked the soundness and validity of the experimental designs. I have no issues for the choices made.

**Methods And Evaluation Criteria:**

The proposed methods are well motivated for the forecasting problem, especially compared to assumptions placed by prior work that would lead to worse privacy-utility tradeoffs. Structured subsampling of the time sequences in a novel way that results in new amplification results is directly applicable to the forecastijng domain.

**Other Comments Or Suggestions:**

None.

**Other Strengths And Weaknesses:**

Strengths:

- Novel amplification theory for time series data
- Well-written theoretical section
- Widely applicable result to enable better DP learning of time series data

Weaknesses:

- Section 5.2 could benefit from improved writing and detail for clarity and motivation

**Questions For Authors:**

1. In the appendix the authors state that they use the `dp_accounting` library in their experiments. I'd like to understand how the amplification result presented in the paper was implemented with that library. Since it doesn't support this new result how was it incorporated? (i.e. how should I think about this result as an analogy to the q parameter that represents the sampling probability of batch)
2. The bi-level sampling seemed primarily motivated by LLMs but there are no natural language results? What do you see as the obstacles for adapting the method for that original motivation?

**Relation To Broader Scientific Literature:**

This paper combines existing techniques in randomization to achieve privacy amplification in a new way for time series data. This builds upon a recent set of work by Li et al. and Kogi et al. to provide amplifcation results in the time series domain. Additionally, it provides a new way to train deep learning time series models with differential privacy. This is especially important for privacy-sensitive domains such as healthcare and finance where time series data is abundant.

**Theoretical Claims:**

I checked the correctness of the proofs in Section 4. While it is possible I may have missed minor details, from my current understanding these proofs are sound.

---

> ### Author Rebuttal · Authors · 2025-03-31
>
> Thank you for your review!
> Please note that we cannot update the manuscript during the current rebuttal phase, but will integrate your feedback as soon as possible.
>
> ### Motivation for Section 5.2
> Upon re-reading the section, we agree that we could have done a better job of explaining what exactly we intend to demonstrate by evaluating the CRPS of different models at different $\epsilon$, especially because our paper may also be read by forecasting practitioners outside the differential privacy community.
>
> **Based on your feedback, we will replace the first paragraph of Section 5.2 with the following**:
> "Our previous experiments show how different parameterizations of the considered subsampling scheme affect the privacy of DP-SGD applied to time series. However, altering how batches are sampled will affect the training dynamics of forecasting models. Parameterizations that offer strong privacy (small $\epsilon$ and $\delta$) could potentially result in low model utility. The following experiments serve to show that we can in fact train neural forecasting models with strong privacy guarantees while retaining better utility than non-neural methods, i.e., DP-SGD for time series offers a good privacy--utility trade-off."
>
> Please let us know if there are other parts of Section 5.2 that you think should be improved.
>
> ### Implementation of Bounds
> For reference, the bounds for $\lambda=1$ in Theorems 4.2, 4.4, and 4.5 are of the form
> $\begin{cases}
> H_\alpha((1-q) \cdot N(0, \sigma) + q \cdot N(2, \sigma)|| N(0,\sigma) ) & \text{if } \alpha \geq 1\\\\
> H_\alpha(N(0,\sigma) || (1-q) \cdot N(0, \sigma) + q \cdot N(2, \sigma) ) & \text{if } \alpha < 1
> \end{cases}$
> where $q$ depends on the neighboring relation, top-level scheme, bottom-level scheme, and amount of Gaussian data augmentation.
>
> The first case is equivalent to a `GaussianPrivacyLoss` with `sensitivity=2`, `sampling_prob=q` and `AdjacencyType.REMOVE`. The second case is equivalent to `sensitivity=2`, `sampling_prob=q` and `AdjacencyType.ADD`. We implement the case distinction through a class `SwitchingPrivacyLoss` that overrides `get_delta_for_epsilon` and evaluates one of the two `PrivacyLoss` objects, depending on the value of $\varepsilon$ (see `src/dp_timeseries/privacy/pld.py` in the supplementary material).
>
> For $\lambda > 1$ and our optimistic bounds (e.g. Theorem 4.3), we need to consider a mixture with more than two components. For this, we can apply the same approach to the existing `MixtureGaussianPrivacyLoss` with `AdjacencyType.REMOVE` and `AdjacencyType.ADD`.
>
> For $\lambda > 1$ and our pessimistic bounds (e.g. Theorem F.2), we need pairs of Gaussian mixtures and weighted sums of privacy profiles. These cannot be implemented via existing child classes and require custom classes (see `DoubleMixtureGaussianPrivacyLoss` and `WeightedSumPrivacyLoss` in `src/dp_timeseries/privacy/pld.py` in the supplementary material)
>
> ### Motivation for Bi-Level Subsampling (LLMs?)
> The bi-level approach is actually not motivated by LLMs, although your comment made us realize that a reader may get this impression from the "Bi-Level Subsampling for LLMs" paragraph in Section 2.
>
> Instead, our goal is to analyze the privacy implications of combining DP-SGD with a batching approach that has already been used to train forecasting models before the advent of LLMs. For example, in GluonTS, top-level sampling is implemented by a [TrainDataLoader](https://ts.gluon.ai/stable/api/gluonts/gluonts.dataset.loader.html) and bottom-level sampling is implemented by an [InstanceSampler](https://ts.gluon.ai/stable/api/gluonts/gluonts.transform.sampler.html).
>
> **Based on your feedback, we will replace the first sentence of Section 4.2 with the following**:
> "Recall from Section 1.1 that typical approaches for training forecasting models may also randomize which sequences $x_n$ contribute to a batch (e.g. by shuffling). As is standard with DP-SGD, we replace this shuffling operation with an independent sampling operation per batch to simplify privacy accounting".
>
> ### Application to LLMs
> There is in fact nothing that would prevent the direct application to token/sentence-level private self-supervised training with teacher forcing.
>
> Just like in forecasting, one samples a finite "context window" from which a "ground truth sequence" is predicted. One would just need to replace the data loader in existing LLM implementations. The main limitation is that this method may be less optimized for memory access in distributed training than the highly refined data loaders that are likely being used to train commercial models on large clusters.
>
>
> Based on your comment, we will replace the last sentence of Section 4.5 with the above paragraph, since "the connection [... to ...] language modeling is immediate" is somewhat ambiguous.
>
> ---
>
> We hope that we addressed all your comments to your satisfaction.
> Please let us know if you have any additional questions during the discussion period.

---

### Official Review · Reviewer_r1DZ · 2025-03-18

**Overall Recommendation:** 4

**Summary:**

This paper studies privacy amplification under subsampling when working with forecasting models on time series data. In these cases, the dataset usually consists of sequences, and subsampling occurs on multiple levels: top-level sampling chooses a subset of sequences, bottom-level sampling chooses a subsequence from each of these layers, and then the subsequences are split into context and forecast (after which a model is trained on this data via Noisy SGD w Gaussian noise). The paper studies thoroughly how these various sampling choices impact privacy amplification by subsampling both analytically and empirically, making the following observations (among some others):

a) They show that as one would expect, when the top-level sampling scheme is deterministic, and the bottom-level sampling is sampling $\lambda$ sequences without replacement, for a fixed batch size, $\lambda=1$ corresponds to the most per-epoch privacy. They also tightly characterize the privacy profile in this case. Next, they consider that when $\lambda > 1$, fewer epochs are needed for the same number of training steps, and hence ask whether $\lambda=1$ is still an optimal choice when you hold the number of training steps fixed (as opposed to a per epoch type guarantee). They numerically show that this is the case, by showing a lower bound on the per epoch privacy profile for $\lambda > 1$ and compare that to the tight profile for $\lambda=1$. Applying composition, they show that in many ranges of $\epsilon$ (and for various settings of other parameters), $\lambda=1$ continues to be an optimal choice.

b)  Next, they consider the effect of using sampling wor for top-level sampling and bottom-level sampling. In this case, they analytically show that per-step privacy profile is indeed additionally amplified roughly by the probability of sampling a sequence in the top-level. They also numerically show that this scheme gives amplification over deterministic top-level sampling at a per-epoch level (by applying composition on their per-step guarantee).

c) Finally, in cases where the values in neighboring datasets can change in bounded ways ($+v$ or $-v$) they consider randomizing the context and forecast by adding Gaussian noise (in addition to top and bottom level sampling wor), and show that this further amplifies the privacy (the quantitative guarantee depends on the variance of the noise added).

They also carry out some experiments to measure whether utility can be maintained while training on common benchmarks using many models of interest. They use tools built in prior work on privacy amplification by subsampling (couplings and dominating pairs) in their theoretical analyses.

**Claims And Evidence:**

Most of the claims (described in the summary) are supported with convincing evidence. The claim about utility (CPRS) not being significantly compromised when training models with privacy was confusing to me- I wasn't able to find in the section a detailed overview of the sampling choices that were made when training these models.

**Essential References Not Discussed:**

Most essential references are discussed. In the intro, when referencing graph data, a number of key early citations on privatizing graph properties are missed and other citations are given instead (see the full version of 'Analyzing Graphs With node Differential Privacy' by Kasivishwanathan Nissim Raskhhodnikova Smith in 2015 and the citations therein). However, this is a relatively tangential set of citations and not as relevant to the paper so I don't think it a big omission.

**Experimental Designs Or Analyses:**

Yes, I checked the soundness of all the experimental designs for the claims described in the summary. One question I had was re the sampling scheme used in the utility experiments (see methods and evaluation criteria), but the experimental design seemed largely reasonable.

**Methods And Evaluation Criteria:**

Yes, the proposed methods and evaluation criteria make sense for the problem at hand- one additional evaluation critera that would have been useful when comparing the sampling schemes is not just the privacy level but the utility obtained for the same privacy parameters with different sampling schemes (for e.g. when using sampling wor at the top level, you'd expect to see the same data point more frequently, which may result in worse utility than the deterministic scheme for the same number of epochs, even though you have better privacy).

**Other Comments Or Suggestions:**

1) In the 'other guarantees' section on page 5 I was confused about the point regarding dominating pairs for substitution relations- shouldn't dominating pairs for addition/removal yield ones for substitution (since substitution corresponds to a removal as well as an addition)?

2) On page 6, some more reasoning about why the epoch-level guarantees are better when using subsampling at both the top and bottom levels would be useful (since on the epoch level one might expect that the benefits of top-level subsampling without replacement vanish).

3) I did not understand the explanation on page 7 for why composing many $\lambda>1$ steps  results in larger privacy loss than $\lambda =1$ for a wider range of $\epsilon$ than for a single step, shouldn't the individual $\delta(\epsilon)$ for each step add resulting in similar effects in the many-step case? More explanation here would be very helpful.

4) On page 16 the authors say "In all cases, we observe that λ = 1 improves its privacy relative to other λ > 1 after 100 training steps"- but this claim seems untrue in figure 5 so a more nuanced claim needs to be made (however, it is true in figures 6 and 7).

5) On page 20, the interpolation of the graphs makes them a lot more confusing, for a while I thought $\lambda=16$ was the best in figure 10a until I read the text in detail. Might be worth presenting without interpolation.

**Other Strengths And Weaknesses:**

The main strength of the paper is its thoroughness; the privacy profiles of many combinations of top-level and bottom-level sampling schemes are explored in detail, both analytically and experimentally and natural choices there-in and how to make them are discussed.

I don't see any major weaknesses to discuss other than clarifications about the utility experiments that would be useful. I do think comparing the utilities of the various sampling choices (as opposed to purely the privacy amplification obtained) would have made the paper even more impactful, but that's a natural point for future work.

**Questions For Authors:**

No further questions.

**Relation To Broader Scientific Literature:**

This paper studies privacy amplification via subsampling for more complex multi-level subsampling schemes frequently used in forecasting models with time series data. It fits into a long line of work on privacy amplification via subsampling, but most prior work has been on subsampling for static datasets. The relatively sparse literature on bi-level sampling schemes for private deep-learning has not considered the interplay of privacy amplification effects of applying sampling to multiple levels.

**Theoretical Claims:**

I skimmed the proofs of the theoretical claims in the supplementary material and they seemed reasonable. I did not check all the proofs in detail.

---

> ### Author Rebuttal · Authors · 2025-03-31
>
> Thank you for your review and great questions!
> Please note that we cannot update the manuscript during the current rebuttal phase, but will integrate your feedback as soon as possible.
>
> ### Sampling choices/parameters for model training
> Thank you for pointing out this ommission. **We will of course specify these additional parameters** in Section 5.2.
>
> Our models were trained using top-level sampling without replacement, bottom-level sampling with replacement, and $\lambda=1$, i.e., one subsequence per sequence. For the datasets (`electricity`, `solar`, `traffic`) we used batch sizes $\Lambda=(128, 128, 256)$, noise multipliers $\sigma=(2.0, 4.0, 4.0)$, clipping constants $C=(10^{-4}, 10^{-4},10^{-4})$, and relative context lengths $L_C \mathbin{/} L_F = (1, 4, 8)$.
> The remaining parameters are already specified in Appendix B.2.
>
> ### Effect of Sampling Choices on Utility
> Based on your comment and a suggestion from Reviewer 3, we investigated the effect of different sampling choices on model utility, which we will include in our revised manuscript.
> Due to the character limit on rebuttals, please see our response to Reviewer 3 for details.
>
> ### Prior Work on Graph DP
> Thank you for pointing us to the work by Kasivishwanathan et al (2015).
> We will reference their work - as well as Nissim et al. (2007), who first studied edge privacy - in Section 1.
>
> ### 1 - Dominating Pairs for Substitution
> You are right, any dominating pair for $K$ insertions and $K$ removals is also dominating for $K$ substitutions. We erroneously assumed that the "insert-$K$-remove-$K$" neighboring relation admitted more pairs of datasets, which would have made the statement for Poisson sampling. But this is of course not the case, thank you.
>
> Please note that this was not intended to be a core contribution, but just a corollary of our more general, novel results for multiple sequences/users.
> Our Theorem E.1 (sampling with replacement under group substitution) also remains novel, since no prior work has analyzed WR sampling under group insertion/removal either.
>
> **We will update ll. 264-266 of the "Other Guarantees" paragraph to**:
> "Thus far, dominating pairs for *sampling with replacement* have only been known for individual substitutions. Thus, these guarantees are of interest beyond forecasting".
>
> ### 3 - Why a Single Subsequence is Preferable Under Composition
> You are right. If we were to use pointwise addition of the $\delta(\epsilon,\lambda)$, then the ordering w.r.t. number of subsequences $\lambda$ would stay the same.
>
> However, tight composition works by (1) determining the privacy loss distribution corresponding to the entire privacy profile $\delta : \mathbb{R} \rightarrow [0,1]$, (2) self-convolving the PLD, and (3) converting back to privacy profiles.
> Unlike pointwise addition, this is a non-linear functional operating on entire privacy profiles and may thus change their ordering.
>
> We posit that $\lambda=1$ is better because $\lambda > 1$ allows more catastrophic failures of privacy (sampling multiple sensitive subsequences), whose probability accumulates under composition (see ll.381-387).
>
> We believe that this effect could be better understood by studying the infinite limit of composition using central limit theorems for DP, which provide a simple formula in terms of privacy loss moments.
>
> **To make this clearer to readers, we will update Section 4.5 as follows**:
> "[...], future work may also want to investigate them analytically, e.g., via central limit theorems (Sommer et al. 2019), in particular to understand why $\lambda=1$ offers better privacy under composition."
>
> ### 2 - Why Top-Level Sampling is Preferable
> As with the previous point, tight composition does not have a nice  analytical expression that could be used to answer this question. Again, we believe that considering the infinite limit of composition (here: epoch length) could be a promising direction towards understanding why top-level subsampling offers better privacy despite requiring more compositions.
>
> ### 4 - Relative Improvement under Composition
> Thank you for pointing out this inconsistency. **We will use the following formulation instead**:
> "We observe that, after 100 training steps, $\lambda=1$ offers better privacy than $\lambda \in \{2,4\}$. It also offers better privacy than $\lambda \in \{8,16\}$ for small top- and bottom-level sampling probabilitites."
>
> We will also include a column for $1000$ steps, where $\lambda=1$ actually outperforms $\lambda=16$ across all $\epsilon$ for the same parameterization as in Fig. 5:
> https://figshare.com/s/4292ea34e216427a0ded
>
>
> ### 5 - Interpolation on Page 20
> We agree that interpolation makes the figures on p. 20 somewhat hard to read, and will remove it:
> https://figshare.com/s/9a3282037c6d268cd37a
>
> ---
>
> Thank you again for your review!
> Please let us know if you have any additional questions during the discussion period.
>
> ---
>
> Nissim et al. Smooth sensitivity and sampling in private data analysis. STOC 2007.

---

### Decision · Program_Chairs · 2025-05-01

**Decision:**

Accept (spotlight poster)

**Comment:**

The paper presents a theoretical and empirical study of privacy amplification in differentially private training of models on time series data. It analyzes how structured subsampling affects differential privacy guarantees. Understanding this is particularly important since existing privacy amplifications mostly hold for unstructured (random) sampling from unstructured/unordered data, which is not true for time series data. The authors derive new amplification results tailored to this sequential data structure.  Empirical results demonstrate that these new techniques enable training forecasting models under strong privacy guarantees without substantially degrading utility.


Reviewers found the paper’s theoretical contributions to be thorough and well-motivated. The reviewers also found the analysis thorough with sound proofs. Multiple reviewers praised the work for extending privacy amplification techniques into a new domain, relevant for sensitive applications (e.g. analyzing time series data in healthcare and finance). The reviewers also raised several concerns. For example, multiple reviewers mentioned that the empirical evaluation could have better isolated the impact of the new privacy analysis on model utility, independent of the subsampling choices.
Also, there was a concern about the breadth of the experimental comparisons. However, the authors response addressed most of the reviewers concerns, and the reviewers agreed on the overall significance of the contributions of the paper.